# Estimations of statistical dependence as joint return period modulator of compound events. Part I: storm surge and wave height.

Thomas I. Petroliagkis

Joint Research Center, Ispra, I-21027, Italy

*Correspondence to*: Thomas I. Petroliagkis (thomas.petroliagkis@ec.europa.eu)

**Abstract.** The possibility of utilising statistical dependence methods in coastal flood hazard calculations is investigated since flood risk is rarely a function of just one source variable but usually two or more. Source variables in most cases are not

independent as they may be driven by the same weather event, so their dependence, which is capable of modulating their joint return period, has to be estimated before the calculation of their joint probability. Dependence and correlation may differ substantially from one another since dependence is focused heavily on tail (extreme) percentiles. The statistical analysis between surge and wave is performed over 32 river ending points along European coasts. Two sets of almost 35-year hindcasts of storm surge and wave height were adopted and results are presented by means of analytical tables and maps referring to

both correlation and statistical dependence values. Further, the top 80 compound events were defined for each river ending point. Their frequency of occurrence was found to be distinctly higher during the cold months while their main low-level flow characteristics appear to be mainly in harmony with the transient nature of storms and their tracks. Overall, significantly strong values of positive correlations and dependencies were found over the Irish Sea, English Channel, south coasts of the North Sea, Norwegian Sea and Baltic Sea, with compound events taking place in a zero-lag mode. For the rest, mostly positive

moderate dependence values were estimated even if a considerable number of them had correlations of almost zero or even negative value.

## 1 Introduction

In the coastal and inter-tidal zones, high waves and extreme tidal surge events can occur simultaneously with extreme precipitation events and high river flows, leading to increased flood severity, duration or frequency as highlighted in Svensson and Jones (2002, 2003, 2004a, 2004b, 2005), Hawkes and Tawn (2000), Hawkes et al. (2005). These interactions are generally referred to as coincident or compound events (IPCC, 2012). In the current Part I, compound events of surge and wave are those events that coincidently are above a certain upper percentile criterion (representing a critical threshold). A key component of any coincident event assessment is to understand the historical relationships between the different factors that may lead to a compound flood event. However, assumptions are often made regarding how these different factors and variables coincide or combine, typically leading to either an under- or over-estimation of the probability of flooding (Coles et al., 2000). In reality, while some events may indeed occur independently from one another, others involve an interaction, or may have compounding consequences when they occur simultaneously, and need to be treated as partially dependent for the estimation of their joint probability or joint return period (https://www.niwa.co.nz/natural-hazards/faq/what-is-a-return-period).

Joint probability values provide the likelihood of source variables taking high values simultaneously resulting in a situation where flooding may occur. Acceptance of joint probability methods has been relatively sparse so far mainly due to the lack of information on dependence among source variables and the intrinsic difficulty in usage and interpretation of the methods as pointed out in Australian Rainfall & Runoff Project 18 (2009), Bevacqua et al., (2017). The main concept of dependence as presented by Reed (1999) refers to the tendency for critical values of source variables to occur at the same time resulting in an increase in frequency of an extreme event. This is because dependence is able of modulating the joint return period as documented in Hawkes (2004), Meadowcroft et al. (2004), White (2007), Australian Rainfall & Runoff Project 18 (2009).

The method for estimating the probability of extreme values from a single variable has been well understood and documented (Coles, 2001). Such probability is usually expressed in the form of a return period. In a similar way, the (joint) probability of two variables producing high or extreme values together, assuming to be fully independent or fully dependent, is also considered straightforward as explained in Defra TR0 Report (2003). On the other hand, examples of coincident flood event studies, which incorporate a measure of the relationship between the input variables, are generally limited due to the complexity of the broader coincident events problem (Bevacqua et al., 2017). Assessing the probability of flooding from the joint occurrence of high waves and high sea level values for instance is not an easy process, as high waves and storm surge tides may be attributed to the same prevailing storm system; thus, independence cannot and should not be assumed. Further, it is more complicated to estimate such conditional (joint) probabilities than those referring to totally independent events (http://onlinestatbook.com/2/probability/basic.html). However, some approachable and user-friendly methods seem to exist for quantifying the statistical dependence between the input variables as noted in Hawkes (2004), White (2007) and applied by Zheng et al. (2013 & 2014) and Klerk et al. (2015).

In the case of independent events, the chance of one event occurring is not changed by the occurrence of the other event. However, if the occurrence of one event is dependent on the occurrence of a second event then the events are termed conditional even if their correlation might be equal to zero. It should be stressed that correlation and dependence might differ substantially

from one another. Two source variables may have low correlation but there may exist considerable statistical dependence between them referring to their upper percentiles where actually extremes reside. Further, it should be well established by now that correlation coefficients measure the degree of straight line or linear relationship only and that there are situations in which correlations are zero but where strong nonlinear relationships exist among variables (Drouet Mari and Kotz, 2004).

Assuming independence between input variables might underestimate considerably the likelihood of flooding resulting in higher risk for the coastal community, since the conditional probability of both events occurring at the same time is different from the product of their individual probabilities (Blank, 1982). Similarly, assuming total dependence could be too conservative (Beersma and Buishand, 2004). What someone should anticipate is the fact that dependence is likely to occur when different processes are linked to some common weather (forcing) conditions. It may also arise when the same process is

studied at different spatial locations or over different periods (Coles et al., 2000). In an estuarine or riverine area, an example would be a storm accompanied by high winds and intense precipitation phenomena. For such cases where two (or more) variables, capable of producing high-impact events, are not totally independent or totally dependent, but may be partially dependent, probabilistic approaches are limited in both their reliability and scope (White 2007).

In this work, the possibility of utilizing statistical dependence methods in coastal flood hazard calculations is investigated, since an estimation of the joint probability (joint return period) is necessary for the calculation of compound flood hazard in a coastal area. Such an approach points to taking into account the variability and exact nature of extreme conditions. The basic idea behind joint probability theory is to identify extreme data within each of the input variables and statistically correlate their linkages and risk of simultaneous occurrence. Therefore, it seems quite important to find an appropriate way to undertake this

task. Understanding such risks, created by the combination of extreme events, is crucial for the design of adequate and cost effective river and coastal defences as well as for the true estimate of flood risk as highlighted in Merz et al. (2009), Australian Rainfall & Runoff Project 18, (2009).

The current work focuses on data preparation, parameter selection, methodology application and estimation of both correlation

and statistical dependence between source variables. It also focuses on the prevailing (higher frequency) and dominant (higher intensity) low-level wind conditions over a set of preselected (top 80) extreme compound events. The critical time period during which such extremes take place is also analysed based on monthly frequency values of occurrence. The dependence analysis utilises 32 river ending points selected to cover a variety of geographical areas along European coasts. The variable-pairs presented in this report, which include enough information for calculations, are storm surge and wave height, relevant to

most coastal flood defence studies. Two main time intervals were considered for the estimation of maximum values: the half-day interval (max12) and the one-day interval (max24).

This study represents the first part (i.e., Part I) of the investigation while Part II (storm surge and river discharge) and Part III (wave height and river discharge) are to follow. The reasoning behind such a separate investigation (by parts) is to allow the reader for a deeper and better understanding of the interaction between different components contributing to a compound coastal event.

In Sect. 2, ~~the concept and implications of estimating statistical dependence are documented, while in Sect. 3, the~~ data and methods used in this study are presented~~. Results~~ while results are shown in Sect. 3~~., 4. while discussion~~ Discussion and conclusions are contained in Sect. ~~5.~~ 4 and Sect. 5 respectively.

## ~~2 Statistical dependence (χ)~~

## 2 Data and methodology

## 2.1 Statistical dependence ($\chi$)

The main concept of the so-called dependence measure $\chi$ (chi) is related to two or more simultaneously observed variables of interest – such as in our case storm surge and wave height – known as observational pairs. If one variable exceeds a certain extreme (high-impact) threshold, then the value of $\chi$ represents the risk that the other variable will also exceed a high-impact threshold as explained in Hawkes (2004), Svensson and Jones (2004a & 2004b), Petroliagkis et al. (2016).

Following Coles et al. (2000), if all of the extreme observations of two variables exceed a given threshold at the same time, this indicates total dependence ($\chi = 1$). If the extreme observations of one variable exceed a given threshold but the second variable does not, this indicates total independence ($\chi = 0$). Similarly, if the extreme observations of one variable exceed a given threshold but the other variable produces lower observations than would normally be expected, this indicates negative dependence ($\chi = -1$). In practice, hydro-meteorological analyses based on real data often lead to an assessment of complete independence that could result to an under-estimation of the joint probability of concurrent extreme events, whereas, an assumption of complete dependence could result to an over-estimation of joint probabilities (Beersma and Buishand, 2004). In reality, as variables reach their extreme values, a special methodology of estimating statistical dependence could be utilised as the one documented in Buishand (1984). A brief description of this method based on Coles et al. (2000) is contained in the Statistical Supplement while the basic theory behind the utilisation of an optimal copula function refers to Nelsen (1998), Joe (1997), Currie (1999), Wahl et al. (2015).

## 2.2 River Ending Points

The current statistical (dependence) analysis is focused over 32 river ending points that have been selected to cover a variety of riverine and estuary areas along European coasts. These points were selected mainly for their proximity to tide gauge recorders although not many observations were found suitable to be exploited due to the lack of long-period coincident wave (buoy) observations in the near-by area. The sea areas used in the study refer to the Mediterranean Sea (central and north Adriatic Sea, Balearic Sea, Alboran Sea and Gulf of Lion), West Iberian, North Iberian, Bay of Biscay, Irish Sea, Bristol Channel, English Channel, North Sea, Norwegian Sea, Baltic Sea and Black Sea. A map showing the position of RIEN (RIver ENding) points used in the study is shown in Fig. 1. Additional details can be found in Table 1 of the Technical Supplement containing the exact location (lat, lon) of all RIEN points.

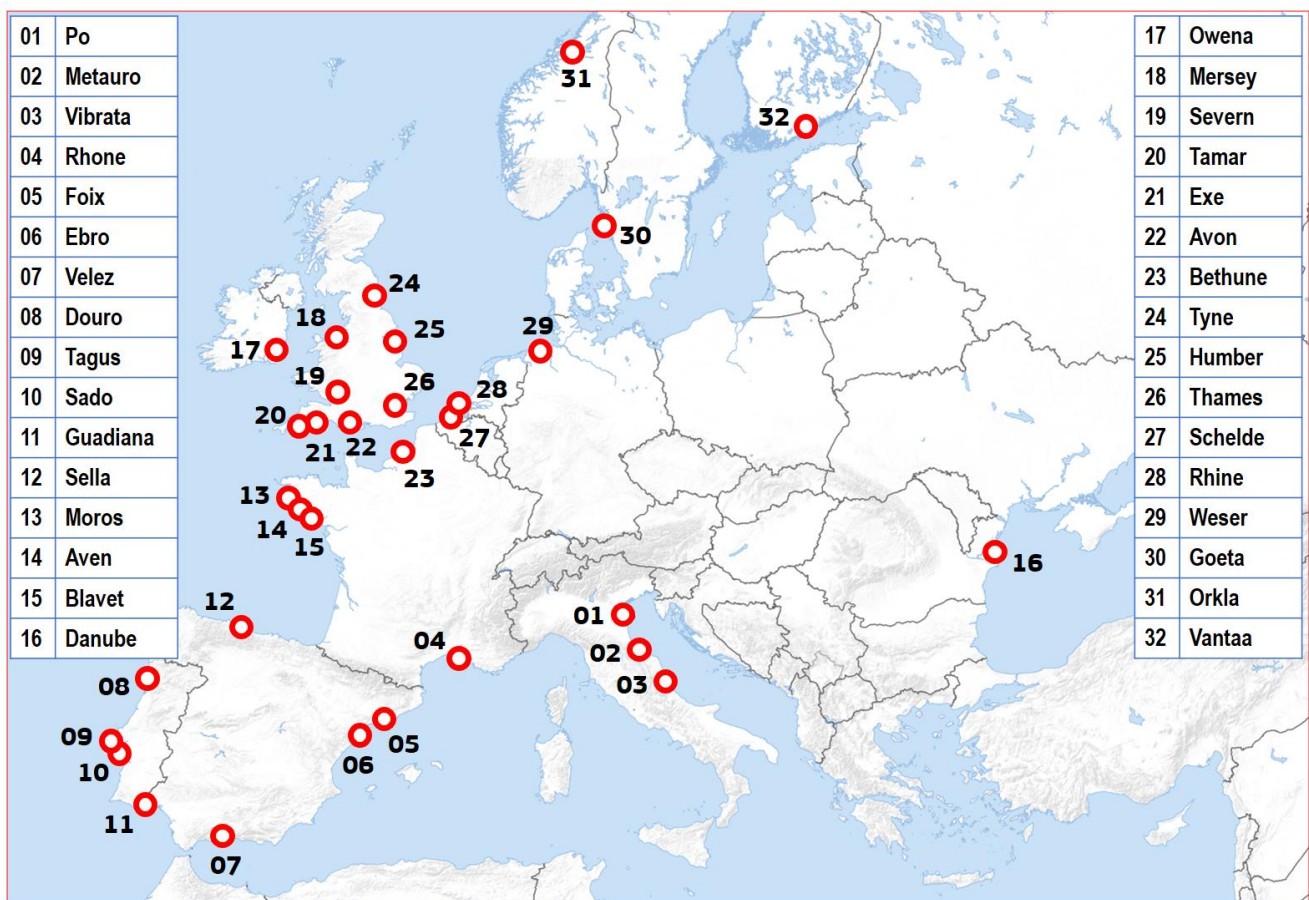

| 01 | Po |
|----|----|
| 02 | Metauro |
| 03 | Vibrata |
| 04 | Rhone |
| 05 | Foix |
| 06 | Ebro |
| 07 | Velez |
| 08 | Douro |
| 09 | Tagus |
| 10 | Sado |
| 11 | Guadiana |
| 12 | Sella |
| 13 | Moros |
| 14 | Aven |
| 15 | Blavet |
| 16 | Danube |

| 17 | Owena |
|----|----|
| 18 | Mersey |
| 19 | Severn |
| 20 | Tamar |
| 21 | Exe |
| 22 | Avon |
| 23 | Bethune |
| 24 | Tyne |
| 25 | Humber |
| 26 | Thames |
| 27 | Schelde |
| 28 | Rhine |
| 29 | Weser |
| 30 | Goeta |
| 31 | Orkla |
| 32 | Vantaa |

**Fig. 1.** Positions of the 32 RIEN (river ending) points used in the study. Names refer to river names. Exact positions (lat, lon) of RIEN points are given in Table 1 (Technical Supplement).

As already mentioned long-period water level data coinciding with wave observations directly or very close to the exact sites of interest (RIEN points) were not available with the exception of the Rhine River (RIEN). For this RIEN, concurrent (close-by) observations with no gaps of sea level, astronomical tide, storm surge, and wave height from a close-by wave buoy were available for a period of about 3 years (1,114 days).

### 3.1 2.3 Storm surge hindcasts

Storm surge is an abnormal rise of water generated by a storm, over and above the predicted astronomical tide values (http://www.nhc.noaa.gov/surge/faq.php). In "observations mode", storm surge is calculated as a residual by subtracting harmonic tidal predictions from the observed sea level (Horsburgh and Wilson, 2007) Such "residual" may contain surge, tide-surge interaction, harmonic prediction errors and timing errors. and depending on inclusion of the nonlinear interaction of tides and storm surges, this "residual" can include tide-surge interaction as well. In this study, tide-surge interaction, harmonic prediction errors and timing errors were not taken into consideration in this study. On the other hand (e.g. in hindcast model simulation mode) a similar "residual" refers to the genuine meteorological contribution to sea level that represents the storm surge term. It should pointed out that the effect of wind and atmospheric pressure (inverse barometric effect) are contained in both the "residual" and storm surge terms. Based on this, it becomes clear that all data (storm surge) sets used in the study contain the effect of the inverse barometric effect besides the effect due to wind. This is the reason why the dedicated model (Delft3DFlow) uses as input both ERA-Interim (Dee et al., 2011) wind and pressure fields.

For assessing dependence, a full set of coincident observation data is needed over a relatively long time period (at least five years) for the primary variables, as pointed out in Defra TR2 Report (2005). Such a demanding requirement is too difficult if not impossible to be fulfilled only by observational data over all 32 RIEN points, so, the methodology of simulating data observations by modelling (hindcasts) was applied resulting in a set of long period model simulations (hindcasts) for the two primary variables (surge and wave).

For storm surge hindcasts, the Delft3D-Flow hydrodynamic module of the open source model Delft3D (Deltares, 2014) was used to compile storm surge time-series due to the combined effect of the wind and the atmospheric pressure gradient. The model (Delft3D-Flow) has been used successfully in similar applications (hindcasts) in the past (Sembiring et al., 2015). In our case, 3-hourly long-term storm surge series for the 32 RIEN points were compiled from a similar hindcast set to that used in Vousdoukas et al. (2016) for estimating projections of extreme storm surge levels along Europe. This was obtained by forcing the Deflt3D-Flow module by 6-hourly wind and pressure fields retrieved from the ECMWF ERA-Interim reanalysis data set (Dee et al., 2011). The ERA-Interim (ERAI) is a global atmospheric reanalysis from 1979 to present. ERAI's main products include global atmospheric and surface parameters from 1 January 1979 to present, at T255 spectral resolution (~75 x 75 km) on 60 vertical levels.

Storm surge hindcasts (1 January 1980 to 30 November 2014) span a total interval of 12,753 days (~35 years), having a time separation of 3 hours with a spatial resolution of about 0.2 degrees (~25 x 25 km) along the European coastline and the NE Atlantic Ocean areas. Hindcast storm surge levels were validated with measurements from 110 tide gauges from the JRC Sea Level Database (http://webcritech.jrc.ec.europa.eu/SeaLevelsDb). Details can be found in Vousdoukas et al. (2016). The relative rms (root mean squared) error for more than 105 stations was found to be less than 20% and for more than 60 stations less than 15%. More specifically, the validation performance of hindcasts is contained in the scatter plot of Fig. 4 (Vousdoukas et al., 2016) for both RMS error in m (a) and as a percentage of the SSL (Storm Surge Level) range (b) for all the available tidal gauge stations. Further, even if there were cases where some extreme storm surge levels were underestimated by the hindcasts, the overall model performance is considered to be satisfactory.

### 3.2 2.4 Wave height hindcasts

In many applications, a selection of heights and periods of the higher waves in a wave train seem to be of practical significance. For this reason, the average height of the highest one-third of the waves, after eliminating the ripples and waves of height less than one foot is considered as a useful statistical measure. This average is commonly named as "significant wave height" (Sverdrup and Munk, 1946) and it is utilised in the current study.

As in the case of storm surge, global fields of 3-hourly (significant) wave data were assembled by utilizing a set of hindcasts produced with the latest stand-alone version of ECWAM wave model (for details see Bidlot et al., 2006, Bidlot, 2012 and ECMWF, 2015, Philips et al., 2017). The ECWAM model was run on a 0.25 degrees lat-lon global grid (~28 x 28 km) with fixed water depth (mean bathymetry, i.e., no surges or tides) being forced by neutral wind fields (as forcing terms) extracted from the ERAI reanalysis. Due to such resolution limitations, the model may not represent the best source of wave data for a particular single coastal location, but it does offer consistent coverage over the area of this study within an acceptable degree of accuracy. The reason is that even if model resolution does not seem capable of simulating local coastal topographical details, the main characteristics of the large-scale wave evolution are expected to be captured (based on wave in situ observations data provided by Dr Jean-Raymond Bidlot (ECMWF) and used for validation and compiling Fig. 2 and Fig. 3). For more details on wave validation and verification data, see https://www.ecmwf.int/en/newsletter/150/meteorology/twenty-one-years-wave-forecast-verification.

For each RIEN point, 3-hourly hindcast wave data time series for the period of 1 January 1979 to 31 December 2015 were assembled (13,149 days) by considering the closest model grid (sea) point, with no missing records. The resulted records consist of significant and maximum wave height values, mean wave period and mean wave direction. The ECWAM model has been configured in its CY41R1 parametrization cycle employing 30 frequencies and 36 directions for the wave spectra (ECMWF, 2015).

Validation of wave hindcasts was made utilising a set of available data collected from 101 buoys over European and NE Atlantic sea areas during the period from 1996 to 2015. The exact position of the buoys used in validation is shown in Fig. 2 (also in Fig. 3). Both bias and rms error scores were considered and results are ~~also~~ shown in Fig. 2, for bias ~~(upper panel)~~ and

5    Fig. 3, for rms error ~~(lower panel)~~. Both scores (bias & rms error) suggest that the model's performance was satisfactory, although bias is lagging slightly in quality compared to rms error mainly due to the weak ERA-Interim winds that seems to affect the bias more than rms error. Additional validation details focusing on extremes are contained in Sect. 2 of the Technical Supplement.

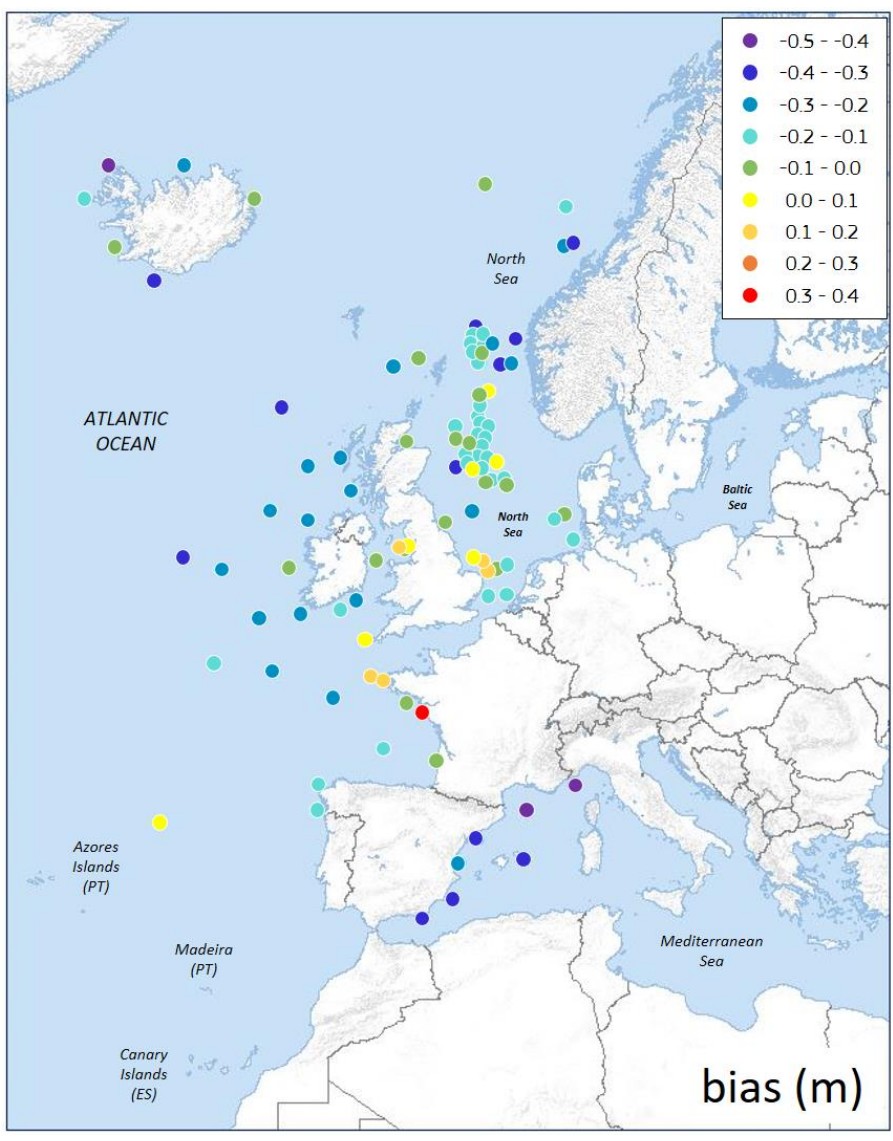

10                               **Fig. 2.** Bias values (m) for wave hindcasts during 1996 to 2015.

It should be noted that the maximum common time interval of 12,753 days (~35 years) for surge and wave variables was considered for the statistical (dependence) analysis over the 32 RIEN points of this study. Further, as already stated above, both sets of hindcasts had already been validated (Vousdoukas et al., 2016, Philips et al., 2017), so, emphasis was given in demonstrating the methodology of estimating the type and strength of statistical dependence. Such an investigating approach

5    was performed initially over the ending point of Rhine River (NL) with very satisfactory results (see next Sect. 2.5) while the same approach (of estimating statistical dependence) was adopted for the rest of ending points (RIENs) of the study.

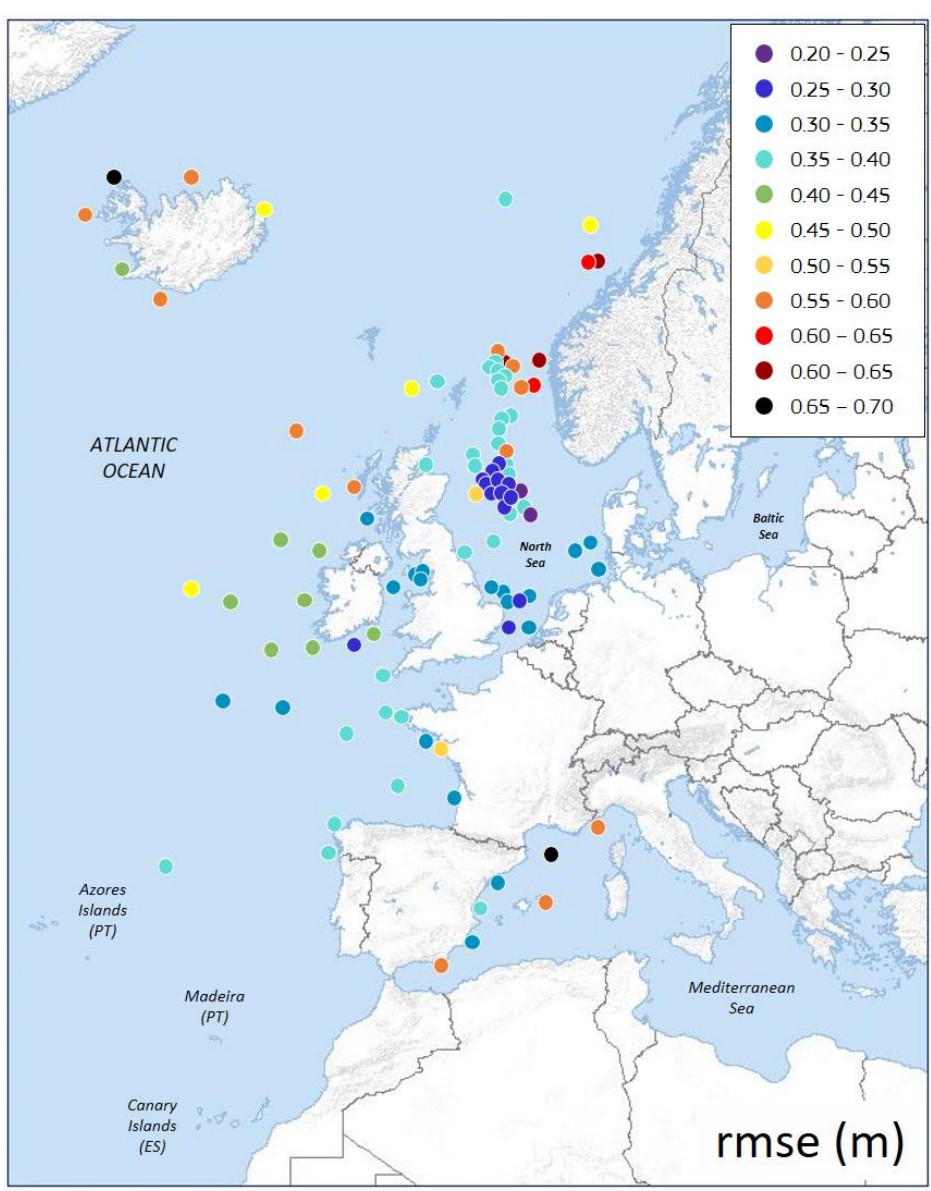

**Fig. 3.** RMSE values (m) for wave hindcasts during 1996 to 2015.

### 3.3 2.5 Local Joint validation for the RIEN of Rhine River (NL)

A joint validation of surge and wave hindcasts utilising a relatively long series of surge and wave observations close to Rockanje (RIEN of Rhine River) was performed for testing the quality of hindcasts during the common period of observation records. For this task, storm surge observations close to Rockanje were downloaded from MATROOS (Deltares Multifunctional Access Tool foR Operational Oceandata Services database - http://noos.deltares.nl/) database. Such surge observations were recorded by the near-by Hook van Holland (HvH) tide gauge recorder positioned at about 15 km northeast of Rockanje (as depicted in Fig. 4). Referring to observation parameters, sea level is the recorded still (i.e. in the absence of waves) water level, and surge is the difference between sea level and predicted tide for that time and location. Similarly, a set of significant wave height observations were retrieved from the close-by wave buoy platform of Lichteiland Goeree I (LiG) moored in North Sea at about 55 km northwest of Rockanje (see details in Fig. 4).

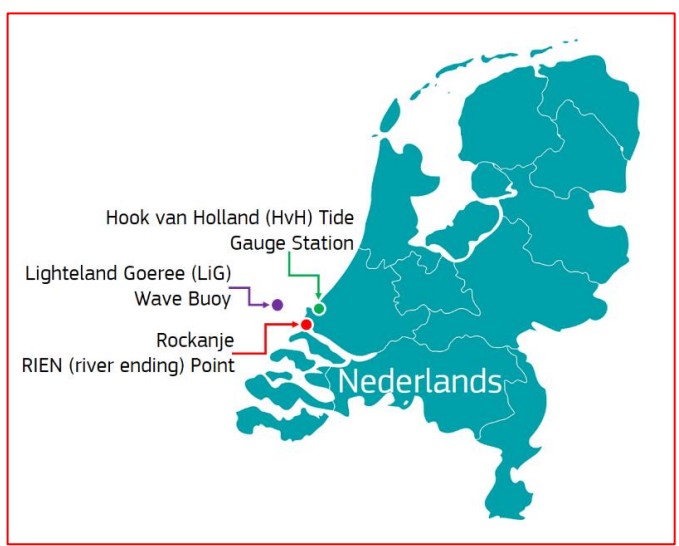

**Fig. 4.** Position of HvH tide gauge station, Rockanje RIEN point and Lichteiland Goeree I (LiG) wave buoy (NL).

A common time interval of 1,114 days (from 22 September 2010 until 9 October 2013) with no gaps was selected for validating surge (SUR) and wave (WAV) hindcast data sets. Such surge and wave hindcasts were different to the ones referring to Rockanje RIEN point since they were performed as close as possible to the exact positions of HvH tide gauge recorder (SUR) and LiG wave buoy (WAV) for obvious reasons. Both types of observation, i.e., surge, over HvH and wave height, over LiG were made on an hourly basis, so, daily (max24-hour) and half-day (max12-hour) maximum values were calculated. In harmony with observations, 3-hourly based storm surge and 3-hourly wave hindcasts were transformed to daily (max24) and half-day (max12) maximum hindcast values. Daily maximum (max24) levels of SUR hindcasts for HvH were compared against daily maximum observations measured by HvH tide gauge over the reference period of 1,114 days. The closest storm surge point used in our analysis was situated ~20 km to the north of HvH (North Sea).

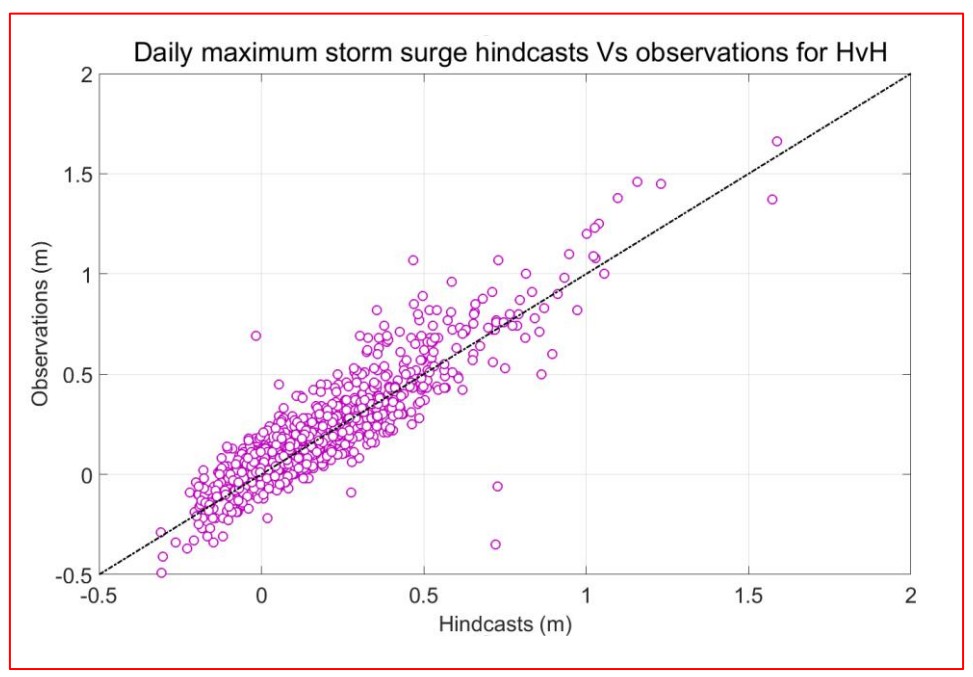

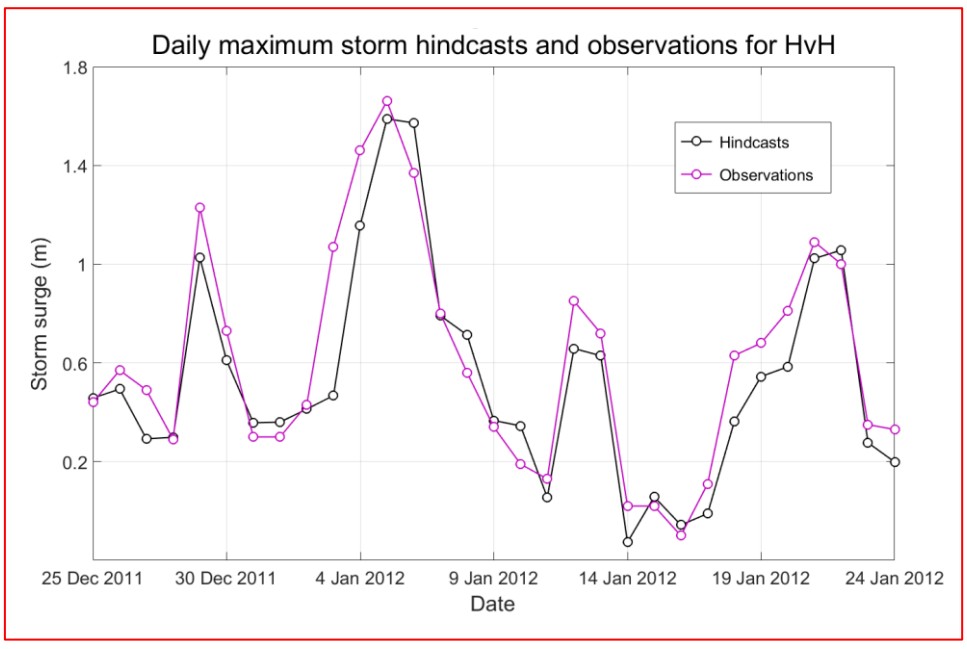

**Fig. 5.** Scatterplot of surge hindcasts against observations for HvH (upper panel) and a subsection of hindcast and observation values during 25 December 2011 to 24 January 2012 (lower panel).

Fig. 5 (upper panel) contains the scatterplot of surge hindcasts against observations in max24 mode. It appears that SUR hindcasts are in most cases lower than their corresponding observation values. This difference might be attributed to the relatively low temporal and spatial resolution of ERA-Interim forcing terms, although hindcasts overall were found capable of
coping well with both the timing and magnitude of extremes.

An example of such an extreme taken place on 5 January 2012 is shown in Fig. 5 (lower panel) that contains a subsection of SUR hindcasts plotted together with surge observations over HvH. This extreme was selected as a multi-purpose demonstrating example (see also Fig. 7 for corresponding maxima of WAV values during the same period and Sect. 3.5 where the de-
clustering technique of Peaks-Over-Threshold approach is explained). It is evident that although SUR hindcasts have a tendency to underestimate observations, the model simulations seem able to resolve both the magnitude and the duration of storm events relatively well. The storm surge peak of 5 January 2012 was found to be linked to a very intense extratropical cyclone (Storm Ulli / Emil) affecting the greater area of the North Sea. The position and details of the storm are contained in the surface weather map of 12UTC of 3 January 2012 shown in Fig. 6 (upper panel). The corresponding (12UTC) satellite
picture capturing Storm Emil is contained in the same Fig. 6 (central panel).

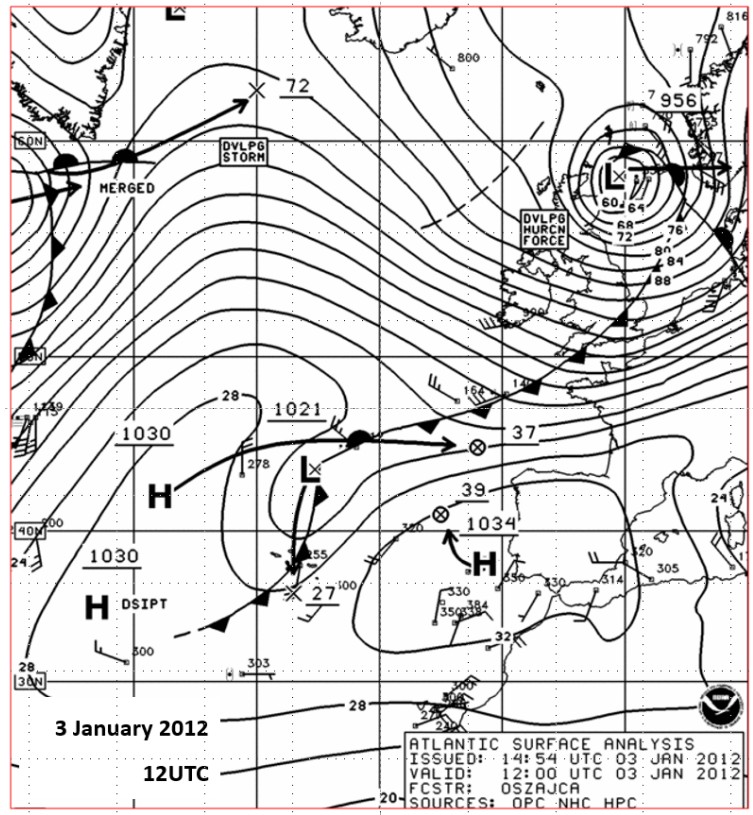

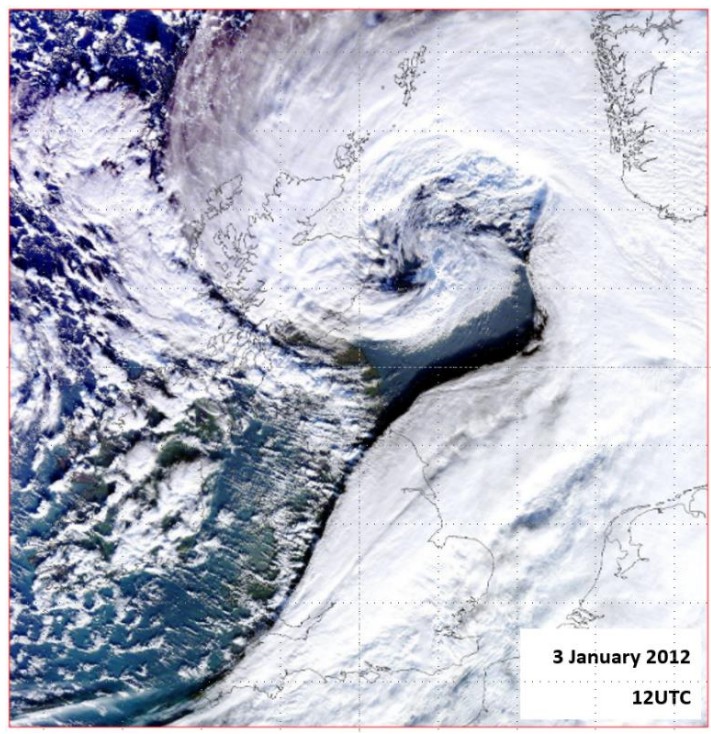

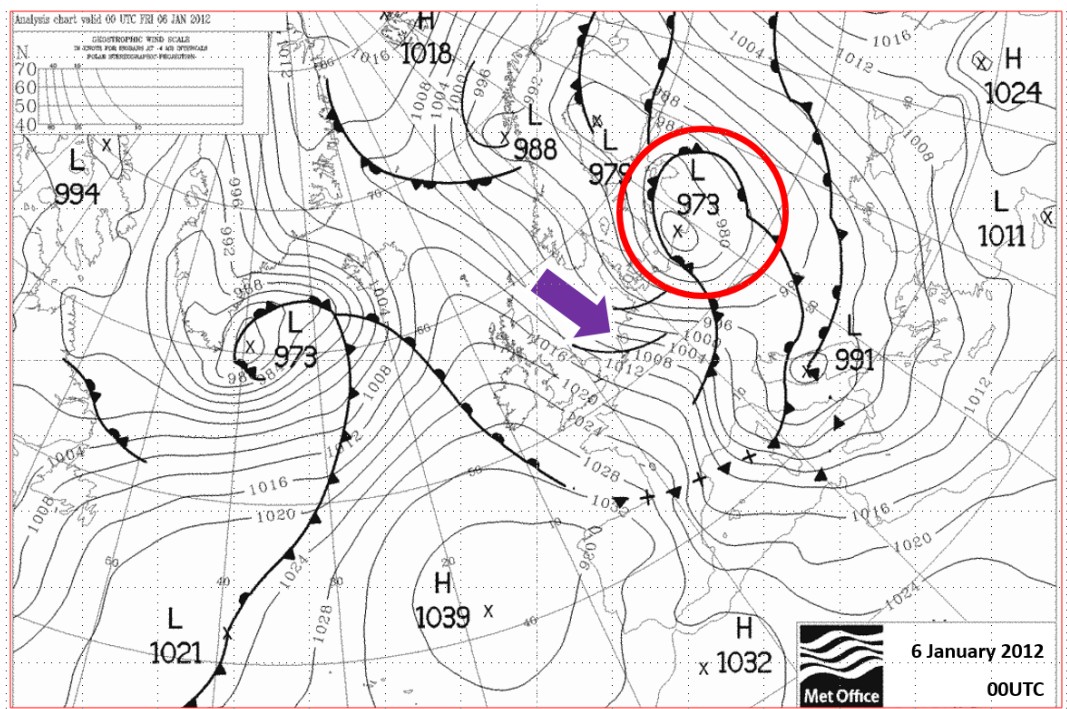

**Fig. 6.** Surface map (upper panel) and its corresponding satellite image (middle panel) valid for 3 January 2012 12UTC. Surface weather map in the late hours (midnight) of 5 January 2017 (lower panel).

Studying closely both surge hindcast and observation values referring to Storm Emil (lower panel of Fig. 5) it seems that hindcasts could resolve and simulate well both the phase and the magnitude of such an extreme event. It is important to realise that such events are linked to intense pressure gradients such as those clearly seen in Fig. 6 (lower panel) prevailing during the late-night hours of 5 January 2012. Besides the intensity of pressure gradients, the orientation of isobars that was almost vertical to the coasts of Holland (as indicated by a purple arrow) it was also contributing strongly to the extremity of the event.

Overall, storm surge hindcasts were found to have a negative bias (defined as difference between the mean of hindcasts and the mean of observations; see details in http://www.cawcr.gov.au/projects/verification/) of about -2.65 cm. It was also found that hindcast and observation values exhibit a very strong correlation reaching a value of ~0.90, while slightly lower correlation was found for max12 (0.88).

Similarly, daily maximum (max24) values of significant wave height (WAV) were compared against daily maximum values of observations measured at the LiG wave buoy platform as shown in Fig. 7 (upper panel). The closest wave model point used in our analysis was suited ~9.5 km northwest of the position of the LiG platform. As in the storm surge case, WAV hindcasts are in most cases lower than their corresponding observation values.

As mentioned already, this might be due to the smoothness of ERAI forcing terms not possessing the required resolution to resolve the exact magnitude of wind components. The systematic bias of wave hindcasts was found to be relatively small, being equal to 20.30 cm. Wave hindcasts were found to exhibit a very strong correlation value (~0.92) to observations, while similar (slightly lower) correlation was found for max12 data pairs (~0.90).

As in the case of SUR hindcasts, Fig. 7 (lower panel) contains a subsection of time series of both WAV hindcasts and observations (with dates being identical to the previous SUR case). WAV hindcasts appear to be lower than observations and besides the unavoidable smoothness of ERAI fields, another explanation for such deviation might be the proximity of the LiG buoy to the coast. It is well known that enclosed areas and near-shore locations are indeed much more difficult to model (http://www.ecmwf.int/en/newsletter/150/meteorology/twenty-one-years-wave-forecast-verification).

Referring to the example (SUR) already presented in the lower panel of Fig. 5, the storm surge extreme of 5 January 2012 was found to be in harmony with the significant wave height extreme observed during the same max24 interval (shown in the lower panel of Fig. 7). This is also an indication that storm surge and wave compound extremes are linked to the same weather system (Storm Emil) with a clear tendency to take place in a zero-lag time mode over the south coasts of the North Sea.

The obvious agreement between surge hindcasts and observations (Fig. 5) is a clear indication of the model's (Delft3D-Flow) capability to simulate efficiently observations (over the whole spectrum of observations) in hindcast mode having as input parameters (wind components and mean sea level pressure) from the ECMWF ERA-Interim reanalysis data set. Same wise the obvious agreement between wave hindcasts and observations (Fig. 7) is a clear indication of the model's (ECMWF / ECWAM) capability to simulate efficiently observations in hindcast mode having as input parameters (wind components) from the ECMWF ERA-Interim reanalysis data set.

Indicative examples of such capabilities can be seen in Table 4 and Table 5 of the Technical Supplement revealing that hindcasts above all were capable of identifying and resolving all seven (7) compound events (based on 98.5% percentile threshold) that took place during the common time interval of 1,114 days over HvH area of interest.

Since this study is focused over maxima taken place over 12- and 24-hours based on 3-hour set of hindcast values, timing errors were investigated over Rhine River (NL) ending point and the overall conclusion has been that hindcasts were able to pick up similar (to observations) maxima during both the 12-and 24-hour intervals. Details and examples of the capability of hindcasts to identify and resolve compound events of surge and waves are contained in Sect. 3 of the Technical Supplement.

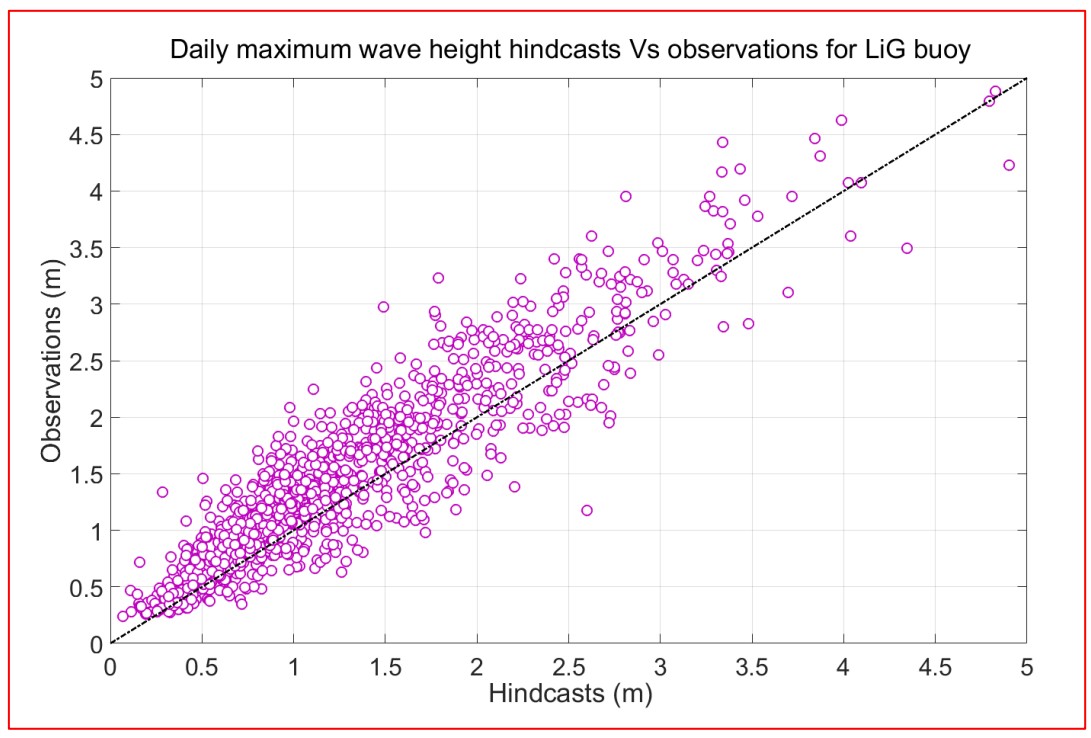

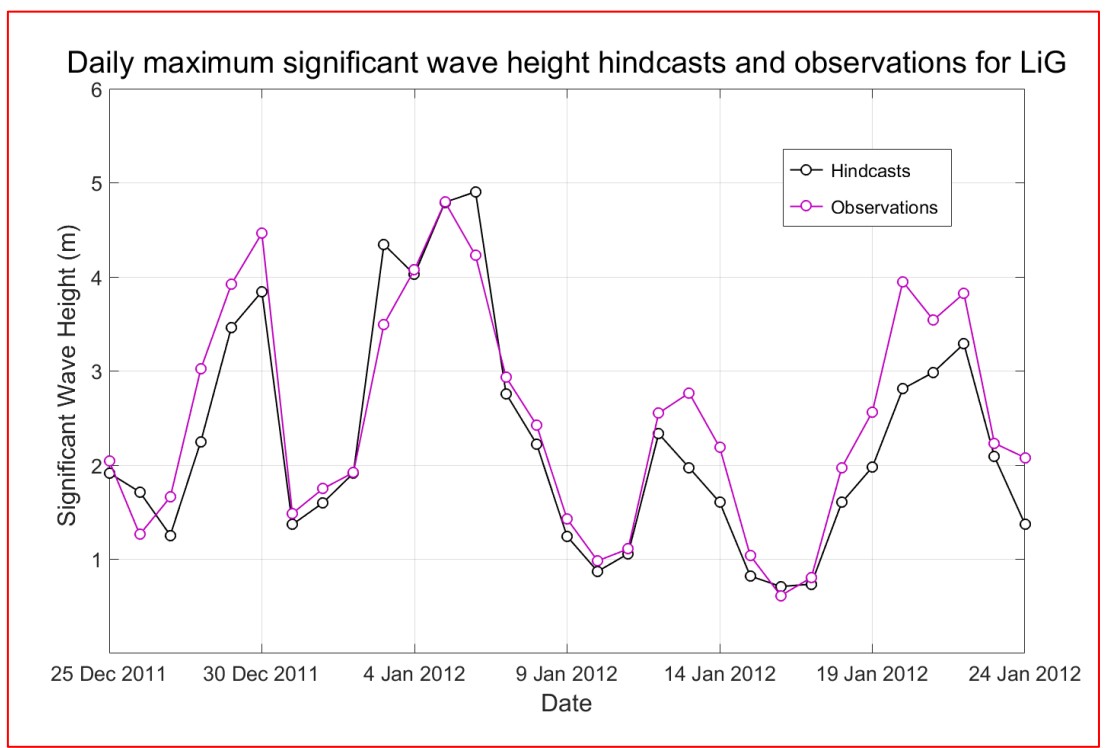

**Fig. 7.** Scatterplot of wave hindcasts against observations (upper panel) and a subsection of hindcast and observation values during 25 December 2011 to 24 January 2012 (lower panel).

Further, an extra investigation based on extreme values of observations (during the common time interval of 1,114 days)
5    exceeding a variety of percentile values for the RIEN of Rhine River revealed that both storm surge and their corresponding wave height hindcasts were able to capture (resolve) almost all of the 12- and 24-hour extremes (not necessarily compound ones) on the same (correct) day but with a weaker intensity (i.e., with a correct footprint of lesser intensity).

## 4 3 Results

The main tools for estimating statistical dependence ($\chi$) are briefly summarised in the next section (Sect. 3.1). Besides the
10    ability of surge and wave hindcasts to simulate correctly observations over HvH tide gauge and LiG wave buoy platforms respectively (as analysed in Sect. 2.5), their potential for resolving the correct type and strength of both correlation and dependence between primary variables in a joint (compound) mode environment is investigated over a common period of 1,114 days in Sect. 3.2. Referring to the full span of hindcasts, analytical maps and tables have been assembled containing both correlation and dependence values between surge and wave over the 32 RIEN points considered in this study.

Both correlation and dependence values were estimated over maximum values of surge and wave during 12- and 24-hour intervals (labelled as max12 and max24 respectively). These results are presented in Sect. 3.3 (southern European areas) and Sect. 3.4 (northern European areas). An evaluation of the low-level flow during the top 80 extreme compound events utilising wind rose diagrams is contained in Sect. 3.5. The critical period (of the year) for such high-impact events to take place is also assessed by considering monthly frequencies of occurrence.

### 4.1 3.1 Main tools for estimating statistical dependence

The main tools for assessing dependence between surge and wave has been a set of Matlab routines (mat_chi) for estimating the asymptotic behaviour of statistical dependent variables. Other Matlab routines such as mat_chibar (see details and examples in the Statistical Supplement) for assessing the asymptotic behaviour of statistical independent variables were also used and main findings are contained in Table 6 and Table 7 of the Technical Supplement. Besides Matlab functions additional routines from the statistical package R, namely "taildep" of module extRemes and "chiplot" of module evd (Extreme Value Distributions) were used for estimating and inter-comparing $\chi$ values (see details in the Statistical Supplement).

An optimal threshold of ~2.3 events on a yearly basis was found to was able to provide quite stable dependence graphs (see details in the Statistical Supplement) while the maximum strength of almost any compound (surge and wave) event tends was found to take place during the same 24-hour (max24) time or during the same 12-hour (max12) period corresponding to zero-lag mode. Exceptions were found for Rhone, Ebro, Danube, Thames and Goeta RIEN points with one-day lag (2 half-days in case of max12), suggesting that storm surge values were (slightly) higher correlated with wave height values of the previous day. Results in Tables and Figures refer to zero-lag values.

### 4.2 3.2 Validation of hindcasts in "compound" mode

First, the (Pearson) correlation between the two source variables (surge & wave) in observations observation mode is estimated while the same type of correlation is calculated in hindcast mode (see details in Table 1) for inter-comparison. Daily maximum (max24) values of storm surge observations collected at HvH station for the common time period (obs_com) are plotted against corresponding significant wave height observations recorded at the LiG buoy as shown in Fig. 8 (upper panel).

Observations of surge and wave seem to be well correlated with a coefficient reaching a value of 0.70. In hindcast mode (hind_com), the exact correlation value (0.70) was found between surge and wave hindcasts during the selected common interval. Surge and wave values in hind_com are plotted in Fig. 8 (lower panel). In both obs_com and hind_com modes the maximum values of correlations were achieved in zero lag mode (i.e., during the same 24-hour interval).

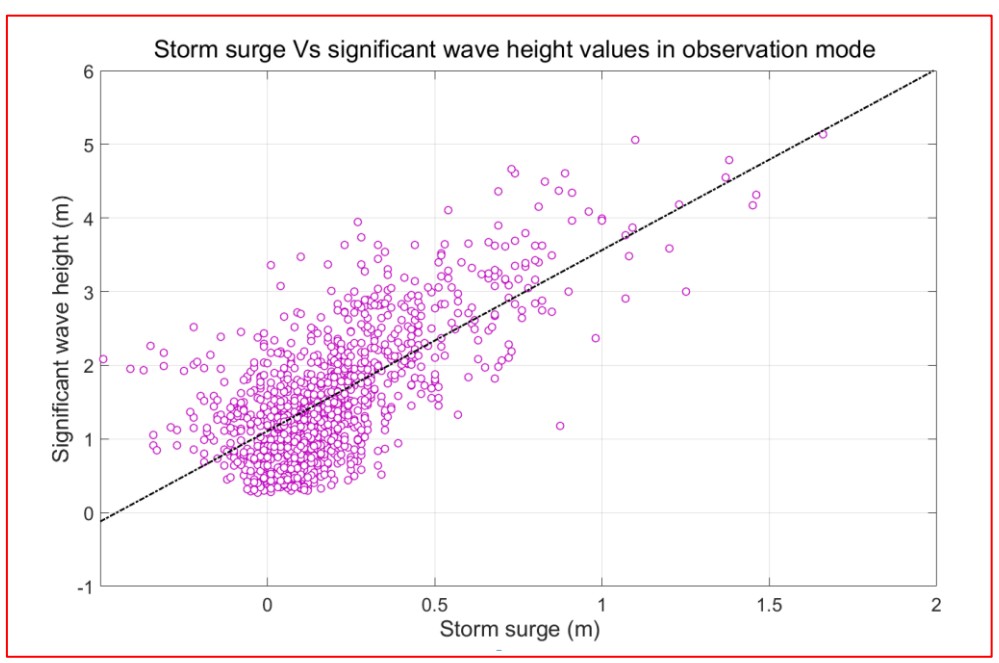

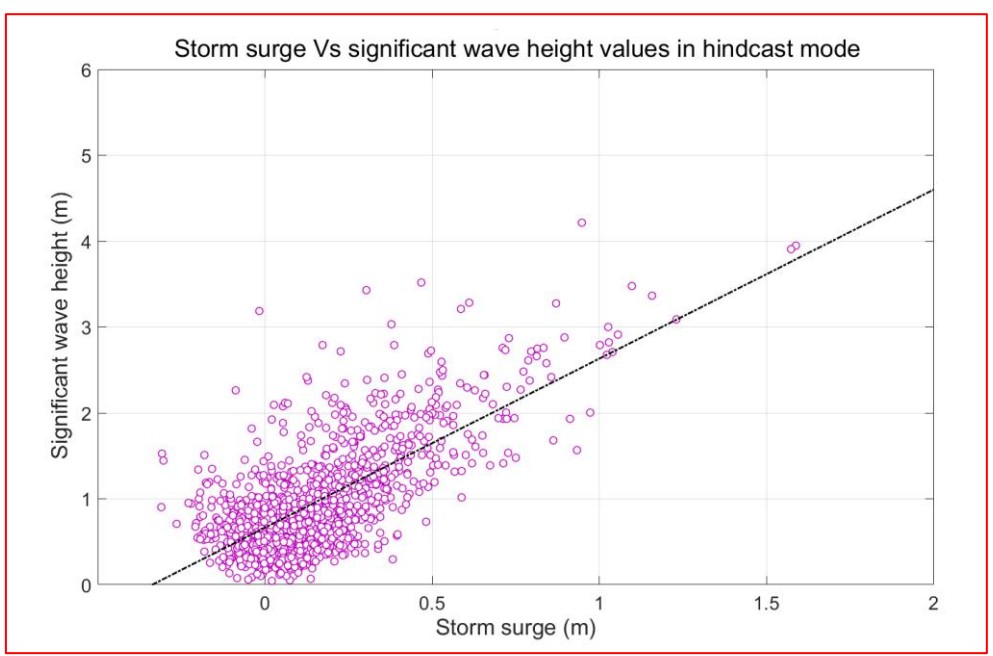

**Fig. 8.** Scatterplots of storm surge against significant wave height in obs_com (upper panel) and hind_com (lower panel) for the common time interval of 1,114 days.

**Table 1.** Details and abbreviations of data sets used in the study.

| | |
|---|---|
| **obs_com** | Observations during the common period (1,114 days) |
| **hind_com** | Hindcasts during the common period (1,114 days) |
| **hind_tot** | Hindcasts during the total period (12,753 days) |

Similar results were found in max12 hind_com case with a (slightly lower) correlation value reaching 0.69 in zero-lag mode. A slightly higher correlation value (0.73) was found in zero-lag mode when the total 12,753 daily max24 data pairs of hindcasts (hind_tot) were used. This deviation should not be considered significant since different number of data pairs was used. Similar (slightly lower) values of correlation were found for the max12 hind_tot data pairs (0.71). Overall, it seems that hindcasts in this case were able of resolving and estimating both the correct type and strength of correlation between source variables.

As in the case of correlations, the capability of hindcasts to resolve correctly the statistical dependence between surge and wave focusing on the upper (extreme) percentiles is investigated by inter-comparing dependencies estimated in obs_com and in hind_com (1,114 days). Fig. 9 shows the full range of $\chi$ values for all different types of data pairs considered in this study. The Peak-Over-threshold (POT) methodology (see details in the Statistical Supplement) was applied for a minimum three-day separation of extremes and an optimal selection of threshold was made not allowing more than ~2.3 events per year to exceed it, resulting in quite stable $\chi$ graphs over a wide range of percentile values as shown in Fig. 9 (for obs_com, hind_com and hind_tot), whereas due to the sparse of data pairs, values of dependence in the area of lower and higher quantiles appear to be quite unstable (i.e., with abrupt fluctuations).

**Table 2.** Dependencies between surge and wave values for observations (obs_com) and hindcasts (hind_com) in common interval and hindcasts over the total period (hind_tot). POT thresholds are shown in parentheses.

| obs_com | hind_com | hind_tot |
|---|---|---|
| 0.5850 (98.2%) | 0.5840 (98.3%) | 0.5629 (98.0%) |

Similar (in harmony among them) strong values of $\chi$ between surge and wave were found for all three configurations (obs_com, hind_com and hind_tot) in zero-lag mode as shown in Table 2. Similar results (for all three configurations) were also found when the same data pairs were considered for running chiplot of R (as shown in Fig. 1 of the Statistical Supplement). Slightly lower (but still in harmony among them) values were found for max12 case (in zero-lag mode).

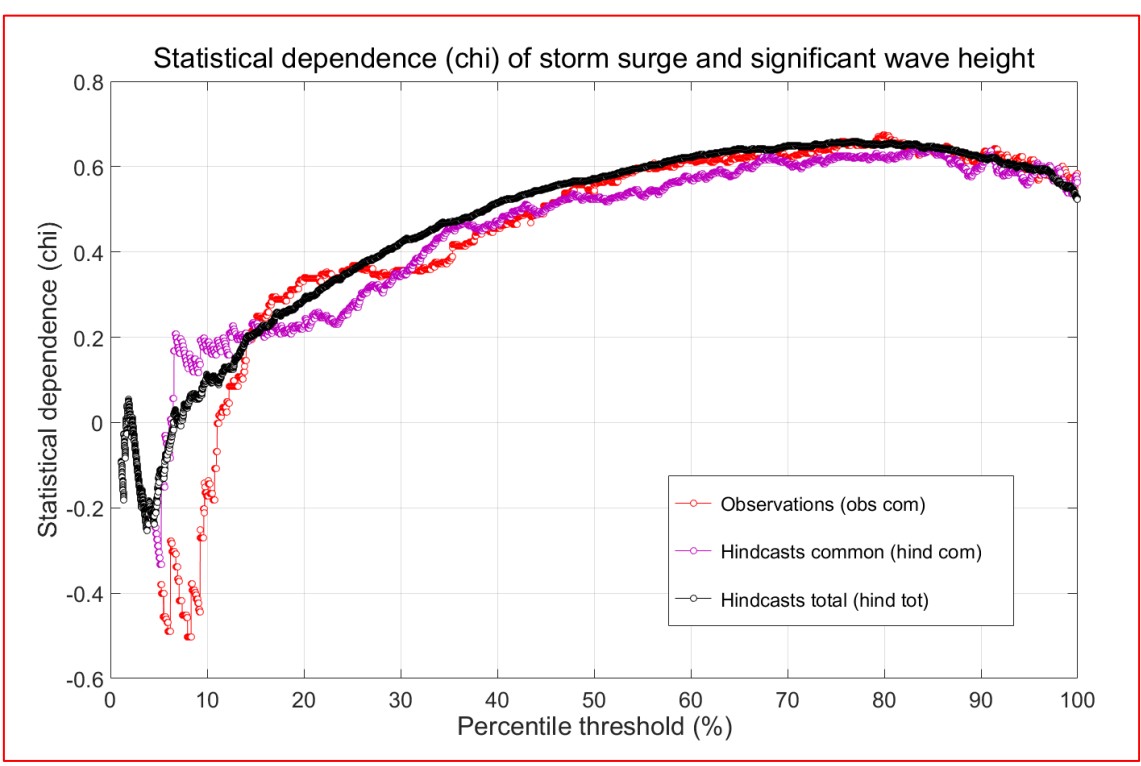

**Fig. 9.** Statistical dependence ($\chi$) of storm surge (HvH) and significant wave height (LiG) max24 values in common obs_com & hind_com (1,114 days) and total hind_tot (12,753 days) mode.

The importance and implications of such high values of dependence can be demonstrated with an example as the one presented in Sect. 7 of the Statistical Supplement by considering the total hindcast (hind_tot) series for surge (HvH) and wave (LiG).

It should be pointed out that the real (correct) statistical dependence is estimated by utilising the formula of Eq. 4 in the Statistical Supplement over a long set of real data (observations) of storm surge coming from a tide gauge and real data of wave height coming from a close by wave buoy. The tide gauge and wave buoy have to be relatively close for obvious reasons. Usually the tide gauge is in the vicinity of the port while the wave buoy is suited some kilometres offshore in front of the port.

Besides observations (that are limited in time length) hindcasts can be used as in our case. It should be also evident by now that even if hindcasts might be missing the exact magnitude of the extremes mainly due to the limited (model) resolution the most important issue here is their ability to resolve and estimate the correct value of both correlation and dependence as it ~~is~~ would have been estimated over real data (observations).

In the case of the RIEN of the Rhine River, the high level of agreement between the dependence estimated utilising (surge and wave) observations and the one utilising (surge and wave) hindcasts, points to the direction that hindcasts are capable of resolving both the correct type and strength of dependence between the source variables.

Overall, considering the complexity of all physical drivers behind such dependencies that are focused intentionally on the upper percentiles where extremes reside, hindcasts of storm and wave seem to perform quite well in their ability to simulate observations and to resolve correctly the type and strength of both correlation and statistical dependence.

### 4.3 3.3 Correlations and dependencies for southern coastal areas

A necessary split of results had to be made for a better and easier visualisation due to the relatively large amount of RIEN points to fit in one single Table. This split also revealed the distinct differences between southern and northern coastal European areas. Details of both correlations and dependencies found over southern RIEN points are presented analytically in Table 6 (based on Matlab routines) and Table 8 (based mainly on R routines) of the Technical Supplement. For the analysis of results, the ensemble mean value of $\chi$ (by averaging mat_chi, chiplot and taildep values) is taken as a reference value. The different

categories of correlation and dependence used later in the text refers to the categorisation adapted by Defra TR1 Report (2005), shown also (as an enclosed table) in Fig. 10.

In Table 6 (Technical Supplement), correlation (corr) and dependence (chi) values for both max12 and max24 intervals are presented together with critical threshold (thrs), significance (sig) and 95% confidence level (lower & upper) max24 values.

Referring to correlation values, a large amount of variability is evident in both max12 and max24 modes. In max12 mode, low ($0.05 \leq \chi < 0.12$), or even negative correlations were found over most coastal areas with the exception of Adriatic Sea RIENs and the RIEN of Aven River (belonging to a higher category), whereas moderate ($0.12 \leq \chi < 0.38$) values of dependence (max12) were estimated for most of those RIEN points. Such differences do not come as a surprise since dependence is focusing selectively on the upper (extreme) percentiles and not on the full range of data pairs, meaning that surge and wave

may have a considerable statistical dependence capable of modulating joint return period even if correlation in some cases is remarkably low (Drouet Mari and Kotz, 2004). Higher correlations were found in max24 mode compared to max12, although low (close to zero) and even negative values were estimated locally over the Balearic Sea, Alboran Sea, North & West Iberian and Black Sea.

Referring to dependence, with the exception of Foix RIEN point (belonging to the low category), the rest of the "comb" dependencies (last column of Table 8 of the Technical Supplement) fall into the moderate category ($0.12 \leq \chi < 0.38$). Besides dependence (chi), chibar values were estimated. Significance values, lower and upper confidence interval values of $\chi$ were

calculated also. In Table 8 (Technical Supplement), a set of R values is shown based on chiplot (extRemes module) and taildep (evd module) routines. Relatively small differences were found in estimations of dependence based on Matlab and R routines. Such differences may be attributed to the methodology for selecting critical percentile thresholds and how to identify and confine POT extremes in every case but nevertheless, in almost all cases both Matlab and R routine estimations were found to

belong in the same category. In addition, except for Foix RIEN both taildep & chiplot estimations of dependence fall well inside the confidence (95%) intervals estimated by mat_chi routines.

Extensive lag tests were made for both correlation and dependence revealing that the maximum strength of almost any compound surge-wave event tends to take place during the same max24 or max12 period (zero-lag mode). Exceptions were

found for Rhone, Ebro and Danube RIENs with one-day (2 half-day interval in max12 case) lag interval revealing that surge values were (slightly) higher correlated / dependent with wave values of the previous day. Further, the agreement of dependence values (among Matlab and R routines) became more pronounced, as the signal (value of dependence) got stronger. The ensemble mean (comb) value of $\chi$ is used hereafter for defining the category of dependence (max24) in all relevant text and maps.

## 15  4.4 3.4 Correlations and dependencies for northern coastal areas

In Table 7 (Technical Supplement), correlation (corr) and dependence (chi) values for both max12 and max24 intervals are presented together with chibar, critical threshold (thrs), significance (sig) and values of 95% confidence levels (max24). Distinctly higher values were found from than those over southern areas for both max12 and max24 cases. All values were achieved in zero lag mode (with the exception of Thames and Goeta RIENs reaching their highest values with one-day lag.

Apart from Thames RIEN (having negative correlation), all max12 correlations fall in the moderate category and above (corr $\geq 0.12$) with the top maximum value (0.59) of Bethune (RIEN) belonging to the strong category ($0.54 \leq$ corr $< 0.70$). Even higher correlation values were found in max24 mode with almost all values falling in the "well" category and above (corr $\geq$ 0.38). Correlations belonging to the "strong" category were estimated for a considerable number of RIENs over Irish Sea,

English Channel, North Sea and Baltic Sea.

Contrary to findings over southern areas, smaller differences between correlation and dependence values were found over the northern areas for both max12 and max24 cases. Significantly high values of dependence belonging to the "well" category and above ($\chi \geq 0.38$) between surge and wave were found over the Irish Sea, English Channel, North Sea, Norwegian Sea and

Baltic Sea in zero-lag mode (except for Goeta RIEN with one day lag time). Besides Bethune RIEN in the English Channel, having a strong dependence (0.65) in max24 mode, strong dependencies were also found for Rhine (0.54) and Weser (0.55) RIENs. Such findings suggest that over the south coasts of the North Sea, when a surge extreme event is anticipated, probabilities are quite high for an extreme wave event to take place at the same time (as a compound event).

As in Table 8 (Technical Supplement), a set of dependence values for northern coastal areas based on R routines is shown in Table 9 (Technical Supplement). Once more, small differences were detected in estimations of statistical dependence between Matlab and R routines but in almost all cases both Matlab and R routine estimations were found to fall in the same category.

5 In addition, both taildep & chiplot estimations of dependence fall well inside the confidence (95%) intervals estimated by mat_chi routines. As in Sect. 3.3, an ensemble mean value of chi contained in the last column of Table 9 (Technical Supplement) is considered as a reference value of dependence (max24).

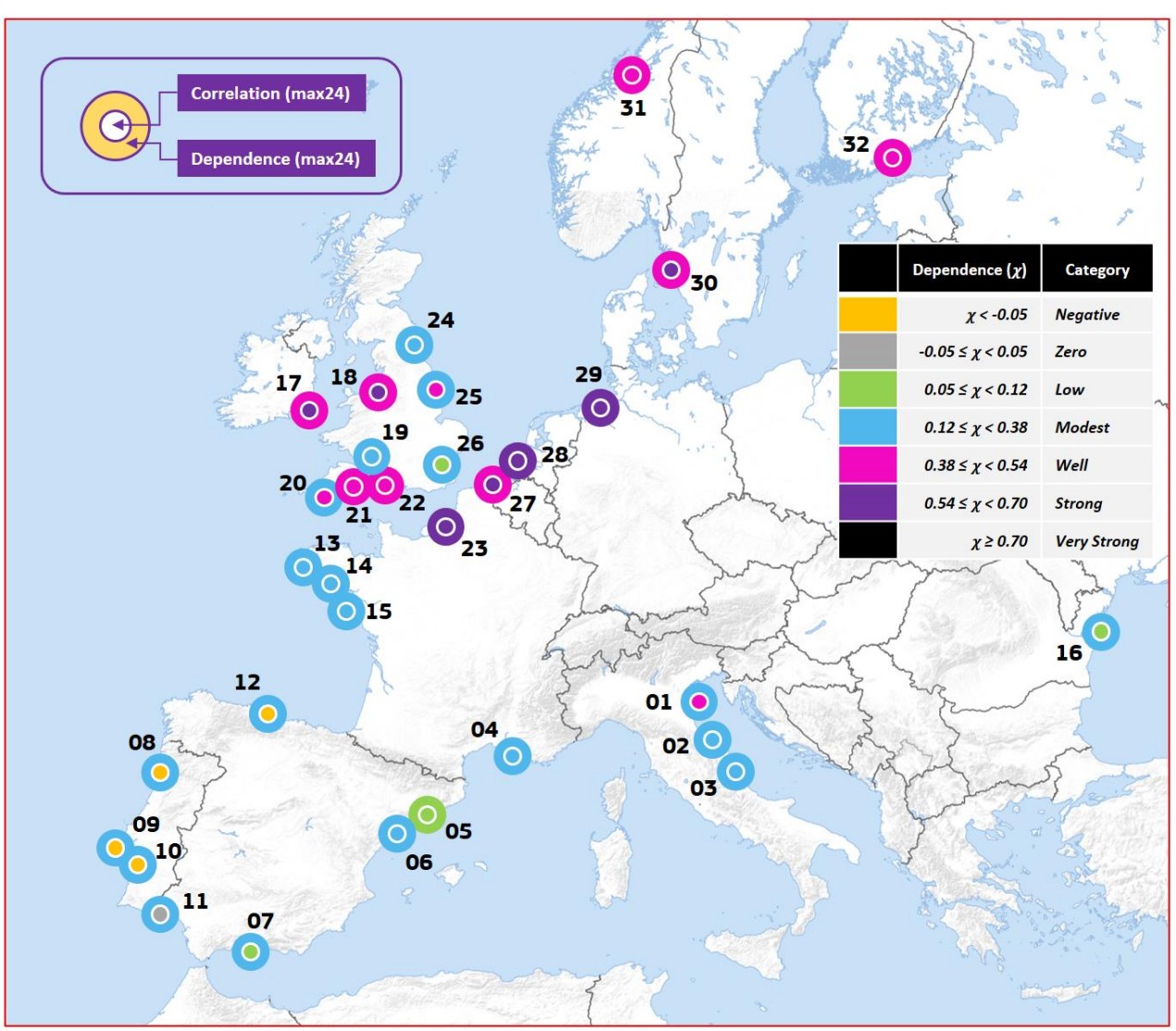

**Fig. 10.** Correlation (corr) and dependence (chi) values valid for max24 interval.

Results referring to the ensemble (comb) value of χ and correlation for max24 cases over all RIEN points are shown in Fig. 10. The categorisation applied in this study (shown graphically as an enclosed table in Fig. 10) is similar to the one introduced by Defra TR1 Report (2005).

Lastly, a full set of lag tests was made for both correlation and dependence. It was found that the maximum strength of almost any compound (surge and wave) event tends to take place during the same 24-hour (max24) time or during the same 12-hour (max12) period corresponding to zero-lag mode. Exceptions were found for Thames (UK), and Goeta Aelv (SE) RIEN points with one-day lag (2 half-days in the case of max12).

#### 4.5 3.5 Wind rose diagrams assessing the low-level flow characteristics during critical compound events

The "Prevailing Wind" is the most common wind direction over an area, i.e., the direction of wind with the highest frequency (AMS, 2017), whereas the "Dominant Wind" is the direction of the strongest wind that might blow from a different direction than the prevailing wind, i.e., from a less common direction (Thomas, 2000). The periods most frequently used for the estimation of prevailing and dominant winds are the observational day, month, season, and year. Methods for determination vary from a simple count of periodic observations to the computation of a wind rose.

Extreme compound surge and wave events are unavoidably linked to severe weather conditions. These conditions include very strong winds and low atmospheric pressure that is caused mainly by intense storms. Focusing on the low-level circulation, a set of wind rose diagrams was compiled for all RIEN points utilising ERAI reanalysis winds spanning over the total period of 12,753 days.

ERAI winds are referring to the four main synoptic hours (00 – 06 – 12 & 18 UTC) of reanalysis. From such 4-term (daily) sets the maximum speed was estimated and kept together with its corresponding direction to be used in the wind rose diagrams. Wind roses are an information packed plot providing frequencies of wind direction and speed. A wind rose diagram can quickly indicate both the prevailing wind referring to the principal or most common wind direction (having the highest percentage of

occurrence) and the dominant wind, indicating the direction of the highest wind speed. Examples of wind roses are given in Fig. 11 referring to daily maximum winds for the River Rhone RIEN during 12,753 days that may be taken as "clima" conditions (upper panel) and during the Top-80 (~2.3 yearly events during 35 years) compound events (lower panel) that may be considered as extreme compound mode conditions.

From Fig. 11 (upper panel) it is obvious that the clima prevailing (highest frequency) wind is north-northwest (NNW), a local type of wind named "Mistral" (www.cs.mcgill.ca/~rwest/wikispeedia/wpcd/wp/w/Wind.htm). The dominant wind (highest intensity) also is of a similar type (Mistral) blowing from a northwest (NW) direction. Mistral is a strong northerly wind blowing over the Gulf of Lion (GoL) and Rhone Valley. The air is usually dry, bringing bright and clear weather with freezing

temperatures to the south of France. The Mistral often reaches gale force especially in winter and is capable of raising heavy sea conditions in a short space of time.

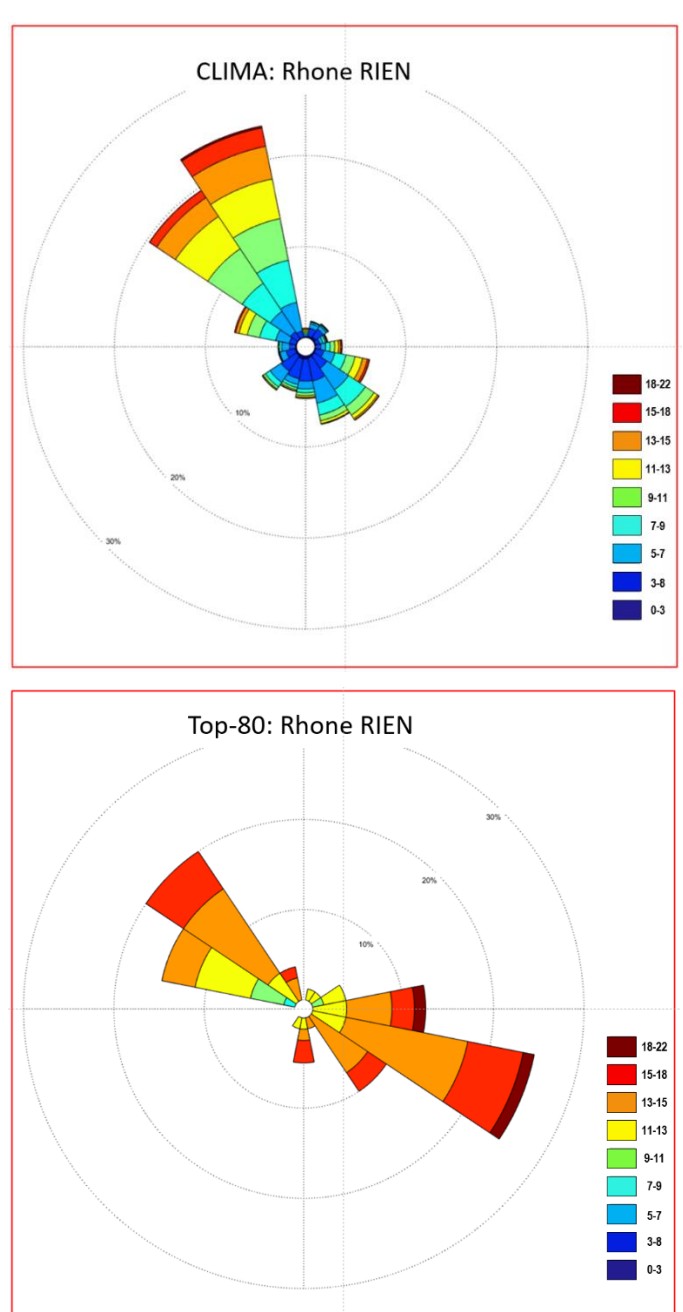

**Fig. 11.** Statistical "clima" average (upper panel) and Top-80 extreme compound (lower panel) daily maximum wind roses for Rhone River RIEN.

The same type of diagram was produced for the Top-80 compound events (lower panel of Fig. 11) revealing a quite different story. The prevailing wind does not belong to the Mistral "family" since it clearly comprises southeast components of another local wind named "Marin" (www.cs.mcgill.ca/~rwest/wikispeedia/wpcd/wp/w/Wind.htm). Marin is a strong wind in the area of GoL blowing from south-easterly directions, and is next in frequency and importance to the Mistral wind. It is generally warm, moist and cloudy, with rain and heavy weather, and is associated with depressions (storms) that enter the GoL area from the west or south-west after traversing southern France and northern Spain.

The implication of such findings is that although the prevailing and dominant wind in clima mode is of the Mistral type, most of the Top-80 extremes take place under Marin conditions in a relatively stronger wind environment (compared to Mistral conditions). On the other hand, Mistral conditions are also found to be responsible for a considerable percentage of Top-80 events accompanied by winds of lesser intensity (compared to the Marin ones). Similar detailed wind (clima & Top-80) roses were produced for the rest of the RIEN points. Distinct differences between southern and northern coastal areas are once more pronounced revealing relatively stronger intensity flow characteristics over the northern areas.

Details of clima and Top-80 flow characteristics are contained in Table 3. A possible exploitation of such information referring to both prevailing and dominant low-level flow characteristics should be considered significant and kept in mind when such extreme events possibly driven by intense storm outbreaks are anticipated over the area of interest (in forecast mode).

Not all prevailing and dominant directions contained in Table 3 fall in the perpendicular onshore category. Especially for the RIEN points of the south North Sea, wind directions appear to be more SWS instead of more northerly directions and this is because combined events had to be de-clustered. This means that a compound event lasting more than one day had to be counted as one (1) event even if this event could have lasted for a few days. After this necessary de-clustering all cases of compound events, are referring to the first day of the event (the first day that both storm surge and wave height found to be above a predefined critical threshold).

With such an approach, a compound event is considered only once and no other (another) event is taken into account for the next three days (even if the same event continues to exist longer than a day). Both prevailing and dominant directions are referring to the maximum daily intensity and if we consider the most common case of an approaching barometric low (storm) the wind in the beginning is more WSW whereas with the passage of the storm tends to veer to a more north-western (northern) direction becoming more perpendicular to the coast.

**Table 3.** Prevailing and dominant winds in clima and Top-80 extreme compound mode.

| | RIEN | clima | | Top-80 | | | RIEN | clima | | Top-80 | |
|---|---|---|---|---|---|---|---|---|---|---|---|
| | | prev | domi | prev | domi | | | prev | domi | prev | domi |
| 1 | Po | NE | ENE | ESE | NE | 17 | Owena | SSW | WSW | SSW | WSW |
| 2 | Metauro | NE | NE | NE | NE | 18 | Mersey | WSW | W | SW | NNW |
| 3 | Vibrata | NNE | SSE | SSE | ENE | 19 | Severn | WSW | SW | SSW | SSE |
| 4 | Rhone | NNW | NW | ESE | ESE | 20 | Tamar | WSW | WSW | SSW | WSW |
| 5 | Foix | S | NNW | WNW | NE | 21 | Exe | SW | WSW | SSW | SW |
| 6 | Ebro | NW | NW | SSW | NW | 22 | Avon | SW | SW | SSW | SW |
| 7 | Velez | ESE | E | E | E | 23 | Bethune | WSW | SW | SW | SW |
| 8 | Douro | NNW | NNW | SW | SW | 24 | Tyne | WSW | SSE | SW | WSW |
| 9 | Tagus | NNW | NW | SW | NW | 25 | Humber | SW | W | NNW | W |
| 10 | Sado | NNW | NW | SW | S | 26 | Thames | SW | W | N | SSW |
| 11 | Guadiana | NNW | W | SSW | SW | 27 | Schelde | WSW | WSW | W | WSW |
| 12 | Sella | NE | WSW | SW | WSW | 28 | Rhine | WSW | WSW | W | WSW |
| 13 | Moros | SW | SSW | WSW | W | 29 | Weser | WSW | WSW | WSW | WSW |
| 14 | Aven | SW | WSW | SSW | WSW | 30 | Goeta | WSW | W | SW | NNW |
| 15 | Blavet | SW | WSW | SSW | WSW | 31 | Orkla | SSE | WSW | WSW | WSW |
| 16 | Danube | NNW | NNE | ENE | NNE | 32 | Vantaa | SW | SSE | ENE | N |

Besides wind roses, the critical time period of the Top-80 events was investigated. For instance, in the case of the Rhone River, most Marin (east-southeast flow) and Mistral (north-west flow) Top-80 extreme compound events took place during the cold period of the year. Such a critical period was confined from October to March containing 91% of all Top-80 compound events.

Similarly, the critical period of Top-80 events was calculated for the rest of the RIEN points based on monthly frequencies of occurrence (Table 4). This critical interval comprised mostly cold months. There were even cases such as for the RIEN of Rhine (NL) and Schelde (BE) where all (100%) Top-80 compound events took place during the cold period (September to April). During these critical intervals (Table 4) there appears to exist a clear tendency for the northern extreme compound events to take place mostly with south-western components of stronger wind intensity (compared to southern events). This

tendency for both prevailing and dominant winds to be clustered around the south-western quadrant is more pronounced over the Irish Sea, English Channel, North Sea and Norwegian Sea.

**Table 4.** Critical period and percentage of occurrence for Top-80 compound events.

| | RIEN | Top-80 | | | RIEN | Top-80 | |
|---|---|---|---|---|---|---|---|
| | | per | % | | | per | % |
| 1 | Po | Oct – Mar | 91 | 17 | Owena | Oct – Mar | 93 |
| 2 | Metauro | Oct – Mar | 88 | 18 | Mersey | Oct – Mar | 96 |
| 3 | Vibrata | Oct – Mar | 91 | 19 | Severn | Sep – Apr | 91 |
| 4 | Rhone | Oct – Mar | 91 | 20 | Tamar | Sep – Apr | 94 |
| 5 | Foix | Sep – Apr | 94 | 21 | Exe | Sep – Mar | 91 |
| 6 | Ebro | Oct – Apr | 88 | 22 | Avon | Oct – Mar | 93 |
| 7 | Velez | Oct – May | 98 | 23 | Bethune | Oct – Mar | 93 |
| 8 | Douro | Oct – Apr | 88 | 24 | Tyne | Oct – Mar | 96 |
| 9 | Tagus | Oct – Apr | 94 | 25 | Humber | Oct – Apr | 98 |
| 10 | Sado | Oct – Apr | 97 | 26 | Thames | Oct – Apr | 91 |
| 11 | Guadiana | Oct – Apr | 93 | 27 | Schelde | Sep – Apr | 100 |
| 12 | Sella | Sep – Apr | 93 | 28 | Rhine | Sep – Apr | 100 |
| 13 | Moros | Sep – Apr | 94 | 29 | Weser | Oct – Apr | 97 |
| 14 | Aven | Sep – Apr | 91 | 30 | Goeta | Sep – Mar | 98 |
| 15 | Blavet | Sep – Apr | 93 | 31 | Orkla | Sep – Mar | 95 |
| 16 | Danube | Nov – Apr | 91 | 32 | Vantaa | Sep – Jun | 98 |

5    The validity of such findings is briefly investigated. For the Irish Sea, extreme surge conditions especially in its eastern side are generated by south-westerly to westerly winds as documented by Brown et al. (2010). For the English Channel, this south-western signature is compatible with the path of (extra-tropical) storms that tend to generate large surges (Henderson and Webber, 1977). For the North Sea such south-western preference seems to partly contradict the fact that the largest wave events occur in the central North Sea when a low-pressure system is situated over southern Scandinavia (such as the one shown in

10   Fig. 6) giving rise to a long northerly fetch associated with strong northerly winds. An obvious explanation could be that southerly-wind events can also create large wave heights despite their limited fetch, since southerly-wind events are associated with the existence of zonal jets (embedded in extratropical cyclones) that intensify rapidly in the left exit region of the jet stream as indicated by Bell et al. (2017). Besides this, depths in the southern North Sea are only about 40m on average adding

to the fact that wind stress is particularly effective in piling up water against the coast in the shallow water as the effect is inversely proportional to water depth (Wang et al., 2008).

It should be kept in mind that besides the prevailing and dominant wind directions responsible for most compound extremes there still exist additional critical directions linked to extremes. For instance in the area of German Bight (southern North Sea), northwest wind components (visible in the wind rose for Weser RIEN) have been identified as having a significant link to both surge and wave extremes (Staneva et al., 2016).

Lastly, for the Norwegian Sea, observations seem to fully support our findings as documented in an earlier work of Gjevik and Røed (1976) showing that large storm surges are caused by strong south-westerly winds acting along a large section of the Norwegian coasts.

Overall, the low-level flow characteristics (prevailing and dominant winds) appear to be first in harmony with the transient nature of (extra-tropical) storms and their footprints (storm tracks). This seems to be consistent with similar findings (even if they apply for different pair of variables) in Defra TR1 & TR3 Reports and in Svensson and Jones (2002, 2004a, 2004b, 2005) documenting that (storm) surge and (river) flow dependence appears to be largely influenced by the storm track of the depressions although it should be kept in mind that a thorough understanding of all factors leading to such compound events is above the scope of this study.

## ~~5~~ 4 Discussion ~~and conclusions~~

The possibility of utilizing statistical dependence methods in coastal flood hazard calculations is investigated, since flood risk is rarely a function of just one source variable but usually two or more. Source variables in most cases are not independent as they may be driven by the same weather event, so their dependence ($\chi$), which is capable of modulating their joint return period, has to be estimated before the calculation of their joint probability. The source variable-pairs presented here, are storm surge and wave height, and their correlation and dependence were assessed over 32 river ending (RIEN) points along European coasts. It should be noted that correlation and dependence may differ substantially from one another. This is because correlation is estimated over the full range of percentiles whereas dependence is focused on the upper (extreme) percentiles.

In the absence of widespread coincident long-term measurements of surge and wave, a set of ~35-year (12,753 days) hindcasts was compiled. Storm surge hindcasts were performed by utilising the hydrodynamic model Delft3D-Flow while wave hindcasts were generated with ECWAM wave (stand-alone) model. Although in some cases extreme surge and wave hindcast levels were underestimated, the overall performance of both surge and wave hindcasts is considered satisfactory. Further, a joint validation in "compound mode" was made over the area of Hook van Holland (HvH) taking into account real measurements of both tides and waves. Overall, hindcasts for the common period of observations (1,114 days) were found capable of resolving and estimating both the correct type and strength of correlation and dependence between source variables.

Since such "compound" validation is impossible to be repeated for all RIEN points, some caution with the exact levels of correlation and dependence should be bear in mind for the rest of the RIEN points.

Results are presented by means of analytical tables and maps for each RIEN point and can be used to calculate the joint return period by inserting the value of dependence ($\chi$) in a simple formula (Eq. 12 of the Statistical Supplement) containing the individual return periods of source variables as documented in Hawkes (2004), Meadowcroft et al. (2004), White (2007), Australian Rainfall & Runoff Project 18 (2009) and Petroliagkis et al. (2016). Some limitations of Eq. 12 (Statistical Supplement) could be overcome if a more complete formula is used such as Eq. 2.15 for instance taken from White's thesis (2017) but this is beyond the scope of the current study.

Further, a necessary split of results revealed distinct differences between southern and northern coastal European areas since significantly higher values of correlation and dependence were found over northern sea areas. Overall, significant correlations and dependencies between surge and wave ranging from "well" and above ($\geq 0.38$) categories were found over the Irish Sea, English Channel, south coasts of the North Sea, Norwegian Sea and Baltic Sea in a zero-lag mode. Over these areas,

dependencies reaching locally up to 0.65 (Bethune RIEN) stress the fact that when the first variable (surge) has an extreme value there exists a high probability that the other one (wave) will also produce an extreme level. For the rest of the RIEN points mostly positive moderate ($0.12 \leq \chi < 0.38$) dependence values were estimated although a considerable number of them had correlations that were almost zero or even negative. This does not come as a surprise since even in cases of very low

correlation there may exist a considerable amount of tail dependence.

Based on these results, it seems that compound events over northern sea areas are mostly driven (forced) by a common extreme wind event resulting in a high value of dependence between surge and wave, whereas a large contribution of atmospheric pressure affecting only storm surge might be one among other reasons for low dependence values over southern sea areas.

An effort for inter-comparing our results with previous studies was made although there were very few relevant journal papers (to our knowledge) focusing on correlations and dependencies over such a wide range of coastal areas. A relevant study (thesis) by Kergadallan (2016) for the coasts of France has documented that the surge wave dependence is medium along the Mediterranean coasts whereas dependence values are more important along the English Channel and the Atlantic coasts, which

seems consistent with our findings.

Other relevant references pointed to a series of U.K. Defra / Environmental Agency Institute Reports (2003 [Defra TR0], 2005 [Defra TR1], 2005 [Defra TR2] & 2005 [Defra TR3]), hereafter referenced as TRx Reports. This set of Reports (TRx) though, refer to a different measure of dependence constituting a "special" correlation coefficient $\rho$, above a chosen threshold (90%).

Over U.K. coasts, such values of $\rho$ were found to be positive as our set of $\chi$ values but considerable higher. Such differences could be attributed partly to the fact that $\chi$ values were estimated by considering a quite different (POT) threshold from the one (90%) used in $\rho$ estimations. It could be also attributed to the different nature (methodology of estimation) between $\chi$ and $\rho$, since it is clearly mentioned in TR1 Report that different statistical models are underlying $\chi$ and $\rho$ values that could cause considerable distortion when converting from one parameter to the other.

Above all, it appears that such values of $\rho$ (coming from TRx Reports) should not be considered as reliable statistical dependence ($\chi$) values as they point to overestimated levels. In support of this, we refer to the methodology of estimating statistical dependence $\chi(u)$ by Coles (2001) utilising a set of reference data for surge and wave over the Port of Newlyn (Cornwell, U.K.). Results taken from Figure 8.11 (Coles, 2001) suggest a dependence value ~0.35 as $\chi(u)$ clearly tends to this

value for the upper percentiles. This is very close to our estimation for the RIEN of Tamar River (0.34) and significantly different from the value found in Table 4.4 of TR1 Report suggesting a value of $\rho$ higher than 0.60.

A further investigation into the low-level flow characteristics of extreme compound events was made for a possible collection of forecasting "rule-of-thumb" guidelines. First, a set of 10-metre wind roses was compiled utilising ERAI wind terms over the total period of 12,753 days. These winds are referring to the four main synoptic hours (00 – 06 – 12 & 18 UTC) based on which the daily maximum speed and its corresponding direction were defined and used for producing a set of "clima" wind roses for all RIENs. Based on such clima wind roses the estimation of the prevailing (highest frequency) and dominant (highest intensity) winds was possible. In addition, the 80 most extreme (Top-80) compound events were defined by applying POT (Peaks-Over-Threshold) methodology and allowing a maximum number (~2.3) of compound events on annual basis. A set of wind roses in such extreme mode was assembled revealing distinct differences between clima and Top-80 events in many cases (Table 3). For instance, in the case of the River Rhone (RIEN), the clima prevailing average conditions were of Mistral (north-western) type whereas the top extremes (Top-80) were mostly of Marin (south-eastern) type conditions.

Detailed wind roses (Top-80 mode) were produced for the rest of RIEN points using a common wind speed scale. It seems that there is a clear tendency for the northern extreme compound events to take place mostly with south-western components of stronger wind intensity (compared to the southern ones) with emphasis during the cold months. This appears to be in harmony with the transient nature of winter storms and their storm tracks as already indicated in Svensson and Jones (2004a & 2004b) in a similar analysis for surge and discharge compound events around Britain.

It should be noted, not all prevailing and dominant directions contained in Table 3 fall in the perpendicular onshore category. Especially for the RIEN points of the south North Sea, wind directions appear to be more SWS instead of more northerly directions and this is because combined events had to be de-clustered. This means that a compound event lasting more than one day had to be counted as one (1) event even if this event could have lasted for a few days. After this necessary de-clustering all cases of compound events, are referring to the first day of the event. With such an approach, a compound event is considered only once and no other (another) event is taken into account for the next three days. Both prevailing and dominant directions are referring to the maximum daily intensity and if we consider the most common case of an approaching barometric low (storm) the wind in the beginning is more WSW whereas with the passage of the storm tends to veer to a more north-western (northern) direction becoming more perpendicular to the coast.

Besides the relevant link between transient storm systems and compound events, the morphological and topographical characteristics of RIEN areas appears to play a significant role in the genesis and evolution of such extremes. For instance, in addition to the local circulation systems such as the Mistral and Marin winds in the case of Rhone RIEN, a similar pattern was seen with the Bora (north-eastern) and Sirocco (south-eastern) winds providing the main dominant and prevailing (respectively) flows during the Top-80 compound events over Po RIEN (North Adriatic Sea).

The critical time period of Top-80 events was also estimated based on monthly frequency values of occurrence. This critical interval comprised mostly cold months (Table 4). There were even cases such as for Rhine RIEN (NL) and Schelde RIEN (BE) where all (100%) Top-80 compound events took place during the cold period (September to April).

**5 Conclusions**

In the absence of widespread coincident long-term measurements of surge and wave a set of ~35-year hindcasts was compiled to assess the correlation and statistical dependence over 32 river ending (RIEN) points along European coasts. A joint validation in "compound mode" was made over the area of Hook van Holland taking into account real measurements of both tides and waves. Hindcasts were found capable of resolving and estimating both the correct type and strength of correlation

and dependence between source variables.

Since such "compound" validation is impossible to be repeated for all RIEN points, some caution with the exact levels of correlation and dependence should be bear in mind for the rest of the RIEN points.

Results are presented by means of analytical tables and maps for each RIEN point and can be used to calculate the joint return period by inserting the value of dependence ($\chi$) in a simple formula containing the individual return periods of source variables.

A necessary split of results revealed distinct differences between southern and northern coastal European areas since significantly higher values of correlation and dependence were found over northern sea areas with compound events taking

place on the same max12 (during half a day) or max24 (daily) interval in a zero-lag mode. More specifically, strong values of positive correlations and dependencies were found over the Irish Sea, English Channel, south coasts of the North Sea, Norwegian Sea and Baltic Sea, with compound events taking place in a zero-lag mode. For the rest of RIEN points, mostly positive moderate dependence values were estimated even if a considerable number of them had correlations of almost zero or even negative value. These results seem to be in agreement with results from relevant studies over the coasts of France

documenting that the surge wave dependence is medium (moderate) along the Mediterranean coasts whereas dependence values are more important along the English Channel and the Atlantic coasts. Another similar study over Tamar River (U.K.) has also suggested values close to our estimations of moderate dependence.

Based on these results, it seems that compound events over northern sea areas are mostly driven (forced) by a common extreme

wind event resulting in a high value of dependence between surge and wave, whereas a large contribution of atmospheric pressure affecting only storm surge might be one among other reasons for low dependence values over southern sea areas.

A further investigation into the low-level flow characteristics of extreme compound events was made for a possible collection of forecasting "rule-of-thumb" guidelines. Detailed wind roses (in extreme-mode) were produced for all RIEN points using a common wind speed scale. It seems that there is a clear tendency for the northern extreme compound events to take place mostly with south-western components of stronger wind intensity (compared to the southern ones) with emphasis during the cold months. This appears to be in harmony with the transient nature of winter storms and their storm tracks as already indicated in similar analyses for surge and discharge compound events around Britain.

Besides the relevant link between transient storm systems and compound events, the morphological and topographical characteristics of RIEN areas appears to play a significant role in the genesis and evolution of such extremes.

The critical period of extreme-mode events was also estimated based on monthly frequency values of occurrence. This critical interval found to comprise mostly cold months, while there were even cases such as for Rhine RIEN (NL) and Schelde RIEN (BE) where all (100%) Top-80 compound events took place during the cold period (September to April).

This work has been a the first step of studying and investigating joint probabilities and return periods of compound events in a relatively low-resolution environment. Having this in mind, results referring to dependence estimations should be considered valid for coastal areas up to a certain distance (a few kilometres) away from the shoreline. Nevertheless, maps and tables can be used to get a valuable indication of the possibility for a combined (compound) hazard based on how the source variables are related (though statistical dependence) over various coastal areas of Europe.

A thorough estimation of the design conditions at the coastal zone would require including the inclusion of more primary and proxy variables in a higher resolution environment. For instance, in addition to the significant wave height, the maximum wave height or/and the period or/and the direction of waves should be also considered. Another important point here is the effect of seasonal circulation and water-mass distribution (currents & tides) besides the prevailing weather system and atmospheric circulation contained in relevant weather maps.

**Acknowledgements**

Dr Jean-Raymond Bidlot (ECMWF) is to be gratefully thanked for providing the set of wave hindcasts and in situ wave observation data besides valuable guidance and suggestions. Evangelos Voukouvalas (JRC) is to be thanked for providing us the set of storm surge hindcasts. A long list of JRC colleagues should be also thanked for their invaluable help and support during the Exploratory Research Project Coastal-Alert-Risk (CoastAlRisk) of the JRC (Joint Research Center). The CoastAlRisk Project (2015-2016) had been an initial effort of developing the first global integrated coastal flood risk

management system with emphasis on such compound events, by linking satellite monitoring, coupled wave, tide and surge forecasting, inundation modelling and impact analysis.

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

*Statistical Supplement of*

# Estimations of statistical dependence as joint return period modulator of compound events. Part I: storm surge and wave height.

Thomas I. Petroliagkis

*Correspondence to*: Thomas I. Petroliagkis (thomas.petroliagkis@ec.europa.eu)

***Contents***

## 1 Statistical dependence ($\chi$)

The main concept of the so-called dependence measure $\chi$ (chi) is related to two or more simultaneously observed variables of interest – such as in our case storm surge and wave height – known as observational pairs. If one variable exceeds a certain extreme (high-impact) threshold, then the value of $\chi$ represents the risk that the other variable will also exceed a high-impact threshold as explained in Hawkes (2004), Svensson and Jones (2004a & 2004b), Petroliagkis et al. (2016).

Following Coles et al. (2000), if all of the extreme observations of two variables exceed a given threshold at the same time, this indicates total dependence ($\chi = 1$). If the extreme observations of one variable exceed a given threshold but the second variable does not, this indicates total independence ($\chi = 0$). Similarly, if the extreme observations of one variable exceed a given threshold but the other variable produces lower observations than would normally be expected, this indicates negative dependence ($\chi = -1$). In practice, in tidal and estuarine environments, assessing the probability of flooding from the joint occurrence of both high storm surge and high wave values is not an easy process, as high surges and waves might be related to the same prevailing meteorological conditions (Beersma and Buishand, 2004), thus independence cannot and should not always be assumed. For instance, if we assume independence between input variables, this might underestimate considerably the likelihood of flooding (estimated by the product of their individual probability) resulting in higher risk for the coastal community. Similarly, assuming total dependence could be too conservative. Further, as variables reach their extreme values, special methodologies of estimating statistical dependence could be utilised as the one documented in Buishand (1984).

A brief description of this methodology based on Coles et al. (2000) is described below (Sect. 2) while the basic theory behind the utilisation of an optimal copula function refers to Nelsen (1998), Joe (1997), Currie (1999), Wahl et al. (2015).

## 2 Estimation of statistical dependence ($\chi$)

For bivariate random variables (X, Y) with identical marginal distributions, the dependence measure ($\chi$) can estimate the probability of one variable being extreme provided that the other one is extreme:

$$\chi = \lim_{z \to z^*} \Pr(Y > z \mid X > z) \qquad (1)$$

where z* is the upper limit of the observations of the common marginal distribution.

For obtaining identical marginal distributions, each set of observations is ranked separately and each rank is then divided by the total number of observations resulting in a data transformation with Uniform [0, 1] margins. At this point, it is convenient

to consider the bivariate cumulative function $F(x, y) = Prob( X \leq x, Y \leq y )$ that describes the dependence between X and Y completely. The effect of different marginal distributions can be diminished by assuming the copula function C in the domain [0, 1] x [0, 1] such as:

$$F(x, y) = C \{ F_x(x), F_y(y) \} \qquad (2)$$

where $F_x$ and $F_y$ can be any marginal distributions. Such utilisation of the copula function has the same effect as if observations were ranked separately and divided by the total number of observations. In addition, the copula C contains the complete information about the joint distribution of X and Y and it is invariant to marginal transformation. This means that C is invariant to marginal transformation and it can be described as the joint distribution function of X and Y. Further, X and Y are

transformed to new variables U and V with Uniform [0, 1] margins. It follows that the dependence measure $\chi(u)$ for a given threshold u can be given by:

$$\chi(u) = 2 - \frac{\ln \Pr(U \leq u, V \leq u)}{\ln P(U \leq u)} \quad \text{for } 0 \leq u \leq 1 \qquad (3)$$

Taken into account the upper limit of the observations (previously defined as $z^*$ in Eq. 1), the dependence measure $\chi(u)$ will

be given by:

$$\chi = \lim_{u \to 1} \chi(u) \qquad (4)$$

Details of deriving Eq. 3 can be found in Coles et al. (2000). Based on Eq. 3, a set of $\chi$ values can be evaluated at different quantile levels u. The selection of a particular level u corresponds to threshold levels (x*, y*) for the two different data series. For applying Eq. 3, the number of appropriate observation-pairs (X, Y) is counted for estimating the numerator and

denominator terms (Eq. 5 & Eq. 6):

$$P(U \leq u, V \leq u) = \frac{\text{Number of (X, Y) such that } X \leq x^* \text{ and } Y \leq y^*}{\text{Total number of (X, Y)}} \qquad (5)$$

and

$$\ln P(U \leq u) = \frac{1}{2} \ln[ \frac{\text{Number of } X \leq x^*}{\text{Total number of X}} \cdot \frac{\text{Number of } Y \leq y^*}{\text{Total number of Y}} ] \qquad (6)$$

In this study, a set of routines (mat_chi) based on Matlab software were coded following Eq. 3 to 6 for estimating $\chi$. Additional

modules and routines based on the integrated statistical package R were also used for estimating dependence terms and inter-

comparing various parameters. Emphasis was given on the routine "taildep" of the module "extRemes" (https://cran.r-project.org/web/packages/extRemes/extRemes.pdf) that is capable of estimating $\chi$ values when a critical percentile (extreme) threshold is considered. Another "powerful" routine capable of providing a variety of dependence graphs and plots (besides single estimated values of $\chi$) has been the routine "chiplot" of the module "evd" (Extreme Value Distributions) of R

(https://cran.r-project.org/web/packages/evd/evd.pdf). The routine chiplot is also capable of providing confidence intervals at any preselected level.

Besides estimating values of $\chi$, similar routines (mat_chibar) were coded in Matlab following Coles et al. (2000) for calculating the "sister" attribute of $\chi$, namely chibar ($\bar{\chi}$). Chibar (chi_bar) parameter refers to the statistical dependence of asymptotically

independent variables whereas chi ($\chi$) refers to the statistical dependence of asymptotically dependent ones. Details on the estimation of chibar are documented in Coles et al. (2000) whereas examples and how to utilise ($\bar{\chi}$) can be found in Coles (2001). The class of asymptotic dependence appears to be the case in Literature, having reached a consensus that there is strong, although not overwhelming, evidence for asymptotic dependence between wave height and surge (Wadsworth et al., 2017).

The concept of asymptotic dependence ($\chi$) is stated with adequate details in Coles et al. (2000). In brief, $\chi$ is on the scale [0, 1] with the set (0, 1] corresponding to asymptotic dependence whereas the measure chibar ($\bar{\chi}$) falls within the range [-1, 1] with the set [-1, 1) corresponding to asymptotic independence. That is why the complete pair of $\chi$ and $\bar{\chi}$ is required as a summary of extremal dependence:

- $\chi > 0$ & $\bar{\chi} = 1$ reveals asymptotic dependence, in which case the value of $\chi$ determines a measure of strength of dependence within the class

 - $\chi = 0$ & $\bar{\chi} < 1$ reveals asymptotic independence, in which case the value of $\bar{\chi}$ determines the strength of dependence within the class.

For estimating both $\chi$ and $\bar{\chi}$ parameters, the general POT (Peaks-Over-Threshold) methodology was followed. Such an approach (POT) is considered as giving a more accurate estimate of the probability distribution than using the annual maximum series (see details in Stedinger et al., 1993). Applying POT as described in detail in Defra TR1 Report (2005), the selection of an optimal threshold for the data pairs (~2.3 events per year) was adopted as suggested in Defra TR3 Report (2005). Care was taken to force two POT extreme compound events not occurring on consecutive days, but separated by at least three days from

each other. Emphasis was also given on the stability of $\chi$ (graph) curves identifying the area that dependence was clearly converging to a specific value (no abrupt fluctuations).

Relatively small differences among various estimates made by chiplot of evd (R), taildep of extRemes (R) and mat_chi (Matlab) were found. This most probably is due to the unavoidable dissimilarities between the criteria being imposed on data pairs when applying POT methodology (selection of different critical thresholds).

5 **3 Selection of critical thresholds**

For selecting a threshold u (referring to a critical percentile) as required in Eq. 3, it seems appropriate to transform the Uniform distribution to an annual maximum non-exceedance probability scale (Defra TR3 Report, 2005). Then the annual maximum non-exceedance probability ($\alpha$) is defined as:

$$\alpha = \text{Prob (Annual maximum} \leq x) \qquad (7)$$

where x is the magnitude of the source variable. Such non-exceedance probability relates to the return period, $T_\alpha$, as:

$$T_\alpha = 1 / (1 - \alpha) \qquad (8)$$

For a transformation from annual maximum to POT series (see details and scope in the previous Sect. 2), we define the "new" non-exceedance probability, the so-called p, referring to a rate of $\lambda$ events per year, relating to the annual maximum of Eq. 7, as:

$$\alpha = \exp\left(-\lambda\left(1 - p\right)\right) \qquad (9)$$

where 1-p is the "new" exceedance probability of the POT series. The term $(1 - p)$ can be estimated graphically (Hazen, 1914) leading to Equation 10:

$$\lambda\left(1 - p\right) = \left(N_e / N\right) * \left(i - 0.5\right) / N_e = \left(i - 0.5\right) / N \qquad (10)$$

where i, represents the rank of the independent POT events, $N_e$ is the number of POT events while N represents the number of years (see details in Defra TR3 Report, 2005). The independence criterion of two POT events to be separated by at least three days (six half-day intervals in the max12 case) was applied for all river ending points. Combining Eq. 9 and Eq. 10, an
30 estimation of $\alpha$ is possible as given by Eq. 11:

$$\alpha = \exp\left(-(i - 0.5) / N\right) \qquad (11)$$

Therefore, going after the magnitude of x in Eq. 7 is equivalent as trying to define the magnitude of the POT element with rank i in Eq. 11 for the same maximum non-exceedance annual probability, alpha (α). After the selection of an optimal threshold (u) based on alpha (α), the estimation of χ is straightforward (Eq. 3). The main idea here is to use χ in a relatively simple formula that also uses as input the individual return periods $T_X$ and $T_Y$ for estimating the joint return period ($T_{X,Y}$), like

the formula described by Eq. 12 following White (2007), Australian Rainfall & Runoff Project 18 (2009).

$$T_{XY} = \sqrt{T_X * T_Y / \chi^2} \qquad (12)$$

Studying Eq. 12 closely it becomes obvious that dependence is capable of substantially modulating the joint return period. For details and potential limitations of Eq. 12, see discussions in White (2007), Hawkes (2004), Meadowcroft et al. (2004),

Australian Rainfall & Runoff Project 18 (2009). In cases of totally dependent variables, Eq. 12 yields the common individual return period of source variables as an estimation of the joint return period. An example of how to utilise Eq. 12 is given in Sect. 4.2 of the main text for the river ending point of Rhine (NL). Further, some limitations of Eq. 12 could be overcome if a more complete formula is used such as Eq. 2.15 for instance taken from White's thesis (2007) but this is above the scope of the current study.

## 4 Significance

The values of dependence (χ) corresponding to the 5% significance level were estimated using a permutation method as described by Good (1994). As in Defra TR3 Report (2005), 199 permutations of the data were made for each surge-wave pair and a new value of χ was calculated each time. All 199 values of χ were subsequently ranked in descending order and the 5%

significance level was defined by selecting the 10th largest value representing the 95% point of the null distribution (the hypothetical distribution occurring if data-pairs were indeed independent). Care was taken to preserve the seasonality since permutation of data was performed by randomly reshuffling intact blocks of one year time period.

It should be kept in mind that the significance level of 5% represents the probability of rejecting the null hypothesis when it is

true. In simple words, it indicates a 5% risk of concluding that a difference exists capable of rejecting the null hypothesis (the population mean equals to the hypothesized mean) when there is no actual difference.

## 5 Confidence intervals

For the estimation of confidence intervals, a well-tested bootstrapping method was applied similar to the permutation method

already used for estimating significance (for details see Defra TR3 Report, 2005). This bootstrapping resulted in the generation

of many new data-sets (resamples). The original sample of observation-pairs was used as the main (reference) distribution from which the resamples were chosen randomly. A large number of data sets were generated for calculating $\chi$ for each of these new data sets. This provided a sample of what would occur for a range of situations. Seasonality was kept intact by sampling in blocks of one year, rather than using individual observation-pairs. The balanced resampling as documented by Fisher (1993) was applied ensuring that each year occurs equally often overall among the total number of bootstrap samples. In total, 199 bootstrap samples of the data were made for each station-pair and a new $\chi$ value was calculated each time. The 199 values were subsequently ranked in descending order and the 10 and 190 largest values were accepted as determining the 90% confidence interval.

To draw the distinction between significance (previous Sect. 4) and confidence levels it should be noted that a confidence interval is a range of values that is likely to contain an unknown population parameter (in our case the statistical dependence) whereas the significance represents the probability of rejecting the null hypothesis when it is true. It follows that if a random sample is drawn many times, a certain percentage of the confidence intervals will contain the population mean. That is the reason behind the usage of confidence intervals for bounding the mean or standard deviation.

**6 Selection of critical thresholds resulting in the consideration of top-80 events**

Extreme value analysis can be carried out using two types of data series (Bezak et al., 2014), annual maximums (MA) or flows above a certain threshold (POT) for Peak Over Threshold. The POT model used in this study can be composed of the Poisson, binomial and negative binomial distributions for modelling the annual number of events above threshold, and of exponential or generalized Pareto distributions for magnitudes of exceedances.

Since values of dependence ($\chi$) can be estimated for any lower or upper threshold, initial trials were performed studying the behaviour of $\chi$ over a wide range of thresholds. Findings were similar to those contained in Defra TR3 Report (2005), justifying the selection of an optimal threshold for "alpha" ($\alpha$) equal to 0.1 corresponding to an annual maximum being exceeded in 9 out of 10 years (see details in Sect. 3). This value (0.1) of alpha was considered for both mat_chi ($\chi$) and mat_chibar ($\bar{\chi}$) routines when utilising POT (Peaks-Over-Threshold) methodology resulting in an annual maximum of ~2.3 compound events.

Such annual threshold of ~2.3 events corresponds to the top 80 (Top-80) compound events taking place during any (POT separated) day of the total 12,753 days and it was dictated mainly by two factors: the threshold had to be low enough to allow a sufficient number of data points to exceed it for estimating dependence reliably, while being high enough for the data points to be regarded as extremes.

**7 Details and examples of the statistical packages used in the study**

In this study, a set of routines (mat_chi) based on Matlab software were coded following Eq. 3 to 6 for estimating $\chi$. Additional modules and routines based on the integrated statistical package R were also used for estimating dependence terms and inter-comparing various parameters. Emphasis was given on the routine "taildep" of the module "extRemes" (https://cran.r-project.org/web/packages/extRemes/extRemes.pdf) that is capable of estimating $\chi$ values when a critical percentile (extreme) threshold is considered.

Another "powerful" routine capable of providing a variety of dependence graphs and plots (besides single estimated values of $\chi$) has been the routine "chiplot" of the module "evd" (Extreme Value Distributions) of R (https://cran.r-project.org/web/packages/evd/evd.pdf). The routine chiplot is also capable of providing confidence intervals at any preselected level. As mentioned above (Sect. 2) relatively small differences among various estimates made by chiplot of evd (R), taildep of extRemes (R) and mat_chi (Matlab) were found and this most probably is due to the unavoidable dissimilarities between the criteria being imposed on data pairs when applying POT methodology.

Examples of estimated statistical dependence ($\chi$) values between surge (HvH) and wave (LiG) max24 values in obs_com (upper panel), hind_com (middle panel) and in hind_tot (lower panel) mode by chiplot routine of evd module (R) are given in Fig. 1.

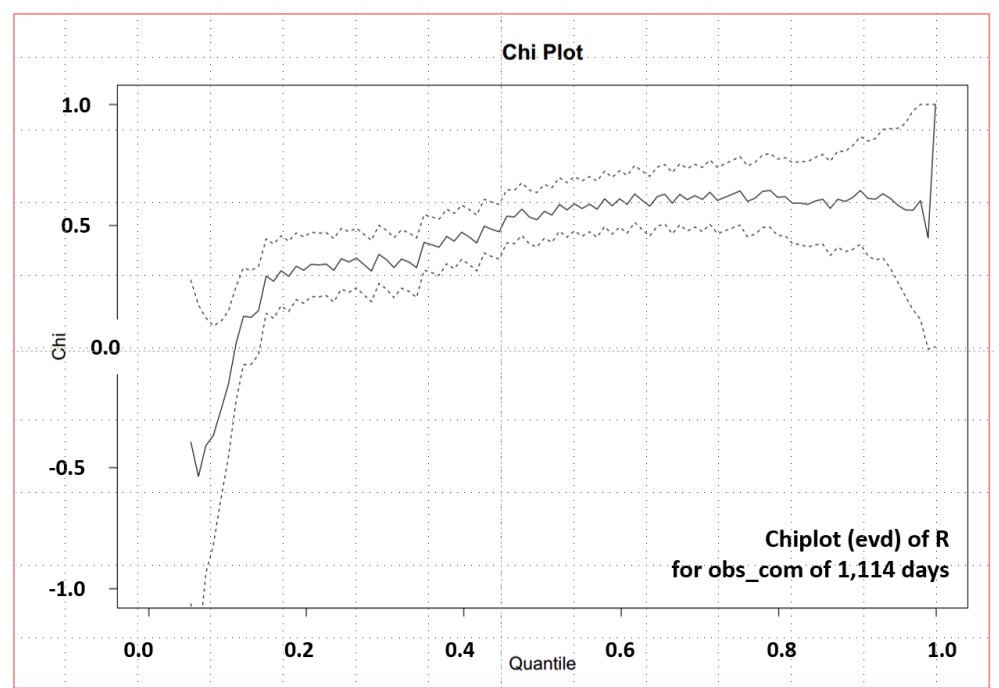

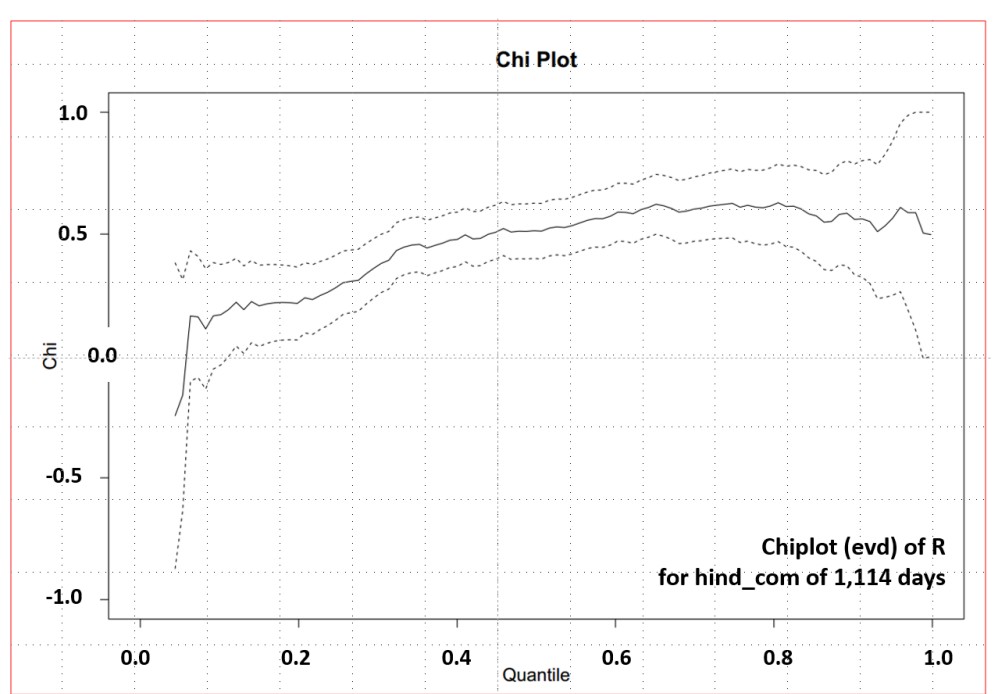

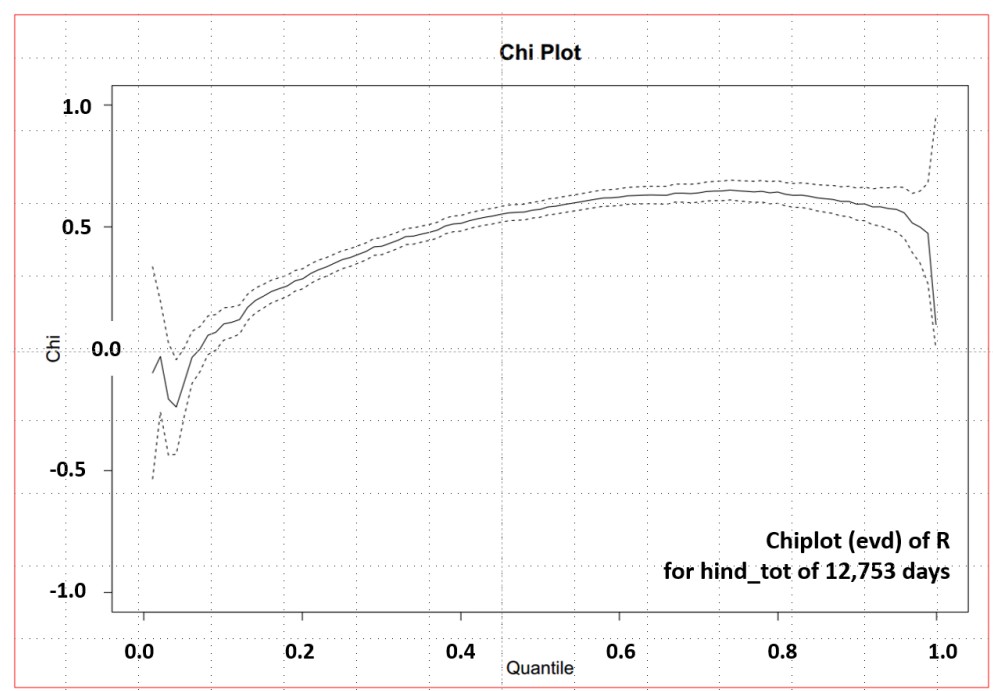

**Fig. 1.** Estimated χ values between surge (HvH) and wave (LiG) max24 values in c (upper panel) & hind_com (middle panel) and in hind_tot (lower panel) mode by chiplot routine of evd module (R).

Studying closely Fig. 1 it becomes obvious that considerable high values of dependence are estimated over all three (obs_com, hind_com & hind_tot) modes. The importance and implications of such high values of dependence can be demonstrated with an example by considering the total hindcast (hind_tot) series for surge (HvH) and wave (LiG). Utilising the Matlab function "gevfit" an estimation of the return levels having a 100-year return period for surge and wave height variables was made (1.78 and 6.05 metres respectively). Inserting the common return period value (100-year) together with the estimated χ value (0.56)

in Eq. 12, the Joint Return Period (JRP) of such a compound event (surge ≥ 1.78 metres and significant wave height ≥ 6.05 metres) was estimated at ~179 years.

Such a value (~179 years) is significantly different from the value of 10,000 years representing the estimated JRP assuming that surge and wave variables were totally independent. In a case like this (of independent events), the dependence would have

been equal to zero and the JRP would be given by the product of their individual probabilities (Blank, 1982).

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

*Technical Supplement of*

# Estimations of statistical dependence as joint return period modulator of compound events. Part I: storm surge and wave height.

Thomas I. Petroliagkis

*Correspondence to*: Thomas I. Petroliagkis (thomas.petroliagkis@ec.europa.eu)

## *Contents*

**1 Details of RIEN (RIver ENding) point positions**

The current statistical (dependence) analysis is focused over 32 river ending points that have been selected to cover a variety of riverine and estuary areas along European coasts. The sea areas used in the study refer to the Mediterranean Sea (central and north Adriatic Sea, Balearic Sea, Alboran Sea and Gulf of Lion), West Iberian, North Iberian, Bay of Biscay, Irish Sea, Bristol Channel, English Channel, North Sea, Norwegian Sea, Baltic Sea and Black Sea. A map showing the position of RIEN (RIver ENding) points used in the study is shown in Fig. 1 of the main text. Additional details can be found in Table 1 (current Technical Supplement) containing the exact location (lat, lon) of all RIEN points

**Table 1.** Positions (lat, lon) of 32 RIEN points used in the study. Names refer to river ending areas.

| | RIEN | lat | lon | | RIEN | lat | lon |
|---|---|---|---|---|---|---|---|
| 1 | Po Della Pila | 44.96 | 12.49 | 17 | Muir Eireann | 52.65 | -6.22 |
| 2 | Madonna Del Ponte | 43.83 | 13.05 | 18 | Wallasey | 53.44 | -3.04 |
| 3 | Martinsicuro | 42.84 | 13.93 | 19 | Severn Bridge | 51.61 | -2.65 |
| 4 | Aries | 43.34 | 4.84 | 20 | Fort Picklecombe | 50.34 | -4.17 |
| 5 | El Foix | 41.20 | 1.67 | 21 | Exmouth | 50.62 | -3.42 |
| 6 | Illa de Buda | 40.71 | 0.89 | 22 | Christchurch District | 50.72 | -1.74 |
| 7 | Rio De Velez | 36.72 | -4.11 | 23 | Dieppe | 49.91 | 1.09 |
| 8 | Matosinhos | 41.18 | -8.71 | 24 | South Tynesid | 55.01 | -1.43 |
| 9 | Carcavelos | 38.69 | -9.26 | 25 | Spurm Point | 53.57 | 0.11 |
| 10 | Setubal | 38.53 | -8.89 | 26 | Sheerness | 51.45 | 0.74 |
| 11 | San Bruno | 37.18 | -7.39 | 27 | Western Scheldt | 51.43 | 3.55 |
| 12 | Punta Del Arenal | 43.47 | -5.07 | 28 | Rockanje | 51.87 | 4.01 |
| 13 | Concarneau | 47.86 | -3.92 | 29 | Wurster Arm | 53.65 | 8.14 |
| 14 | Riviere De Belon | 47.81 | -3.72 | 30 | Kattegat | 57.77 | 11.76 |
| 15 | Larmor-Plage | 47.71 | -3.38 | 31 | Trondheimsfjord | 63.32 | 9.82 |
| 16 | Musura Bay | 45.22 | 29.73 | 32 | Vanhankaupunginselka | 60.24 | 24.99 |

**2 Additional validation of wave hindcasts (focusing on extremes).**

The set of storm surge hindcasts originated from Vousdoukas et al. (2016) have specifically used for projections of extreme storm surge levels along Europe and it seems as an appropriate dataset for the current paper. On the other hand, wave hindcasts based on the ERA5 significant wave reanalysis dataset have not been thoroughly tested as for the validity of their extreme

values since ERA5 has been in production phase (https://www.ecmwf.int/en/about/media-centre/science-blog/2017/era5-new-reanalysis-weather-and-climate-data).

Due to this (limitation), an investigation was performed over a set of 13 wave buoys along European coasts capable of providing enough hourly data for such an analysis. The details of wave buoys used are shown in Table 1. Buoys over

Mediterranean are denoted as MED, over Bay of Biscay as BIS, over Irish Sea as IRI and over North Sea as NOS. HvH-LiG refers to the wave buoy Lighteland Goeree stationed near the coastal area of Hook van Holland (NL).

**Table 2.** Details of wave buoys used in the validation of wave extremes.

|       | id     | name            | lat   | lon   | days  | corr |
|-------|--------|-----------------|-------|-------|-------|------|
| M     | 61217  | Adriatic Sea    | 42.41 | 14.54 | 203   | 0.94 |
| E     | 61218  | Adriatic Sea    | 43.83 | 13.72 | 1,588 | 0.87 |
| D     | 61280  | Balearic Sea    | 40.69 | 1.48  | 2,764 | 0.87 |
| B I S | 62001  | Bay of Biscay   | 45.20 | -5.00 | 6,012 | 0.97 |
| I     | 62091  | Irish Sea       | 53.48 | -5.43 | 4,991 | 0.93 |
| R     | 62094  | Irish Sea       | 51.70 | -6.70 | 3,727 | 0.94 |
| I     | 62301  | Irish Sea       | 52.40 | -4.70 | 5,339 | 0.93 |
|       | 62303  | Bristol Channel | 51.50 | -5.10 | 7,426 | 0.94 |
| N     | 62127  | North Sea       | 54.00 | 0.70  | 1,158 | 0.92 |
| O     | 62142  | North Sea       | 53.00 | 2.10  | 5,537 | 0.92 |
| S     | 62145  | North Sea       | 53.10 | 2.80  | 5,796 | 0.92 |
|       | 63115  | North Sea       | 61.60 | 1.30  | 2,922 | 0.97 |
|       | HvH-LiG| North Sea       | 51.93 | 3.40  | 1,114 | 0.92 |

Based on these 13 wave buoys listed in Table 2, a dataset of 48,547 pairs of daily maxima was compiled comprising hindcast

and observation values. The mean error (bias) of hindcasts was found to be equal to -0.29 m with a corresponding rmse of 0.56 m. From a closer investigation, it became obvious that there were cases with hindcasts not capturing the exact magnitude of extremes. In such cases emphasis was given to the possibility of capturing (resolving) the extremes as spikes, i.e., as "footprints" of extreme values.

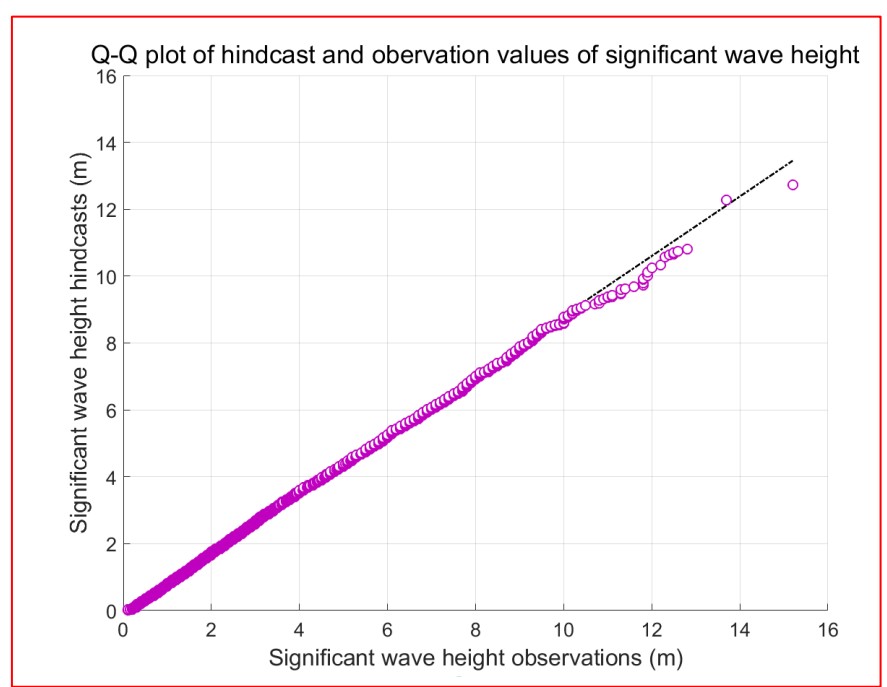

Figure 1. Q-Q (quantile-quantile) plot of hindcast and observation values of significant wave height.

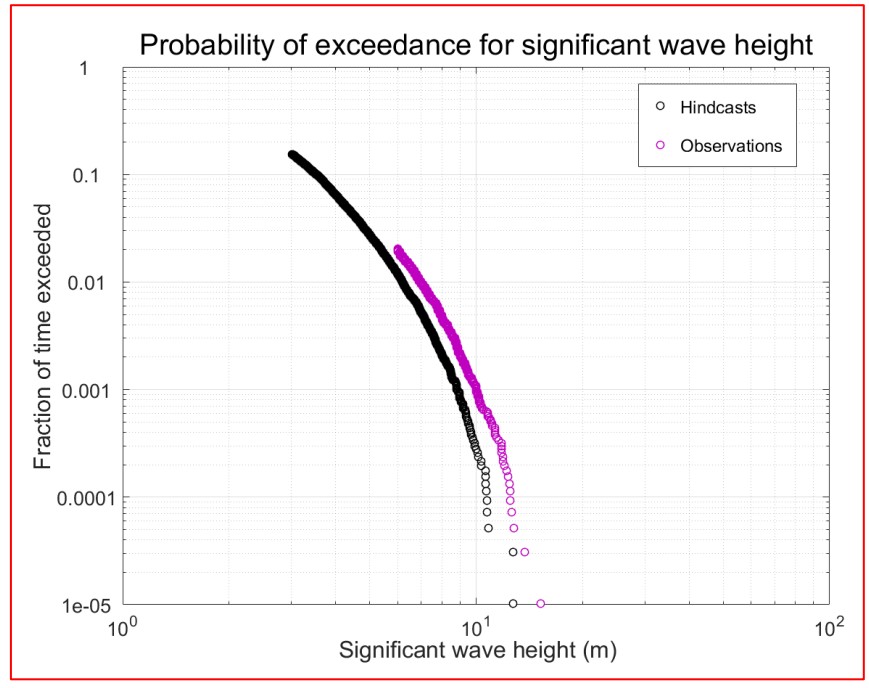

Figure 2. PoE (Probability of Exceedance) for wave observations (red color) and hindcasts (blue color).

In addition, both Figure 1 (Quantile-Quantile Plot) and Figure 2 (Probability of Exceedance Plot) seem to support this unavoidable limitation since due to the low resolution models used for reproducing time series of significant weather parameters (as in this case), extremes cannot be captured with their exact (high-impact) value but in most cases their "footprint" signal can be resolved as a spike of a lesser value. A relevant example can be seen in Petroliagis and Pinson (2012) where the footprints of extreme wind speed values over Bremen airport are captured by ERA-Interim as footprint spikes although significantly underestimated (compared to observations) but still capable of resolving extremes as shown in Figure 7 of Petroliagis & Pinson.

Table 3: Number of hits for various hindcast and observation thresholds (percentiles).

| thrs | hind | obs | events | hits | score |
|------|------|------|--------|-------|-------|
| 55 | 1.51 | 1.80 | 6,617 | 6,129 | 93 % |
| 60 | 1.81 | 2.10 | 5,673 | 5,135 | 91 % |
| 65 | 1.97 | 2.30 | 5,041 | 4,586 | 91 % |
| 70 | 2.17 | 2.50 | 4,506 | 4,023 | 89 % |
| 75 | 2.41 | 2.80 | 3,737 | 3,329 | 89 % |
| 80 | 2.69 | 3.05 | 3,281 | 2,821 | 86 % |
| 85 | 3.05 | 3.40 | 2,527 | 2,147 | 85 % |
| 90 | 3.53 | 4.00 | 1,699 | 1,439 | 85 % |
| 91 | 3.65 | 4.10 | 1,586 | 1,314 | 83 % |
| 92 | 3.77 | 4.30 | 1,394 | 1,159 | 83 % |
| 93 | 3.91 | 4.50 | 1,243 | 1,033 | 83 % |
| 94 | 4.08 | 4.70 | 1,070 | 881 | 83 % |
| 95 | 4.29 | 4.90 | 948 | 768 | 81 % |
| 96 | 4.56 | 5.20 | 775 | 620 | 80 % |
| 97 | 4.89 | 5.60 | 601 | 479 | 80 % |
| 98 | 5.38 | 6.20 | 417 | 337 | 81 % |
| 99 | 6.18 | 7.20 | 233 | 186 | 80 % |

In a similar way, during the estimation of statistical dependence such footprints seem to be capable of determining the days of the most extreme wave daily maxima. The main issue in estimating dependence is not the exact magnitude of extremes (i.e., how well are resolved by hindcasts) but rather if a spike (footprint of an extreme) exists on a specific day denoting the exceedance over a critical percentile threshold of hindcasts. If such (correct) footprint is considered as a hit, Table 3 was
compiled containing the number of hits over a set of critical (hindcast & observation) wave thresholds in a POT (Peaks Over Threshold) environment.

Taking into consideration that during the estimation of dependence (Table 6 and Table 7 of Technical Supplement), threshold (percentile) wave values ranging from 86.2 to 98.8% were used, this corresponds to 80 to 85% hits (i.e., correct footprint spikes
of daily maxima denoting an extreme).

Lastly, in cases of **compound (surge & wave)** footprints of extremes (resolved by hindcasts), Table 4 (Technical Supplement) has been compiled where the 98.5% percentile extremes of storm surge observations are compared to their corresponding hindcast values (falling in the same 98.5% category). Same way in Table 5 (Technical Supplement), the footprints of significant
wave height observation extremes are compared to their corresponding hindcast (or lesser intensity) values.

It becomes obvious that although hindcasts could not resolve the exact extremity of both surge and wave events, they were able to capture their footprints quite well. It is important to point out that hindcasts above all were capable of identifying and resolving all seven (7) compound events that took place during the common time interval of 1,114 days referring to the RIEN
point of Rhine River.

**2 3 Capability of storm surge and wave hindcasts to identify and resolve compound events.**

As already mentioned long-period water level data coinciding with wave observations directly or very close to the exact sites of interest (RIEN points) were not available with the exception of the Rhine River (RIEN). For this RIEN, concurrent (close-by) observations with no gaps of sea level, astronomical tide, storm surge, and wave height from a close-by wave buoy were available for a period of about 3 years (1,114 days).

In Table 4, extreme storm surge (above 98.5% percentile) values for both observations and hindcasts for HvH tide gauge station over the common time interval of 1,114 days are shown. Same way extreme significant wave height (above 98.5% percentile) values for both observations and hindcasts for LiG wave buoy station over the common time interval are contained in Table 5.

**Table 4.** Extreme storm surge (above 98.5% percentile) values for observations (>0.95m) and hindcasts (>0.89m) for HvH tide gauge station over the common time interval of 1,114 days. Compound events of surge and wave (i.e., both surge & wave above critical threshold) are marked by orange shade.

| # | Date | Observations | hindcasts |
|---|------|--------------|-----------|
| 1 | 12 Nov 2010 | 1.38 | 1.10 |
| 2 | 4 Feb 2011 | 1.20 | 1.00 |
| 3 | 27 Nov 2011 | 1.25 | 1.04 |
| 4 | 28 Nov 2011 | 0.98 | 0.93 |
| 5 | 3 Dec 2011 | 1.08 | 1.03 |
| 6 | 7 Dec 2011 | 1.10 | 0.95 |
| 7 | 9 Dec 2011 | 1.45 | 1.23 |
| 8 | 29 Dec 2011 | 1.23 | 1.03 |
| 9 | 3 Jan 2012 | 1.07 | 0.47 |
| 10 | 4 Jan 2012 | 1.46 | 1.16 |
| 11 | 5 Jan 2012 | 1.66 | 1.59 |
| 12 | 6 Jan 2012 | 1.37 | 1.57 |
| 13 | 21 Jan 2012 | 1.09 | 1.02 |
| 14 | 22 Jan 2012 | 1.00 | 1.07 |
| 15 | 30 Jan 2013 | 1.07 | 0.73 |
| 16 | 10 Sep 2013 | 0.96 | 0.59 |

Compound events of surge and wave are marked by orange shade (in both Table 4 and Table 5) based on joint observations of storm surge and significant wave height. It becomes obvious that hindcasts were able to resolve all seven (7) compound events that took place during the common time period of 1,114 days.

**Table 5.** Extreme wave height (above 98.5% percentile) values for observations (> 4.07m) and hindcasts (>3.38m) for LiG wave buoy station over the common time interval of 1,114 days. Compound events of surge and wave (i.e., both surge & wave above critical threshold) are marked by orange shade.

| # | Date | Observations | hindcasts |
|---|------|--------------|-----------|
| 1 | 12 Nov 2010 | 4.79 | 3.99 |
| 2 | 14 Jul 2011 | 4.61 | 3.34 |
| 3 | 7 Oct 2011 | 4.34 | 3.34 |
| 4 | 7 Dec 2011 | 5.06 | 4.83 |
| 5 | 8 Dec 2011 | 4.49 | 3.87 |
| 6 | 9 Dec 2011 | 4.17 | 3.53 |
| 7 | 24 Dec 2011 | 4.37 | 3.27 |
| 8 | 29 Dec 2011 | 4.18 | 3.46 |
| 9 | 30 Dec 2011 | 4.66 | 3.84 |
| 10 | 4 Jan 2012 | 4.31 | 4.02 |
| 11 | 5 Jan 2012 | 5.14 | 4.79 |
| 12 | 6 Jan 2012 | 4.55 | 4.90 |
| 13 | 20 Jan 2012 | 4.15 | 2.81 |
| 14 | 31 Aug 2012 | 4.11 | 3.24 |
| 15 | 24 Sep 2012 | 4.61 | 3.43 |
| 16 | 25 Nov 2012 | 4.36 | 4.09 |

10 Further, an extra investigation based on extreme values of observations (during the common time interval of 1,114 days) exceeding a variety of percentile values (for the RIEN point of Rhine River) showed that both storm surge and their corresponding wave height hindcasts were able to capture almost all of the 24-hour extremes on the same (correct) day but with a weaker intensity (i.e., with a correct footprint of lesser intensity).

~~3~~ 4 **Analytical values of correlation and statistical dependence based on Matlab routines.**

A necessary split of results had to be made for a better and easier visualisation due to the relatively large amount of RIEN points to fit in one single Table. This split also revealed the distinct differences between southern and northern coastal European areas. Details of both correlations and dependencies found over southern and northern RIEN points are presented analytically in Table 6 and Table 7 based on Matlab routines. Correlation (corr) and dependence (chi) values for both max12 and max24 intervals are presented together with critical threshold (thrs), significance (sig) and 95% confidence level (lower & upper) max24 values. Referring to correlation values, a large amount of variability is evident in both max12 and max24 modes

**Table 6.** Correlation and statistical dependence values for storm surge and significant wave heights over Mediterranean (ADR: Adriatic Sea – GOL: Gulf of Lion – BAL: Balearic Sea – ALB: Alboran Sea), West and North Iberian coasts (WIB & NIB), Bay of Biscay (BOB) and Black Sea (BLK) based on Matlab routines.

| | RIEN | sea | max12 | | | max24 | | | | | | |
|---|---|---|---|---|---|---|---|---|---|---|---|---|
| | | | corr | thrs | chi | corr | thrs | chi | chibar | sig | lower | upper |
| 1 | Po | ADR | 0.26 | 97.4 | 0.28 | 0.39 | 97.1 | 0.29 | 0.43 | 0.02 | 0.21 | 0.37 |
| 2 | Metauro | ADR | 0.23 | 96.8 | 0.26 | 0.35 | 95.7 | 0.22 | 0.30 | 0.05 | 0.03 | 0.35 |
| 3 | Vibrata | ADR | 0.23 | 96.6 | 0.35 | 0.37 | 96.5 | 0.32 | 0.36 | 0.04 | 0.23 | 0.37 |
| 4 | Rhone | GOL | 0.08 | 94.6 | 0.20 | 0.13 | 93.8 | 0.21 | 0.17 | 0.04 | 0.13 | 0.30 |
| 5 | Foix | BAL | 0.09 | 92.2 | 0.03 | 0.10 | 91.2 | 0.03 | 0.05 | 0.03 | 0.00 | 0.08 |
| 6 | Ebro | BAL | 0.04 | 94.7 | 0.19 | 0.12 | 94.5 | 0.22 | 0.22 | 0.03 | 0.10 | 0.30 |
| 7 | Velez | ALB | 0.02 | 93.9 | 0.19 | 0.06 | 93.1 | 0.11 | 0.13 | 0.04 | 0.05 | 0.17 |
| 8 | Douro | WIB | -0.18 | 97.0 | 0.30 | -0.06 | 95.7 | 0.30 | 0.30 | 0.05 | 0.11 | 0.38 |
| 9 | Tagus | WIB | -0.30 | 94.3 | 0.05 | -0.22 | 93.7 | 0.14 | 0.16 | 0.03 | 0.09 | 0.22 |
| 10 | Sado | WIB | -0.26 | 94.9 | 0.10 | -0.19 | 93.9 | 0.13 | 0.17 | 0.03 | 0.06 | 0.21 |
| 11 | Guadiana | WIB | -0.04 | 95.9 | 0.22 | 0.03 | 95.7 | 0.28 | 0.29 | 0.02 | 0.15 | 0.36 |
| 12 | Sella | NIB | -0.25 | 93.2 | 0.10 | -0.17 | 86.2 | 0.14 | 0.07 | 0.05 | 0.07 | 0.19 |
| 13 | Moros | BOB | 0.07 | 96.2 | 0.32 | 0.22 | 96.2 | 0.30 | 0.34 | 0.03 | 0.17 | 0.39 |
| 14 | Aven | BOB | 0.13 | 97.0 | 0.34 | 0.25 | 96.7 | 0.35 | 0.39 | 0.01 | 0.23 | 0.42 |
| 15 | Blavet | BOB | 0.11 | 96.5 | 0.33 | 0.25 | 96.7 | 0.34 | 0.39 | 0.02 | 0.22 | 0.40 |
| 16 | Danube | BLK | -0.01 | 96.7 | 0.21 | 0.09 | 96.3 | 0.24 | 0.35 | 0.05 | 0.07 | 0.38 |

**Table 7.** As in Table 6 but for Irish Sea (IRS), Bristol Channel (BRC), English Channel (ENC), North Sea (NRS), Norwegian Sea (NOS) and Baltic Sea (BAS). Owena stands for Owenavarragh RIEN (IE) while Goeta is Goeta Aelv RIEN (ES).

| | RIEN | sea | max12 | | | max24 | | | | | | |
|---|---|---|---|---|---|---|---|---|---|---|---|---|
| | | | corr | thrs | chi | corr | thrs | chi | chibar | sig | lower | upper |
| 17 | Owena | IRS | 0.50 | 98.4 | 0.46 | 0.59 | 97.9 | 0.45 | 0.53 | 0.05 | 0.30 | 0.55 |
| 18 | Mersey | IRS | 0.45 | 98.2 | 0.43 | 0.56 | 97.4 | 0.43 | 0.48 | 0.03 | 0.29 | 0.52 |
| 19 | Severn | BRC | 0.19 | 96.1 | 0.29 | 0.30 | 94.9 | 0.30 | 0.24 | 0.04 | 0.22 | 0.35 |
| 20 | Tamar | ENC | 0.28 | 97.8 | 0.35 | 0.39 | 96.9 | 0.35 | 0.41 | 0.02 | 0.24 | 0.49 |
| 21 | Exe | ENC | 0.31 | 97.9 | 0.38 | 0.41 | 97.1 | 0.40 | 0.43 | 0.03 | 0.29 | 0.54 |
| 22 | Avon | ENC | 0.37 | 98.1 | 0.44 | 0.50 | 97.9 | 0.48 | 0.55 | 0.04 | 0.35 | 0.58 |
| 23 | Bethune | ENC | 0.59 | 99.1 | 0.62 | 0.68 | 98.8 | 0.64 | 0.77 | 0.02 | 0.55 | 0.73 |
| 24 | Tyne | NRS | 0.14 | 91.7 | 0.31 | 0.28 | 94.5 | 0.26 | 0.21 | 0.05 | 0.10 | 0.39 |
| 25 | Humber | NRS | 0.18 | 97.3 | 0.35 | 0.38 | 96.6 | 0.35 | 0.37 | 0.04 | 0.20 | 0.49 |
| 26 | Thames | NRS | -0.10 | 92.6 | 0.22 | 0.06 | 92.7 | 0.22 | 0.11 | 0.05 | 0.11 | 0.31 |
| 27 | Schelde | NRS | 0.31 | 97.6 | 0.54 | 0.54 | 97.5 | 0.53 | 0.50 | 0.01 | 0.45 | 0.61 |
| 28 | Rhine | NRS | 0.52 | 98.5 | 0.57 | 0.67 | 98.0 | 0.56 | 0.57 | 0.03 | 0.41 | 0.64 |
| 29 | Weser | NRS | 0.56 | 99.0 | 0.58 | 0.65 | 98.5 | 0.56 | 0.69 | 0.02 | 0.42 | 0.63 |
| 30 | Goeta | NRS | 0.43 | 97.2 | 0.53 | 0.55 | 96.8 | 0.51 | 0.39 | 0.05 | 0.44 | 0.61 |
| 31 | Orkla | NOS | 0.35 | 97.6 | 0.46 | 0.46 | 97.0 | 0.41 | 0.43 | 0.03 | 0.33 | 0.50 |
| 32 | Vantaa | BAS | 0.30 | 97.0 | 0.43 | 0.44 | 96.9 | 0.44 | 0.42 | 0.03 | 0.36 | 0.50 |

**4 5 Analytical values of correlation and statistical dependence based mainly on R routines.**

Details of both correlations and dependencies found over southern and northern RIEN points are presented analytically in Table 8 and Table 9 based mainly on R routines.

**Table 8.** As in Table 6, but based mainly on R (chiplot & taildep) routines. Ensemble mean (comb) values of dependence are also shown (last column).

| | RIEN | sea | R lower | R upper | R chiplot | R taildep | MAT mat_chi | ENS comb |
|---|---|---|---|---|---|---|---|---|
| 1 | Po | ADR | 0.13 | 0.34 | 0.23 | 0.27 | 0.29 | 0.26 |
| 2 | Metauro | ADR | 0.08 | 0.26 | 0.17 | 0.22 | 0.22 | 0.20 |
| 3 | Vibrata | ADR | 0.13 | 0.32 | 0.23 | 0.36 | 0.32 | 0.30 |
| 4 | Rhone | GOL | 0.06 | 0.21 | 0.14 | 0.22 | 0.21 | 0.19 |
| 5 | Foix | BAL | 0.01 | 0.13 | 0.07 | 0.16 | 0.03 | 0.09 |
| 6 | Ebro | BAL | 0.14 | 0.30 | 0.22 | 0.28 | 0.22 | 0.24 |
| 7 | Velez | ALB | 0.03 | 0.18 | 0.10 | 0.16 | 0.11 | 0.12 |
| 8 | Douro | WIB | 0.17 | 0.33 | 0.26 | 0.31 | 0.30 | 0.29 |
| 9 | Tagus | WIB | 0.07 | 0.21 | 0.14 | 0.22 | 0.14 | 0.17 |
| 10 | Sado | WIB | 0.08 | 0.21 | 0.14 | 0.21 | 0.13 | 0.17 |
| 11 | Guadiana | WIB | 0.19 | 0.34 | 0.27 | 0.32 | 0.28 | 0.29 |
| 12 | Sella | NIB | 0.05 | 0.19 | 0.12 | 0.18 | 0.14 | 0.15 |
| 13 | Moros | BOB | 0.14 | 0.32 | 0.23 | 0.28 | 0.30 | 0.27 |
| 14 | Aven | BOB | 0.18 | 0.37 | 0.27 | 0.31 | 0.35 | 0.31 |
| 15 | Blavet | BOB | 0.17 | 0.36 | 0.27 | 0.30 | 0.34 | 0.30 |
| 16 | Danube | BLK | 0.13 | 0.32 | 0.23 | 0.26 | 0.24 | 0.24 |

**Table 9.** As in Table 7, but based mainly on R (chiplot & taildep) routines. Ensemble mean (comb) values of dependence are also shown (last column).

| | RIEN | sea | R | | | | MAT | ENS |
| --- | --- | --- | --- | --- | --- | --- | --- | --- |
| | | | lower | upper | chiplot | taildep | mat_chi | comb |
| 17 | Owena | IRS | 0.26 | 0.52 | 0.39 | 0.40 | 0.45 | 0.41 |
| 18 | Mersey | IRS | 0.26 | 0.48 | 0.38 | 0.38 | 0.43 | 0.40 |
| 19 | Severn | BRC | 0.16 | 0.32 | 0.24 | 0.30 | 0.30 | 0.28 |
| 20 | Tamar | ENC | 0.21 | 0.41 | 0.31 | 0.34 | 0.35 | 0.33 |
| 21 | Exe | ENC | 0.25 | 0.46 | 0.36 | 0.38 | 0.40 | 0.38 |
| 22 | Avon | ENC | 0.33 | 0.57 | 0.45 | 0.46 | 0.48 | 0.46 |
| 23 | Bethune | ENC | 0.49 | 0.80 | 0.64 | 0.66 | 0.64 | 0.65 |
| 24 | Tyne | NRS | 0.11 | 0.27 | 0.19 | 0.26 | 0.26 | 0.24 |
| 25 | Humber | NRS | 0.20 | 0.40 | 0.30 | 0.33 | 0.35 | 0.33 |
| 26 | Thames | NRS | 0.08 | 0.22 | 0.15 | 0.25 | 0.22 | 0.21 |
| 27 | Schelde | NRS | 0.36 | 0.58 | 0.47 | 0.48 | 0.53 | 0.49 |
| 28 | Rhine | NRS | 0.41 | 0.64 | 0.52 | 0.54 | 0.56 | 0.54 |
| 29 | Weser | NRS | 0.40 | 0.67 | 0.55 | 0.54 | 0.56 | 0.55 |
| 30 | Goeta | NRS | 0.35 | 0.53 | 0.44 | 0.46 | 0.51 | 0.47 |
| 31 | Orkla | NOS | 0.25 | 0.45 | 0.35 | 0.38 | 0.41 | 0.38 |
| 32 | Vantaa | BAS | 0.27 | 0.48 | 0.37 | 0.40 | 0.44 | 0.40 |

For the analysis of results, the ensemble mean value of $\chi$ (by averaging mat_chi, chiplot and taildep values) is taken as a reference value (contained in the last column of Table 8 and Table 9). The different categories of correlation and dependence used in the main text (and in Fig. 10) refers to the categorisation adapted by Defra TR1 Report (2005) and TR3 Report (2005).

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
