# Peer review of "Estimations of statistical dependence as joint return period modulator of compound events. Part I: storm surge and wave height."

_Natural Hazards and Earth System Sciences, 2017_

## Referee Comment (RC1) · Anonymous Referee #1 · 21 Aug 2017

—— General Comment

This paper addresses an important issue: the probability of marine storms characterized by the simultaneous presence of high waves and large storm surges. On the basis of two hindcast studies (one for waves and another for surges) it describes the dependence and the correlation between the two components of marine storminess at 32 points, which are located in correspondence of river mouths along the coastline of north and south Europe.

I find the subject interesting and results potentially worth to be published. However, I recommend that the author improves his manuscript. Some, hopefully helpful, suggestions are in the specific comments here below.

In fact, the paper needs major improvements for being publishable. It relies on intense/extreme events simulated by hindcast studies without providing sufficient information on their validation. The description of the statistical method should be more precise The presentation of results should be improved by optimizing tables and figures. Causes of spatial variation of dependent and correlation should be better discussed. Parts containing details in a form similar to a technical report should be removed from the main body of the text.

————Specific comments

1) Validation of hincasts and its presentation

No sufficient attention is paid to assessing whether storm surge and wave simulations are capable of reproducing extremes values. Correlation and bias are not representative in this sense. Further, the presentation is strongly asymmetric between waves and surges, with a discussion of percent errors for surge and absolute errors for waves. Further, there is no information on the spatial distribution of storm surge errors. The paper needs to present information on the spatial distribution of percent errors in reproduction of high storm surges and waves (possibly of their extremes). In general, I suggest to use maps with percent errors, which are much more effective than tables to present such information. Without this it is difficult to estimate how realistic conclusions are. Errors in timing are important and are not discussed. In my view, the statement in the conclusions "the overall performance of both surge and wave hindcasts is considered satisfactory" is not documented in the results

The local validation of maxima at the Rhine River ending point is very convincing. It is anyway not clear whether such good performance of the models can be extended to other selected stations. Is this validation possible in other stations in other parts of the domain so that reader can be convinced that results in terms of correlation and dependence are convincing across the domain?

Section 4.2 line 16-17 the statement "Overall, it seems that hindcasts in this case were able of resolving and estimating both the correct type and strength of correlation between source variables." Could this be better enlightened at least for the Rhine station where data are available. How do we assess what is the real correct statistical dependence between surge and waves?

2) Description of statistical methods.

The description of the method should be clear also to a reader not familiar with the involved statistical methods. Some details appear confusing. Eventually, if clarifying them requires too much text, I suggest the author to publish it in the supplementary material. Here is a list of points that I recommend to clarify.

Line 1 page 5 writes that a transformation is adopted (please describe it) to produce identical marginal distribution. Line 4 writes that a copula function is used to diminish the effect of different marginal distributions. The two statement do not appear consistent to me.

In eq(1) the dependence chi is defined for z* (upper limit of the observations), while in eq (3) is defined for any generic level u. Please explain this apparent inconsistency

The derivation of eq.(3) does not appear straightforward to me. Please add a reference

In eqs.(3-5) the relation between U, V, u and X, Y, x* is not provided in the text.

The way in which chiˆbar (statistical dependence of asymptotically independent variables) is computed is not given. Distinction between chi and chiˆbar is not well explained

The concept of asymptotic dependence is not explicitly stated

It is not described how correlation is computed. Is it correlation between time series of hourly (or 3-hourly or 6-hourly) values of surge levels and wave heigh? Is correlation between the sequence of daily maxima? Between the sequence of maxima in 12 hours

long windows?

Provide a precise definition of definition of compound events as adopted in this study

Clarify the criterion leading to the selection of top 80 evennts

In Chapter 2.2, after the discussion, I cannot find the information on the values of alpha and u actually used in this study.

page 4 lines 16 The statement "hydro-meteorological analyses based on real data often lead to an assessment of complete independence that could result to an underestimation of the joint probability of concurrent extreme events" is written in an ambiguous form. Please explain how joint probability is underestimated if data are "real" and the analysis is correct.

I am confused by section 2.2 (which I fail to follow concerning the selection of the chi value) and section 2.4. Establishing a confidence interval (section 2.3) should be sufficient for assessing the significance of the computed dependence values. Is here a duplication of information?

3) Spatial variations of dependence and correlation, interpretation of results

The discussion of the spatial distribution of correlation and dependence and explanation for the differences is rather inconclusive. The author writes that "dependence is likely to occur when different processes are linked to some common weather (forcing) conditions" but no convincing investigation is made on that respect. Lack of dependence could for instance be explained by a substantial contribution of inverse barometer effect to storm surges, but there is no mention of this in the paper.

Figures with the spatial distribution of correlation and dependence would be very useful. I suggest to replace the corresponding tables with maps

Section 4.5 des not provide interesting interpretation of results. Interpretation of results in term of understanding factors leading to compound events is not provided Further

annotation in figure 12 is not readable . Interpretation of results at the Rhone river mouth does not account for the possibility that many surge events are produced by inverse barometric effect and not by winds.

At some stations, wind during compound events is blowing offshore. Local high waves are unlikely caused by those winds

Actual definition of prevailing and dominant wind is not clear to me (page 3, line 29-30)

4) Parts to be removed from the main body the text

A part of the paper is devoted to differences between the results produced by two software packages: R and Mathlab. Lines such as 19-26 at page 5 are interesting in a technical report, but of limited interest for a scientific paper. The cause of differences is not discussed and it is not clear whether it has a scientific relevance. Lines 16-18 at page 6 write that "Relatively small differences among various estimates made by chiplot of evd (R), taildep of extRemes (R) and mat_chi (matlab) were found. This most probably is due to the unavoidable dissimilarities between the criteria being imposed on data pairs when applying POT methodology (selection of different critical thresholds)". Continuing along this comment. . . Table3 and 4(and analogously 5 and 6) are presented as a comparison between packages, which is correct in a technical report but not in a scientific paper. I suggest to skip this discussion or eventually use the possibility of providing supplementary material for explaining technical differences between software packages and how they are used.

——— Other points and technical corrections:

Table7 Is redundant with respect figure 7

Figure 8 wind rose and related annotation in this figure redundant in my opinion I failed to find the " Defra/Environment Agency R&D Technical Report FD2308/TR3 on-line. I recommend the web link for downloading this and other technical reports to be provided in the reference list

Page 7, line 6 graphically or empirically?

Abstract line 14 adapted or adopted?

Lines 13-14 ref to personal communication (which cannot be properly documented) looks useless here

page 11, line 22-23 refer to "personal communication", which I think is not suitable in this form

Table 1 is not needed in the main body of the text Figure1 provides the same information

I do not find a clear explanation on which data are grouped under the lable hind_com,obs_com and Hind_tot . One can guess but a clear description should be given in data and method.

Results section contain description of tools (lines 12-18, page 18) . This should be moved to section 2 or 3, or (preferably in my view) removed or transferred to a supplement.

Fig.10 I cannot see the negative and zero dependence values that are mentioned in the text (page40, line 15). . .

---

## Referee Comment (RC2) · Anonymous Referee #2 · 9 Oct 2017

General Comment The paper addresses compound events defined by combined high surges and high wind waves along European coastlines, especially in estuaries/river mouths. Statistical methods are used to investigate joint probabilities of compound events and the statistical dependency, since flood risk is not a function of one parameter (storm surges with peak value and duration) but usually of more (e.g. wind waves, river runoff). Large scale weather systems can cause either high storm surges or high wind waves and further more high precipitation and river runoff/discharges. Two sets of almost 35-year hindcasts of storm surges and wave heights were used to analyse the correlation and statistical dependency. As expected the frequency of the occurrence of the top compound events in different coastal areas were found to be higher during

the winter months. In the introduction the hydrological and meteorological conditions for high wind waves and extreme tidal surge events which can occur simultaneously with extreme precipitation events and high river flows (compound events) leading to increased flood risk is highlighted clearly. But the paper and the used methodology focused only on very few parameters. What is the background of the generalization? The subject of the paper is interesting yet a little confusing especially in the context of coastal engineering therefor the manuscript should be major improved. The paper and its structure is not easy to understand and the description of different data sets (and different time spans) of observed and modelled hindcast data is confusing (e.g. a lot of unusual abbreviations). The number of tables and especially the huge amount of data should be reduced as they are displayed in figures. The selected 32 stations at the end of the rivers or estuaries cover a wide variety of geographical areas and meteorological, oceanographical and hydrological (currents and tides) systems in coastal zones along European coasts. E.g. the tidal range varies from nearly zero to some meters and within the deterministic part of compound events in comparison to the stochastic part (surges, wind waves and river flow) of these compound events. Further discussion of the deterministic and the stochastic part of the compound events and the effects in the statistical analyses (dependency of different parameters) is recommended (page 41, line 26-30). In general I agree completely with reviewer # 1!

Comment 1 The description whether storm surge and/or wind waves are capable of reproducing extreme values is incomplete (e.g. river runoff?). It has to be explained, why river runoff is not taken into account!

Comment 2 In the context of the paper a very interesting problem is discussed where copula functions should be taken into account, so far only a simple approach for copula functions has been taken into consideration, the discussion of different copula functions within the scope of the addressed topic is to be considered, more references to copula functions could be helpful (e.g. Wahl, T., Jain, S., Bender, J., Meyers, S. D., & Luther, M. E. (2015). Increasing risk of compound flooding from storm surge and rainfall for

major US cities. Nature Climate Change, 5(12), 1093-1097).

Comment 3 A point of criticism is that the meteorological conditions and oceanographic system were not sufficiently described and the temporal developments of surges and wind waves are also not clearly described. E.g. in fig 10 and 12 it is shown that for the Weser (RIEN 29) the dependence for the prevailing (highest frequency) and dominant (highest intensity) wind during the top 80 extreme compound events are caused by wind direction from WSW. This is completely in contrast to my experience and has to be explained (same for e.g. RIEN 3, 12,... 23, 24, .... and 32)! From my point of view, it would be advisable to consider a subarea, e.g. only the North Sea, and after a successful investigation of the statistical dependence then implicate other areas.

Comment 4 (Length of observations/hindcasts) As I understood the water level data/storm surge/wind waves: The 32 RIEN (Table 1, page 10) were selected mainly because of their proximity to tidal gauges, although many of them cannot be evaluated due to lack of long-term measurements. For most RIENs, there are no data from nearby open wave buoys. Only for the Rhine (RIEN 28) are the tide and sea data (without data gaps) available from a nearby wave buoy for a period of 3 years. The validation of the combined hindcasts (tide and wind waves) was done on the basis of measured data at the Rhine (NL) was done on the tidal data at Hoek von Holland (HvH), wave buoy: Lichteiland (LiG) over a period of $\sim$ 3 years on measurement data without gaps and comparison of daily and half-day maxima. The generation of the hindcast of storm surge data was done with Delft3D-Flow (according to Vousdoukas et al. (2016) and the generation of the hindcast of the wind waves data was done with ECWAM wave model (according to Bidlot et al. (2006), Bidlot (2012), ECMWF (2015), Philips (2017)), e.g. $\sim$36 years, wind- and pressure fields from ERA-Interim (ERAI) (time resolution: 1 h, spatial resolution: 28x28 km, fixed water level, signif. wave height, max. wave height, mean wave period, mean wave direction and validation based on available records from 101 wave buoys throughout Europe + North Atlantic (1996-2015) (Fig. 2)) The overlapping period of the two hindcasts ($\sim$ 35 years) was used in statistical analysis.

The methodology of the research (using the hindcast data sets and observed data) has to be explained more detailed and especially what that means for the interpretation of the results (for all 32 RIEN). A time series of observed water level and wave buoy of only 3 years and only for one station in the area at Hoek van Holland seems to me as being not sufficient and much too short for comparison/evaluation with the modelled (hind cast) data and the conclusions. There should much more field data (water level, surges, wind waves, river runoff) available around the 32 RIEN! Approx. 2.3 "extreme events" (at least 3 days between peaks) per year (total 80 top events) were chosen. It has to be explained more detailed why 2.3 "extreme events" where chosen and what that means for the interpretation of the results.

Improvements: The number of tables and graphs should be reduced and more summarized. The paper is not easy to understand for a wide diversified audience, the length of the paper is too long and has partly too much redundancy (e.g. table 1 and fig. 1) The pure agreement between hindcast and observation of daily maximum of storm surges in Fig. 4 has to be explained. Why are small storm surges, e.g. below 0.5 m are taken in to account? What is the definition of a storm surge? What is the reason to use the storm between 25th December 2012 and 24th January 2013? The pure agreement between hindcast and observation of daily maximum of the significant wave height in Fig. 6 has to be explained. Fig. 8: The fairly pure agreement (chi) of the statistical dependence (chi) of storms surge and significant wave height between observation and hindcasts has to be explained. Fig. 9: For the lower and higher quantiles the chi plots have to be explained and discussed. Fig. 10: I do not find the category dependence "negative" and "zero"? Symbol and wind N to NNW is not necessary. The description of tables and figures should be improved.

English writing: Overall the writing style is good.

Some more suggestions:

- I do not find a clear definition of highest intensity, → page 34, row 2 and page 41,

row 9, does it mean only the dominant wind? Direction and/or speed? - I do not find a clear definition of negative bias: Systematically underestimated parameter? Minor improvements page 2 row 18 "This is", page 5/6 row 19/1 "Matlab" page 7 row 22 "also uses", page 8 row3 "….Good (1994)"page 14 row 14 "to the", page 16 rows 10-14 "Storm Emil" as well as page 18 rows 1 and 30. p.42, row 10: providing "us"?
* * *

---

## Author Response (AR1)

**General Comment**

*This paper addresses an important issue: the probability of marine storms characterized by the simultaneous presence of high waves and large storm surges. On the basis of two hindcast studies (one for waves and another for surges) it describes the dependence and the correlation between the two components of marine storminess at 32 points, which are located in correspondence of river mouths along the coastline of north and south Europe. I find the subject interesting and results potentially worth to be published. However, I recommend that the author improves his manuscript. Some, hopefully helpful, suggestions are in the specific comments here below. In fact, the paper needs major improvements for being publishable. It relies on intense/extreme events simulated by hindcast studies without providing sufficient information on their validation. The description of the statistical method should be more precise. The presentation of results should be improved by optimizing tables and figures. Causes of spatial variation of dependent and correlation should be better discussed. Parts containing details in a form similar to a technical report should be removed from the main body of the text.*

I truly thank the reviewer for his/her comments on the manuscript.

Next, I will address all referee's comments specifically.

**Specific Comments (Ref1)**

*(1)      Validation of hindcasts and its presentation*

*No sufficient attention is paid to assessing whether storm surge and wave simulations are capable of reproducing extremes values. Correlation and bias are not representative in this sense. Further, the presentation is strongly asymmetric between waves and surges, with a discussion of percent errors for surge and absolute errors for waves. Further, there is no information on the spatial distribution of storm surge errors.*

As a general comment:

The lack of observations (and especially those in compound mode) make hindcasts necessary. The scope of this paper has not been to re-validate or further validate the hindcasts of storm surge neither those of (significant) wave height. The validation of storm surge hindcasts have already been performed in Vousdoukas et al. (2016) whereas the validation of wave height hindcasts in Philips et al. (2017) in which the validation of the new ECMWF ERA5 reanalysis wave data set (including such long time series of waves) is documented. Both studies are falling in the category of peer-reviewed papers. Additional details referring to these relevant validations are provided below (by parts). Based on such validated data (hindcasts) over the main European coasts the statistical dependence analysis has been performed for 32 points of interest. Further, the study is mainly demonstrating the possibility of utilizing joint probability methods in coastal flood management by considering and putting emphasis on the statistical dependence between source variables. It also demonstrates that dependence, which is capable of modulating their joint return period, has to be estimated before the calculation of their (final) joint probability.

In brief: the main idea of this study has been to adopt a set of two already validated hindcasts (of surge and wave) and investigate over extreme compound events and their joint probability by utilising the so-called statistical dependence.

In addition, two new Supplements (Technical and Statistical) have been compiled and submitted providing explanations and clarifications to both technical and statistical issues.

*References*

*- Phillips, B.T., Brown, J.M. and Bidlot, J.-R. and Plater, A.J.: Role of Beach Morphology in Wave Overtopping Hazard Assessment. J. Mar. Sci. Eng. 5(1), 2017*

*- Vousdoukas, M.I., Voukouvalas, E., Annunziato, A., Giardino, A. and Feyen, L.: Projections of extreme storm surge levels along Europe. Clim. Dyn. 47(9): 3171-3190, doi:10.1007/s00382-016-3019-5, 2016.*

*Valid_Ref1_01: The paper needs to present information on the spatial distribution of percent errors in reproduction of high storm surges and waves (possibly of their extremes). In general, I suggest to use maps with percent errors, which are much more effective than tables to present such information. Without this, it is difficult to estimate how realistic conclusions are.*

As a general comment:

when low resolution models are used (as in this case) for reproducing time series of significant weather parameters, extremes cannot be captured with their exact (high-impact) value but in most cases only their footprints can be resolved (as extremes of a lesser value). A previous example can be seen in Petroliagis and Pinson (2012) where the footprints of extreme wind speed values over Bremen airport are captured by ERA-Interim (as footprint spikes) but they are considerably underestimated. In a similar approach, the scope of the study is to take (at least) into account such spikes (footprints) of extremes and study the statistical dependence of these spikes of storm surge and (significant) wave height.

Such footprints of extremes (resolved by hindcasts) can be found in Table 2 (Technical Supplement) where the 98.5% percentile extremes of storm surge observations are compared to their corresponding hindcast values (falling in the same 98.5% category). It becomes obvious that although hindcasts could not resolve the exact extremity of events at least their footprints were well captured. In a similar way in Table 3 (Technical Supplement) the footprints of significant wave height observation extremes are resolved by their corresponding hindcast (less intense) values.

It is important to point out that hindcasts above all were capable of identifying and resolving all seven (7) compound events that took place during the common time interval of 1,114 days.

On the same track, the set of storm surge hindcasts used in the current paper was already validated against 110 tidal gauge stations as described in Vousdoukas et al. (2016) reference paper. Vousdoukas et al. (2016) utilised both RMSE and relative (%) RMSE metrics. Overall, the model showed to reproduce satisfactory the measurements as shown in examples given in Figure 3 (Vousdoukas et al., 2016) over four tide-gauge stations in various coastal points of European coasts (Saint-Nazaire in France, Millport in UK, Hirsthals in Denmark and Rorvik in Norway). Studying closely Figure 3 it becomes obvious that hindcasts were able to simulate quite well the available set of observations capturing also efficiently local extremes. Further, the period of validation (2008-2014) had been characterized by an increased marine storm activity including high impact events as mentioned in Bertin et al. 2014; Breilh et al. 2013; Met Office and Centre for Ecology and Hydrology 2014; Vousdoukas et al. 2012.

Referring to the suggestion of using percent maps a new reference in text will be made pointing to Figure 4 (Vousdoukas et al., 2016) scatter plot showing RMS error in m (a) and as a percentage of the SSL (Storm Surge Level) range (b) for all the available tidal gauge stations.

Concerning the validation of wave hindcasts, the set used in the study is considered as a validated set with further details to be provided in Philips et al. (2017). The data are based on a dedicated re-run of the European Centre for Medium-Range Weather Forecasts

(ECMWF) ECWAM Wave Model (ECMWF, 2016) Cycle 41R1 at 28-km resolution. The model is forced by a six hourly ERA-interim (Dee et al., 2011) wind field with no wave data assimilation. The effect of water level change and surface current due to tides and surge is neglected. This global hindcast set has been produced in preparation of the ECMWF next reanalysis (ERA5).

I will add in the main text a reference to Figure 4 (Vousdoukas et al., 2016) scatter plot showing RMS error in m (a) and as a percentage of the SSL range (b) for all available tidal gauge stations. This reference will be in harmonisation with Figure 2 (current study) that is referring to the validation of wave hindcasts (RMSE values).

*References*

*- Bertin X., Li K., Roland A., Zhang Y.J., Breilh J.F., Chaumillon E.: A modeling-based analysis of the flooding associated with Xynthia, central Bay of Biscay. Coastal Eng 94:80–89, 2014.*

*- Breilh J.F., Chaumillon E., Bertin X., Gravelle M.: Assessment of static flood modeling techniques: application to contrasting marshes flooded during Xynthia (western France). Nat Hazards Earth Syst Sci 13:1595–1612, 2013.*

*- Met Office, Centre for Ecology & Hydrology: The recent storms and floods in the UK. p 29, 2014.*

*- Petroliagis, T. I. and Pinson, P.: Early warnings of extreme winds using the ECMWF Extreme Forecast Index, Meteorol. Appl., 21, 171–185, 10 doi:10.1002/met.1339, 2014.*

*- Phillips, B.T., Brown, J.M. and Bidlot, J.-R. and Plater, A.J.: Role of Beach Morphology in Wave Overtopping Hazard Assessment. J. Mar. Sci. Eng. 5(1), 2017*

*- Vousdoukas M.I., Almeida L.P., Ferreira Ó.: Beach erosion and recovery during consecutive storms at a steep-sloping, mesotidal beach. Earth Surf Process Landforms 37:583–691, 2012.*

*- Vousdoukas, M.I., Voukouvalas, E., Annunziato, A., Giardino, A. and Feyen, L.: Projections of extreme storm surge levels along Europe. Clim. Dyn. 47(9): 3171-3190, doi:10.1007/s00382-016-3019-5, 2016.*

*Valid_Ref1_02: Errors in timing are important and are not discussed.*

This study is focused over maxima taken place over 12- and 24-hours based on 3-hour set of hindcast values. These kind of (timing) errors were investigated over Rhine River (NL) ending point and the overall conclusion has been that hindcasts were able to pick up similar (to observations) maxima during both the 12-and 24-hour intervals.

An extra investigation based on extreme values of observations (during the common time interval of 1,114 days) exceeding a variety of percentile values (for the RIEN of Rhine River) showed that both storm surge and their corresponding wave height hindcasts were able to

capture almost all of the 24-hour extremes on the same (correct) day but with a weaker intensity (i.e., with a correct footprint of lesser intensity).

I will include the results of this latter investigation concerning various percentile extremes in the Technical Supplement and I will add a relevant reference (to the Technical Supplement) in the main text.
* * *
*Valid_Ref1_03: In my view, the statement in the conclusions "the overall performance of both surge and wave hindcasts is considered satisfactory" is not documented in the results.*

As already stated above, both sets of hindcasts had already been validated (Vousdoukas et al., 2016, Philips et al., 2017). Emphasis was given if these two sets were suitable to allow someone to go the extra step of resolving correctly the type and strength of both correlation and statistical dependence. Such an investigation was performed over the ending point of Rhine River (NL) with very satisfactory results. The same approach (of estimating statistical dependence) was adopted for the rest of ending points of the study.

I will point out and stress (in the Introduction) the fact that both sets of hindcasts are considered to be (already) validated and provide the reader with the relevant references.
* * *
*Valid_Ref1_04: The local validation of maxima at the Rhine River ending point is very convincing. It is anyway not clear whether such good performance of the models can be extended to other selected stations. Is this validation possible in other stations in other parts of the domain so that reader can be convinced that results in terms of correlation and dependence are convincing across the domain?*

Although the results of this study are based on already validated hindcasts of surge and wave, it is not straightforward how these hindcasts could guarantee for the exact (correct) estimation of both correlation and dependence between source variables (in places other than the Rhine River ending point) but nevertheless, the results of this study represent the first step on this direction.

Further, I agree that such specific type of validation (referring to correlation and dependence estimation) should be extended to other ending points of the study by utilising appropriate sets of observations. I am afraid this could be proved quite difficult if not impossible due to the necessity of long-period co-existing (real-time) observations of surge and wave over the areas of interest.

For time being, the study is mainly demonstrating the possibility of utilizing joint probability methods in coastal flood management by considering and putting emphasis on the statistical dependence between source variables. It also demonstrates that dependence, which is capable of modulating their joint return period, has to be estimated before the calculation of their (final) joint probability.

*Valid_Ref1_05: Section 4.2 line 16-17 the statement "Overall, it seems that hindcasts in this case were able of resolving and estimating both the correct type and strength of correlation between source variables." Could this be better enlightened at least for the Rhine station where data are available? How do we assess what is the real correct statistical dependence between surge and waves?*

The real (correct) statistical dependence is estimated by utilising the formula of Equation 3 over a long set of real data (observations) of storm surge coming from a tide gauge and real data of wave height coming from a close by wave buoy. The tide gauge and wave buoy have to be relatively close for obvious reasons. Usually the tide gauge is in the vicinity of the port while the wave buoy is suited some kilometres offshore in front of the port.

Besides observations (that are limited in time length) hindcasts can be used as in our case. Storm surge hindcasts were compiled by the Delft3D-Flow hydrodynamic model pinpointing the position of various tide gauges whereas wave height hindcasts were made by another (wave) model (ECWAM of ECMWF) pinpointing the position of relevant close by wave buoys.

It should be evident by now that even if hindcasts might be missing the exact magnitude of the extremes mainly due to the limited (model) resolution the most important issue here is their ability to resolve and estimate the correct value of both correlation and dependence as it is estimated over real data (observations).

In the case of the RIEN of the Rhine River, the high level of agreement between the dependence estimated utilising (surge and wave) observations and the one utilising (surge and wave) hindcasts, points to the direction that hindcasts are capable of resolving both the correct type and strength of dependence between the source variables.

I will stress this point (how we access the real correct statistical dependence) in the main text by presenting the concept behind estimating similar (if not almost the exact) dependence values by utilising both observation and hindcast sets of data.

**(2)    Description of statistical methods**

*The description of the method should be clear also to a reader not familiar with the involved statistical methods. Some details appear confusing. Eventually, if clarifying them requires too much text, I suggest the author to publish it in the supplementary material. Here is a list of points that I recommend to clarify.*

As a general comment:

A Statistical Supplement has been compiled clarifying missing or confusing details.

*Stat_Ref1_01: Line 1 page 5 writes that a transformation is adopted (please describe it) to produce identical marginal distribution. Line 4 writes that a copula function is used to diminish the effect of different marginal distributions. The two statement do not appear consistent to me.*

The transformation refers to the separately ranking of observations and the division of each rank by the total number of observations. It is considered as a trivial methodology of obtaining identical marginal distributions with Uniform [0, 1] margins. The utilisation of the copula C function does exactly this. At the same time, copula C contains the complete information about the joint distribution of X and Y.

I will rewrite the paragraph providing the required information for consistency with additional clarifying details that will be available in the new Statistical Supplement.

… For obtaining identical marginal distributions, each set of observations is ranked separately and each rank is then divided by the total number of observations resulting in a data transformation with Uniform [0, 1] margins. At this point, it is convenient to consider the bivariate cumulative function $F(x, y) = \text{Prob}( X \leq x, Y \leq y )$ that describes the dependence between X and Y completely. The effect of different marginal distributions can be diminished by assuming the copula function C in the domain [0, 1] x [0, 1] such as:

$$F(x,y) = C\{F_x(x), F_x(y)\} \qquad (2)$$

where $F_x$ and $F_y$ can be any marginal distributions. Such utilisation of the copula function has the same effect as if observations were ranked separately and divided by the total number of observations. In addition,  the copula C contains the complete information about the joint distribution of X and Y and it is invariant to marginal transformation …
* * *
*Stat_Ref1_02: In eq (1) the dependence chi is defined for z\* (upper limit of the observations), while in eq (3) is defined for any generic level u. Please explain this apparent inconsistency.*

In eq (1), z\* represents the upper limit of the observations but after the data transformation to Uniform [0, 1] margins, this upper limit is equal to (becomes) 1. For completeness eq (4) is added providing the (final) estimation for statistical dependence (chi)

$$\chi = \lim_{u \to 1} \chi(u) \qquad (4)$$

I will rewrite the relevant paragraph to provide the required explanation and clarifying details in the new Statistical Supplement.

… Taken into account the upper limit of the observations (previously defined as $z^\cdot$ in Eq. 1 but now being equal to 1), the dependence measure $\chi(u)$ will be given by:

$$\chi = \lim_{u \to 1} \chi(u) \qquad (4) \dots$$

A necessary update to the numbering for the rest of equations will be applied …

*Stat_Ref1_03: The derivation of eq (3) does not appear straightforward to me. Please add a reference.*

The main reference of Coles et al. (2000) is mentioned in line 21 page 4. Subsection 2.1 contains only a brief description of the methodology that is described in details in Coles et al. (2000).

I will include the relevant reference at an earlier point in the new Statistical Supplement (Section 2).

… Details of deriving Eq. 3 can be found in Coles et al. (2000). Based on Eq. 3, a set of $\chi$ values can be evaluated at different quantile levels u . The selection of a particular level u corresponds to threshold levels (x*, y*) for the two different data series.
* * *
*Stat_Ref1_04: In eqs. (3-5) the relation between U, V, u and X, Y, x* is not provided in the text.*

I will provide details of the relation between all mentioned terms in eqs  (3-6) by rewriting and incorporating relevant statements in the new Statistical Supplement.

… In addition,  the copula C contains the complete information about the joint distribution of X and Y and it is invariant to marginal transformation. This means that C can be described as the joint distribution function of X and Y. Further, X and Y are transformed to the new variables U and V with Uniform [0, 1] margins. It follows that the dependence measure $\chi(u)$ for a given threshold u can be given by …
* * *
*Stat_Ref1_05: The way in which chi_bar (statistical dependence of asymptotically independent variables) is computed is not given. Distinction between chi and chi_bar is not well explained.*

The calculation of chi_bar is clearly mentioned in line 1 page 6. It refers to the methodology described in Coles et al. (2000). More details on chi_bar and examples of how differs from chi are given in Coles (2001).

I will rewrite the relevant paragraph and add references with examples in the new Statistical Supplement.

… Chibar (chi_bar) parameter refers to the statistical dependence of asymptotically independent variables whereas chi ($\chi$) refers to the statistical dependence of asymptotically dependent ones. Details on the estimation of chibar are documented in Coles et al. (2000) whereas examples and how to utilise ($\bar{\chi}$) can be found in Coles (2001). The  class of

asymptotic dependence appears to be the case in Literature, having reached a consensus that there is strong, although not overwhelming, evidence for asymptotic dependence between wave height and surge …

*References*

- Coles, S.G., Heffernan, J. and Tawn, J.A.: Dependence measures for extreme value analyses. Extremes, 2, 339-365, 2000.

- Coles, S.G.: An Introduction to Statistical Modelling of Extreme Values. Springer Series in Statistics. Springer Verlag London. 208p, 2001
* * *
*Stat_Ref1_06: The concept of asymptotic dependence is not explicitly stated.*

The concept of asymptotic dependence (chi) is stated with adequate details in the main reference of Coles et al. (2000).

In summary, chi is on the scale [0, 1] with the set of values (0, 1] corresponding to asymptotic dependence whereas the measure chibar falls within the range [-1, 1] with the set of values [-1, 1) corresponding to asymptotic dependence. That is why the complete pair of chi and chibar is required as a summary of extremal dependence:

- chi > 0 & chibar = 1 reveals asymptotic dependence, in which case the value of chi determines a measure of strength of dependence within the class.

- chi = 0 & chibar < 1 reveals asymptotic independence, in which case the value of chibar determines the strength of dependence within the class.

Based on the main reference of Coles et al. (2000), I will incorporate the main concept behind asymptotic dependence in the new Statistical Supplement.

… The  class of asymptotic dependence appears to be the case in Literature, having reached a consensus that there is strong, although not overwhelming, evidence for asymptotic dependence between wave height and surge (Wadsworth et al., 2017).

The concept of asymptotic dependence ($\chi$) is stated with adequate details in Coles et al. (2000). In brief, $\chi$ is on the scale [0, 1] with the set (0, 1] corresponding to asymptotic dependence whereas the measure chibar ($\bar{\chi}$) falls within the range [-1, 1] with the set [-1, 1) corresponding to asymptotic independence. That is why the complete pair of $\chi$ and $\bar{\chi}$ is required as a summary of extremal dependence:

   - $\chi$ > 0 & $\bar{\chi}$ = 1 reveals asymptotic dependence, in which case the value of $\chi$ determines a measure of strength of dependence within the class

   - $\chi$ = 0 & $\bar{\chi}$ < 1 reveals asymptotic independence, in which case the value of $\bar{\chi}$ determines the strength of dependence within the class …

*Stat_Ref1_07: It is not described how correlation is computed. Is it correlation between time series of hourly (or 3-hourly or 6-hourly) values of surge levels and wave height?*

Both correlation and dependence estimations refer to maximum values during 12- or 24-hour time intervals. This is mentioned in Section 1 (Introduction) where the definition of max12 (maxima over a time interval of 12 hours) and max24 (maxima over a time interval of 24 hours) are introduced for the first time.

I will introduce and stress appropriately the definition of both max12 and max24 intervals in the Results Section.

... Referring to the full span of hindcasts, analytical maps and tables have been assembled  containing to both correlation and dependence values between surge and wave over the 32 RIEN points considered in this study. Both correlation and dependence values were estimated over maximum values of surge and wave during 12- and 24-hour intervals (labelled as max12 and max24 respectively) ...
* * *
*Stat_Ref1_08: Is correlation between the sequences of daily maxima? Between the sequence of maxima in 12 hours long windows?*

As mentioned above (Stat-Ref1_08), both types of correlations have been estimated. Correlation values over daily maxima are referred as max24 whereas correlations over 12-hour interval (half-day) maxima are referred as max12. This separation has been kept for both correlation and dependence estimations throughout the paper.
* * *
*Stat_Ref1_09: Provide a precise definition of definition of compound events as adopted in this study.*

Compound events of surge and wave are those events that coincidently are above a certain joint upper percentile criterion (here playing the role of a critical threshold).

I will add this definition in Section 1 (Introduction) for clarity reasons.

... These interactions are generally referred to as coincident or compound events (IPCC, 2012). In the current Part I, compound events of surge and wave are those events that coincidently are above a certain upper percentile criterion (representing a critical threshold) ...
* * *
*Stat_Ref1_10: Clarify the criterion leading to the selection of top 80 events.*

The selection is defined by the parameter alpha (a) representing the annual maximum non-exceedance probability taken equal to 0.1 following Defra TR3 Report suggestions. Such a

value (0.1) of alpha corresponds to ~2.3 compound POT (Peaks-Over-Threshold) events per year exceeding the corresponding optimal selected percentile threshold (the one providing ~2.3 compound events).

Since both surge and wave time series are almost 35 years long this points to ~80 (~2.3 x 35) events over the total time period.

I will add a more detailed explanation (Section 6) in the new Statistical Supplement taking into consideration the basic guidelines documented in Defra TR3 Report (2005).

**6 Selection of criterion thresholds resulting in the consideration of top-80 events**

Since values of dependence ($\chi$) can be estimated for any lower or upper threshold, initial trials were performed studying the behaviour of $\chi$ over a wide range of thresholds. Findings were similar to those contained in Defra TR3 Report (2005), justifying the selection of an optimal threshold for "alpha" ($\alpha$) equal to 0.1 corresponding to an annual maximum being exceeded in 9 out of 10 years (see Sect. 2.2 of the main text for details). This value (0.1) of alpha was considered for both mat_chi and mat_chibar routines when utilising POT (Peaks-Over-Threshold) methodology resulting in an annual maximum of ~2.3 compound events.

Such an annual threshold of ~2.3 events corresponds to the top 80 (Top-80) compound events taking place during any (POT separated) day of the total 12,753 days and it was dictated mainly by two factors: the threshold had to be low enough to allow a sufficient number of data points to exceed it for estimating dependence reliably, while being high enough for the data points to be regarded as extremes.

*References*

*- Defra TR3 Report by Svensson, C. and Jones, D.A.: Joint Probability: Dependence between extreme sea surge, river flow and precipitation: a study in south and west Britain. Defra/Environment Agency R & D Technical Report FD2308/TR3, 62 pp. + appendices (http://evidence.environment-agency.gov.uk/FCERM/Libraries/FCERM_Project_Documents/FD2308_3430_TRP_pdf.sflb.ashx), 2005.*

*Stat_Ref1_11: In Chapter 2.2, after the discussion, I cannot find the information on the values of alpha and u actually used in this study.*

The old Section 2.2 (now Section 3 in the new Statistical Supplement) contained the main theoretical concept behind the alpha (the annual maximum non-exceedance probability) and u parameter (percentile) threshold values. It was Section 4.1 (lines 20 to 28 of page 20) that an extensive explanation about the selection of alpha ($\alpha$) value being equal to 0.1 was documented. Now lines 20 to 28 (of page 20) have moved in Section 6 of the new Statistical Supplement. Based on such predefined alpha value the selection of an optimal threshold

percentile (u) is straightforward. Further, alpha (α) is capable of modulating the (optimal) percentile threshold (u) in such way to allow ~2.3 compound events (of 80 in total) to take place on a yearly basis.

Such information is now contained in Section 6 of the new Statistical Supplement being in harmony with Stat_Ref1_10 (see previous comment).
* * *
*Stat_Ref1_12: Page 4 lines 16 The statement "hydro-meteorological analyses based on real data often lead to an assessment of complete independence that could result to an underestimation of the joint probability of concurrent extreme events" is written in an ambiguous form. Please explain how joint probability is underestimated if data are "real" and the analysis is correct.*

I will rewrite the statement and add required details clarifying ambiguous terms. Examples of under- and over-estimating joint probabilities are also included in the new Statistical Supplement (Section 7).

... Similarly, if the extreme observations of one variable exceed a given threshold but the other variable produces lower observations than would normally be expected, this indicates negative dependence ($\chi = -1$).

In practice,  ... in tidal and estuarine environments, assessing the probability of flooding from the joint occurrence of both high storm surge and high wave values is not an easy process, as high surges and waves might be related to the same prevailing meteorological conditions, thus independence cannot and should not always be assumed. For instance, if we assume independence between input variables, this might underestimate considerably the likelihood of flooding (estimated by the product of their individual probability) resulting in higher risk for the coastal community. Similarly, assuming total dependence could be too conservative ...
* * *
*Stat_Ref1_13: I am confused by section 2.2 (which I fail to follow concerning the selection of the chi value) and section 2.4. Establishing a confidence interval (section 2.3) should be sufficient for assessing the significance of the computed dependence values. Is here a duplication of information?*

The old Section 2.2 (now Section 3 of the Statistical Supplement) was referring to the selection of an optimal percentile threshold (u) based on the annual maximum non-exceedance probability alpha (a). Then the estimation of dependence ($\chi$) was straightforward as described analytically in the old Section 2.1 (now Section 2 of the new Statistical Supplement). Further, the old Section 2.3 (now Section 4 of the new Statistical Supplement) was referring to the estimation of 5% significance level using a permutation method whereas the old Section 2.4 (now Section 5 of the new Statistical Supplement) was

referring to the estimation of confidence levels. All related values (significance and lower & upper confidence levels) are now contained in Table 4 and Table 5 (of the new Technical Supplement) following a similar approach as the one documented in TR1 Defra Report.

I will move for clarity reasons the old Section 2.1, Section 2.2, Section 2.3 and Section 2.4 to the new Statistical Supplement whereas I will move the old Table 3 and Table 5 to the new Technical Supplement stressing the difference between assessing the significance and confidence intervals based on the methodology documented in the relevant reference (TR1 Defra Report).

*References*

*- Defra TR1 Report by Hawkes, P.J. and Svensson, C.: Joint probability: dependence mapping & best practice. R & D Final Technical Report FD2308/TR1 to Defra. HR Wallingford and CEH Wallingford, U.K. (http://evidence.environment-agency.gov.uk/FCERM/Libraries/FCERM_Project_Documents/FD2308_3428_TRP_pdf.sflb.ashx), 2005.*

**(3)    Spatial variations of dependence and correlation, interpretation of results**

*The discussion of the spatial distribution of correlation and dependence and explanation for the differences is rather inconclusive. The author writes that "dependence is likely to occur when different processes are linked to some common weather (forcing) conditions" but no convincing investigation is made on that respect. Lack of dependence could for instance be explained by a substantial contribution of inverse barometer effect to storm surges, but there is no mention of this in the paper.*

As a general comment:

Storm surge is an abnormal rise of water generated by a storm, over and above the predicted astronomical tide values (http://www.nhc.noaa.gov/surge/faq.php). In observations mode, storm surge is calculated as a residual by subtracting harmonic tidal predictions from the observed sea level (Horsburgh and Wilson, 2007). Such "residual" may contain surge, tide-surge interaction, harmonic prediction errors and timing errors. Tide-surge interaction, harmonic prediction errors and timing errors are not taken into consideration in this study. On the other hand (e.g. in hindcast mode) a similar "residual" refers to the genuine meteorological contribution to sea level that represents the storm surge term. It should pointed out that the effect of wind and atmospheric pressure (inverse barometric effect) are contained in both the "residual" and storm surge terms. Based on this, it becomes clear that all data (storm surge) sets used in the study contain the effect of the inverse barometric effect besides the effect due to wind. This is the reason why the dedicated model (Delft3D-Flow) uses as input both ERA-Interim wind and pressure fields.

I will add the work of Horsburgh and Wilson (2007) in the list of References (shown below):

Horsburgh, K. J. and Wilson, C.: Tide-surge interaction and its role in the distribution of surge residuals in the North Sea, J. Geophys. Res., 112, C08003, doi:10.1029/2006JC004033, 2007.

Below I am addressing specifically the points.
* * *
*Spat_Ref1_01: Figures with the spatial distribution of correlation and dependence would be very useful. I suggest to replace the corresponding tables with maps*

Since the spatial distribution of both correlation and dependence should be displayed, a necessary change in the old Figure 10 (now Figure 9 in the main text) has been made to incorporate both correlation and dependence values using the same seven (7) relevant / reference categories. The exact values of correlation and dependence contained in the old Tables 3 to 6 have moved in the new Technical Supplement as Tables 4 to 7.

I will include the spatial distribution of both correlation and statistical dependence in the new Figure 9 (old Figure 10) shown below utilising seven (7) relevant / reference categories. Prevailing and dominant winds are to be left out of this new Figure 9 for more clarity. Their exact details contained in the old Table 7 can be found in the new Table 3.
* * *
[Figure]

*New Figure 9 (Old Figure 10)*

*Spat_Ref1_02: Section 4.5 does not provide interesting interpretation of results. Interpretation of results in term of understanding factors leading to compound events is not provided. Further annotation in figure 12 is not readable. Interpretation of results at the Rhone river mouth does not account for the possibility that many surge events are produced by inverse barometric effect and not by winds.*

In Section 4.5 an effort has been made to assess the low-level flow characteristics during critical compound events and not to provide a thorough explanation of the exact conditions leading to such compound events (that is beyond the scope of the paper). It also seems logical for someone to expect that in cases of surge events driven by winds this should be well captured in the corresponding wind roses relating to such prevailing and dominant (climatological) winds. Referring to the inverse barometric effect (as explained previously), all hindcast (storm surge) sets used in the study contain the effect of the inverse barometric effect besides the effect due to wind. This is the reason why the model (Delft3D-Flow) used for the production of hindcasts had as input both ERA-Interim wind and pressure fields.

I will add and explain accordingly that a thorough understanding of all factors leading to a compound event is above the scope of this study and I will skip Figure 12 (since main characteristics of both prevailing and dominant winds were contained in the old Table 7 (now in the new Table 3).
* * *
*Spat_Ref1_03: At some stations, wind during compound events is blowing offshore. Local high waves are unlikely caused by those winds.*

Combined events had to be de-clustered if they lasted longer than 24 hours. This means that a compound event lasting more than one day had to be counted as one (1) event even if this event could have lasted for a few days due to an approaching storm (barometric low). An example of such a compound event lasting for three consecutive days can be seen in Table 2 and Table 3 of the new Technical Supplement (referring to the time interval between 2 to 4 January 2012). After de-clustering this event will count only once and it will refer to its first date (4 Jan 2012) since after the necessary de-clustering all cases of compound events are referring to the first day of the event (the first day that both storm surge and wave height found to be above a predefined critical threshold). With such an approach, a compound event is considered only once and no other (another) event is taken into account for the next three days (even if the same event of day 1 continues to exist). Both prevailing and dominant directions are referring to the time of maximum daily wind intensity and if we consider the most common case of an approaching barometric low (storm) from the west the wind in the beginning is more WSW whereas with the passage of the storm tends to veer to a more northwest (northern) direction. I have checked the validity of this during the second, third and even the fourth day of an extended compound event and such a distinct veering is true.

Another important point is that not only an incoming onshore perpendicular wind leads to a significant storm surge or even to compound event. As an example Mistral (of north direction) that is heading to the open sea – Marin (of south direction) that is heading toward the coast of Marseille are capable of producing extreme storm surge events of equal intensity (during

distinct periods of rough seas) meaning that there exist other directions as well besides the ones blowing perpendicular to the coast relating to extremes as well.

I will refer and stress this unavoidable disagreement due to the veering of the wind and provide necessary explanations for such discrepancies in the main text (as analysed above).

... Details of clima and Top-80 flow characteristics are contained in Table 7. A possible exploitation of such information referring to both prevailing and dominant low-level flow characteristics should be considered significant and kept in mind when such extreme events possibly driven by intense storm outbreaks are anticipated over the area of interest (in forecast mode) ...

... Not all prevailing and dominant directions contained in Table 7 fall in the perpendicular onshore category. Especially for the RIEN points of the south North Sea, wind directions appear to be more SWS instead of rather more northerly directions and this is because combined events had to be de-clustered. This means that a compound event lasting more than one day had to be counted as one (1) event even if this event could have lasted for a few days. After this necessary de-clustering all cases of compound events, are referring to the first day of the event (the first day that both storm surge and wave height found to be above a predefined critical threshold). With such an approach, a compound event is considered only once and no other (another) event is taken into account for the next three days (even if the same event continues to exist). Both prevailing and dominant directions are referring to the maximum daily intensity and if we consider the most common case of an approaching barometric low (storm) the wind in the beginning is more WSW whereas with the passage of the storm tends to veer to a more north-western (northern) direction ...

*Spat_Ref1_04: Actual definition of prevailing and dominant wind is not clear to me (page 3, line 29 30).*

Prevailing Wind is the most common wind direction over an area, i.e., the direction of wind with the highest frequency (AMS, 2017), whereas Dominant Wind is the direction of the strongest wind that might blow from a different direction than the prevailing wind, i.e., from a less common direction (Thomas, 2000). The periods most frequently used for the estimation of prevailing and dominant winds are the observational day, month, season, and year. Methods for determination vary from a simple count of periodic observations to the computation of a wind rose.

I will provide definitions of both prevailing and dominant wind (as presented above) and add relevant references.

Old Table 3, Table 4, Table 5 and Table 6 can be moved to the new Technical Supplement (as Table 4, Table 6, Table 5 and Table 7 respectively). The relevant discussion points relating

to the main characteristics among various dedicated statistical packages are to move to the new Statistical Supplement.

The main results referring to both correlations and dependence (contents of the new Table 4, 5, 6 & 7 of Technical Supplement) are now contained in the new Figure 9 (old Figure 10) in graphical mode.

I will move Table 3, Table 4, Table 5 and Table 6 to the new Technical Supplement. I will make all necessary changes to old Figure 10 (new Figure 9) to include the main results referring to both correlation and dependence values.

*(5)      Other points and technical corrections:*

*Points_Ref1_01: Table7 is redundant with respect figure 7.*

Most probably meant Figure 12 (instead of figure 7) since Figure 12 refers to the main elements of Table 7 (in graphical mode).

I will skip Figure 12 and keep old Table 7 (new Table 3) in the main text that contains all relevant information of prevailing and dominant winds that was graphically presented in Figure 12 (upper and lower panels) over the selected 32 RIEN points.

*Points_Ref1_02: Figure 8 wind rose and related annotation in this figure redundant in my opinion.*

Most probably meant Figure 11 (wind roses over the ending point of river Rhone). I trust that this set of the two wind roses (in "clima" and in "Top-80" extreme mode) are necessary for the reader to get a feeling of the difference between prevailing and dominant wind as captured in a wind rose diagram.

Further, after skipping Figure 12, I strongly believe that at least an example of a wind rose diagram should remain for explanatory and demonstrating reasons to the reader.

Taken into account the deletion of Figure 12, I will keep old Figure 11 (new Figure 10 in the main text) as an example of wind rose diagram and reference point of how to differentiate prevailing from dominant wind conditions.

*Points_Ref1_03: I failed to find the "Defra/Environment Agency R&D Technical Report FD2308/TR3 on-line. I recommend the web link for downloading this and other technical reports to be provided in the reference list.*

I will include the web links referring to all Technical Reports contained in the list of references as shown below:

- Australian Rainfall & Runoff Project 18: Coastal Processes and Severe Weather Events: Discussion Paper, Water Technology report to Australia Rainfall & Runoff (2009) referring to the report of Department of Science, IT, Innovation and the Arts – Science Delivery (October 2012) "Coincident Flooding in Queensland: Joint probability and dependence methodologies" (https://www.longpaddock.qld.gov.au/coastalimpacts/inundation/coincident_flood_technical_review.pdf), 2009.

- Defra TR0 Report by Hawkes, P.J.: Extreme water levels in estuaries and rivers: the combined influence of tides, river flows and waves. R & D Technical Report FD0206/TR1 to Defra. HR Wallingford, U.K., (http://randd.defra.gov.uk/Document.aspx?Document=FD0206_5270_TRP.pdf), 2003.

- Defra TR1 Report by Hawkes, P.J. and Svensson, C.: Joint probability: dependence mapping & best practice. R & D Final Technical Report FD2308/TR1 to Defra. HR Wallingford and CEH Wallingford, U.K. (http://evidence.environment-agency.gov.uk/FCERM/Libraries/FCERM_Project_Documents/FD2308_3428_TRP_pdf.sflb.ashx), 2005.

- Defra TR2 Report by Hawkes, P.J.: Use of joint probability methods in flood management: a guide to best practice. R & D Technical Report FD2308/TR2 to Defra. HR Wallingford, U.K. (http://www.estuary-guide.net/pdfs/FD2308_3429_TRP.pdf), 2005.

- Defra TR3 Report by Svensson, C. and Jones, D.A.: Joint Probability: Dependence between extreme sea surge, river flow and precipitation: a study in south and west Britain. Defra/Environment Agency R & D Technical Report FD2308/TR3, 62 pp. + appendices (http://evidence.environment-agency.gov.uk/FCERM/Libraries/FCERM_Project_Documents/FD2308_3430_TRP_pdf.sflb.ashx), 2005.

- Hawkes, P.J.: Use of joint probability methods for flood & coastal defence: a guide to best practice. R&D Interim Technical Report FD2308/TR2 to Defra. HR Wallingford, U.K. (http://www.estuary-guide.net/pdfs/FD2308_3429_TRP.pdf), 2004.

- Hawkes, P.J.: Use of joint probability methods for flood & coastal defence: a guide to best practice. R&D Interim Technical Report FD2308/TR2 to Defra. HR Wallingford, U.K. (http://www.estuary-guide.net/pdfs/FD2308_3429_TRP.pdf), 2004.

- IPCC: Managing the Risks of Extreme Events and Disasters to Advance Climate Change Adaptation. A Special Report of Working Groups I and II of the Intergovernmental Panel on Climate Change [Field, C.B., V. Barros, T.F. Stocker, D. Qin, D.J. Dokken, K.L. Ebi, M.D. Mastrandrea, K.J. Mach, G.-K. Plattner, S.K. Allen, M. Tignor, and P.M. Midgley (eds.)]. Cambridge University Press, Cambridge, UK, and New York, NY, USA, 582 pp (https://www.ipcc.ch/pdf/special-reports/srex/SREX_Full_Report.pdf), 2012.

- Petroliagkis, T.I., Voukouvalas, E., Disperati, J. and Bidlot, J.: Joint Probabilities of Storm Surge, Significant Wave Height and River Discharge Components of Coastal Flooding Events, JRC Technical Report EUR 27824 EN, doi:10.2788/677778, http://publications.jrc.ec.europa.eu/repository/bitstream/JRC100839/lbna27824enn.pdf, 2016.

- Svensson, C. and Jones, D.A.: Dependence between extreme sea surge, river flow & precipitation: a study in south & west Britain. R&D Interim Technical Report FD2308/TR3 to Defra. CEH Wallingford, UK (http://evidence.environment-agency.gov.uk/FCERM/Libraries/FCERM_Project_Documents/FD2308_1135_INT_pdf.sflb.ashx), 2003.

*Points_Ref1_04: Page 7, line 6 graphically or empirically?*

The correct one is graphically using Hazen's (1914) formula (a reference that to be added in the list of References).

I will keep the (correct) term "graphically" and I will add the relevant paper (Hazen, 1994) to the list of references.

I will skip the reference (personal communication) whereas I will refer to the necessary details of data validation as explained below:

The reason is that even if model resolution does not seem capable of simulating local coastal topographical details, the main characteristics of the large-scale wave evolution are expected to be captured ( based on the validation data used for compiling Fig. 2).

*Points_Ref1_08: Table 1 is not needed in the main body of the text Figure1 provides the same information.*

Table 1 in contrary to Figure 1, contains the exact names of the RIEN (River Ending) points (with lat / lon) whereas in Figure 1 only the names of the rivers and in most cases these names are different from the names of the RIENs.

Since such topographical details will help the reader to locate easier and as close as possible the points of interest (RIENs), Table 1 could be move to the Technical Supplement.

I will move Table 1 to the Technical Supplement. It will be referenced as shown below:

Additional details can be found in Table 1 of the Technical Supplement containing the exact location (lat, lon) of RIEN points.

*Points_Ref1_09: I do not find a clear explanation on which data are grouped under the label hind_com, obs_com and hind_tot. One can guess but a clear description should be given in data and method.*

I will add a clear description of data (as new Table 1). See below for details:

First, the (Pearson) correlation between the two source variables (surge & wave) in observations mode is estimated while the same type of correlation is calculated in hindcast mode (see details in Table 1) for inter-comparison.

**(New) Table 1.** Details and abbreviations of main data sets used in the study.

| | |
|---|---|
| *obs_com* | Observations during the common period (1,114 days) |
| *hind_com* | Hindcasts during the common period (1,114 days) |
| *hind_tot* | Hindcasts during the total period (12,753 days) |

*Points_Ref1_10: Results section contains description of tools (lines 12-18, page 18). This should be moved to section 2 or 3, or (preferably in my view) removed or transferred to a supplement.*

Main parts of Section 4.1 (as most of the technical details contained in the old Section 2.1) could be moved to the new Statistical Supplement.

I will move main parts of Section 4.1 (and most of the technical parts of the old Section 2.1) to the new Statistical Supplement. See below details (referring to changes of Section 4.1 in the main text):

**4.1 Main tools for estimating statistical dependence**

The main tools for assessing dependence between surge and wave has been a set of matlab routines (mat_chi) for estimating the asymptotic behaviour of statistical dependent variables. Other Matlab routines such as mat_chibar (see details in  the Statistical Supplement) for assessing the asymptotic behaviour of statistical independent variables were also used and main findings are contained in  Table 4 and Table 5 of the Technical Supplement). Besides matlab functions additional routines from the statistical package R, namely "taildep" of module extRemes and "chiplot" of module evd (Extreme Value Distributions) were used for estimating and inter-comparing $\chi$ values.

~~Since values of $\chi$ can be estimated for any lower or upper threshold, initial trials were performed studying the behaviour of $\chi$ over a wide range of thresholds. Findings were similar to those contained in Defra TR3 Report (2005), justifying the selection of an optimal threshold for "alpha" (α) equal to 0.1 corresponding to an annual maximum being exceeded in 9 out of 10 years (see Sect. 2.2 for details). This value (0.1) of alpha was considered for both mat_chi and mat_chibar routines when utilising POT (Peaks-Over-Threshold) methodology resulting in an annual maximum of ~2.3 compound events. Such an annual threshold of ~2.3 events corresponds to the top 80 (Top-80) compound events taking place during any (POT separated) day of the total 12,753 days and it was dictated mainly by two factors: the threshold had to be low enough to allow a sufficient number of data points to exceed it for estimating dependence reliably, while being high enough for the data points to be regarded as extremes. Lastly, this threshold (~2.3 events) also proved optimal for providing quite stable dependence graphs. A full set of lag tests was performed for both correlation and dependence.~~ An optimal threshold of ~2.3 events on a yearly basis was found to provide quite stable dependence graphs (see details in the Statistical Supplement).  The maximum strength of almost any compound (surge and wave) event tends to take place during the same 24-hour (max24) time or during the same 12-hour (max12) period corresponding to zero-lag mode. Exceptions were found for Rhone, Ebro, Danube, Thames and Goeta RIEN points with one-day lag (2 half-days in case of max12), suggesting that storm surge values were (slightly) higher correlated with wave height values of the previous day. Results in Tables and Figures refer to zero-lag values.

*Points_Ref1_11: Fig.10 I cannot see the negative and zero dependence values that are mentioned in the text (page40, line 15).*

The old Figure 10 (new Figure 9) contained only dependence values (no correlations). Zero and negative values refer to a certain number of correlations contained in the old Table 3

and Table 5 (now moved to the new Technical Supplement as Table 4 and Table 5 respectively) valid for both max12 and max24 configurations.

In the new Figure 9 (old Figure 10) that now contains both correlations and dependence (max24) values, zero correlations are marked by a grey colour whereas negative correlations by a yellow one.

I will compile the new Figure 9 (old Figure 10) containing both correlation and dependence values. For more clarity, the prevailing and dominant components will be skipped since they are also presented analytically in the relevant old Table 7 (new Table 3) in the main text.

....

The new updated main text (manuscript) combined with the two new supplements (Statistical & Technical) has been uploaded as

nhess-2017-177-manuscript-version4

*The paper addresses compound events defined by combined high surges and high wind waves along European coastlines, especially in estuaries/river mouths. Statistical methods are used to investigate joint probabilities of compound events and the statistical dependency, since flood risk is not a function of one parameter (storm surges with peak value and duration) but usually of more (e.g. wind waves, river runoff). Large scale weather systems can cause either high storm surges or high wind waves and further more high precipitation and river runoff/discharges. Two sets of almost 35-year hindcasts of storm surges and wave heights were used to analyse the correlation and statistical dependency. As expected the frequency of the occurrence of the top compound events in different coastal areas were found to be higher during the winter months. In the introduction the hydrological and meteorological conditions for high wind waves and extreme tidal surge events which can occur simultaneously with extreme precipitation events and high river flows (compound events) leading to increased flood risk is highlighted clearly. But the paper and the used methodology focused only on very few parameters. What is the background of the generalization? The subject of the paper is interesting yet a little confusing especially in the context of coastal engineering therefor the manuscript should be major improved. The paper and its structure is not easy to understand and the description of different data sets (and different time spans) of observed and modelled hindcast data is confusing (e.g. a lot of unusual abbreviations). The number of tables and especially the huge amount of data should be reduced as they are displayed in figures. The selected 32 stations at the end of the rivers or estuaries cover a wide variety of geographical areas and meteorological, oceanographical and hydrological (currents and tides) systems in coastal zones along European coasts. E.g. the tidal range varies from nearly zero to some meters and within the deterministic part of compound events in comparison to the stochastic part (surges, wind waves and river flow) of these compound events. Further discussion of the deterministic and the stochastic part of the compound events and the effects in the statistical analyses (dependency of different parameters) is recommended (page 41, line 26-30). In general I agree completely with reviewer # 1!*

I truly thank the reviewer for his/her comments on the manuscript.

Next, I will address all referee's comments specifically.

**Comments (Ref2)**

*Com_Ref2_01: The description whether storm surge and/or wind waves are capable of reproducing extreme values is incomplete (e.g. river runoff?). It has to be explained, why river runoff is not taken into account!*

This study is the first part (Part I: storm surge and wave height) of investigating how statistical dependence can act as modulator referring to the joint return period of compound events. It is clear that this is the case of surge and wave events, so, no river runoff was taken into account. For the preparation of Part II (storm surge and river discharge) and Part III (wave height and river discharge) the effect of runoff will be included and be given special emphasis. I truly believe that such a separate investigation (by parts) allows for a deeper and better understanding of the different components contributing to a compound coastal event. Further, a study including all three components would have become too lengthy and difficult for the reader to follow.

I will explain in more detail (in the Introduction) the reasoning behind this separate investigation of the different components contributing to coastal compound events. I will also refer to the preparation of Part II (storm surge and river discharge) and Part III (wave height and river discharge).

…  The current work focuses on data preparation, parameter selection, methodology application and estimation of both correlation and statistical dependence between source variables. It also focuses on the prevailing (higher frequency) and dominant (higher intensity) low-level wind conditions over a set of preselected (top 80) extreme compound events. The critical time period during which such extremes take place is also analysed based on monthly frequency values of occurrence. The dependence analysis utilises 32 river ending points selected to cover a variety of geographical areas along European coasts. The variable-pairs presented in this report, which include enough information for calculations, are storm surge and wave height, relevant to most coastal flood defence studies. Two main time intervals were considered for the estimation of maximum values: the half-day interval (max12) and the one-day interval (max24) …

… This study represents the first part (i.e., Part I) of the investigation while Part II (storm surge and river discharge) and Part III (wave height and river discharge) are to follow. The reasoning behind such a separate investigation (by parts) is to allow the reader for a deeper and better understanding of the interaction between different components contributing to a compound coastal event.

*Com_Ref2_02: In the context of the paper a very interesting problem is discussed where copula functions should be taken into account, so far only a simple approach for copula functions has been taken into consideration, the discussion of different copula functions within the scope of the addressed topic is to be considered, more references to copula functions could be helpful (e.g. Wahl, T., Jain, S., Bender, J., Meyers, S. D., & Luther, M. E.*

*(2015). Increasing risk of compound flooding from storm surge and rainfall for major US cities. Nature Climate Change, 5(12), 1093-1097).*

The study follows the methodology proposed by Coles et al. (2000) where the basic theory behind the utilisation of an optimal copula function refers to Nelsen (1998), Joe (1997) and Currie (1999). I agree that the inclusion of more references as the suggested one, i.e., Wahl et al. (2015) definitely helps the reader to get more insight in the use of copulas when joint probability methodologies are taken into account.

I will include the suggested reference (Wahl et al., 2015). In addition, I will include the extra references of Nelsen (1998), Joe (1997) and Currie (1999) in the main text.

* * *
*Com_Ref2_03: A point of criticism is that the meteorological conditions and oceanographic system were not sufficiently described and the temporal developments of surges and wind waves are also not clearly described. E.g. in fig 10 and 12 it is shown that for the Weser (RIEN 29) the dependence for the prevailing (highest frequency) and dominant (highest intensity) wind during the top 80 extreme compound events are caused by wind direction from WSW. This is completely in contrast to my experience and has to be explained (same for e.g. RIEN 3, 12,: : : 23, 24, : : :. and 32)! From my point of view, it would be advisable to consider a subarea, e.g. only the North Sea, and after a successful investigation of the statistical dependence then implicate other areas.*

As a general comment: Directions falling in the WSW category do not count for the total percentage of the Top-80 events but besides this, there exists a logical explanation since the combined events had to be de-clustered. This means that a compound event lasting more than one day had to be counted as one (1) event even if this event could have lasted for a few days. An example of such a compound event lasting for three consecutive days can be seen in Table 2 and Table 3 of the new Technical Supplement (referring to the time interval between 2 to 4 January 2012). After de-clustering this event will count only once and it will refer to its first date (4 Jan 2012) since after the necessary de-clustering all cases of compound events are referring to the first day of the event (the first day that both storm

surge and wave height found to be above a predefined critical threshold). With such an approach, a compound event is considered only once and no other (another) event is taken into account for the next three days (even if the same event continues to exist longer than a day). Both prevailing and dominant directions are referring to the maximum daily intensity and if we consider the most common case of an approaching barometric low (storm) the wind in the beginning is more WSW whereas with the passage of the storm tends to veer to a more northwest (northern) direction. I have checked the validity of this during the second, third and even the fourth day of a compound event and such a distinct veering is true.

Another important point is that not only an incoming onshore perpendicular wind leads to a significant storm surge or even to compound event. As an example Mistral (of north direction) that is heading to the open sea – Marin (of south direction) that is heading toward the coast of Marseille are capable of producing extreme storm surge events of equal intensity (during distinct periods of rough seas) meaning that there exist other directions as well besides the ones blowing perpendicular to the coast relating to extremes as well.

I will stress this unavoidable disagreement due to the veering of the wind and provide necessary explanations for such discrepancy.

… Details of clima and Top-80 flow characteristics are contained in Table 7. A possible exploitation of such information referring to both prevailing and dominant low-level flow characteristics should be considered significant and kept in mind when such extreme events possibly driven by intense storm outbreaks are anticipated over the area of interest (in forecast mode) …

… Not all prevailing and dominant directions contained in Table 7 fall in the perpendicular onshore category. Especially for the RIEN points of the south North Sea, wind directions appear to be more SWS instead of rather more northerly directions and this is because combined events had to be de-clustered. This means that a compound event lasting more than one day had to be counted as one (1) event even if this event could have lasted for a few days. After this necessary de-clustering all cases of compound events, are referring to the first day of the event (the first day that both storm surge and wave height found to be above a predefined critical threshold). With such an approach, a compound event is considered only once and no other (another) event is taken into account for the next three days (even if the same event continues to exist). Both prevailing and dominant directions are referring to the maximum daily intensity and if we consider the most common case of an approaching barometric low (storm) the wind in the beginning is more WSW whereas with the passage of the storm tends to veer to a more north-western (northern) direction …

*Com_Ref2_04: (Length of observations/hindcasts) As I understood the water level data/storm surge/wind waves: The 32 RIEN (Table 1, page 10) were selected mainly because of their proximity to tidal gauges, although many of them cannot be evaluated due to lack of long-term measurements. For most RIENs, there are no data from nearby open wave buoys. Only for the Rhine (RIEN 28) are the tide and sea data (without*

*data gaps) available from a nearby wave buoy for a period of 3 years. The validation of the combined hindcasts (tide and wind waves) was done on the basis of measured data at the Rhine (NL) was done on the tidal data at Hoek von Holland (HvH), wave buoy: Lichteiland (LiG) over a period of ~ 3 years on measurement data without gaps and comparison of daily and half-day maxima. The generation of the hindcast of storm surge data was done with Delft3D-Flow (according to Vousdoukas et al. (2016) and the generation of the hindcast of the wind waves data was done with ECWAM wave model (according to Bidlot et al. (2006), Bidlot (2012), ECMWF (2015), Philips (2017)), e.g. ~36 years, wind- and pressure fields from ERA-Interim (ERAI) (time resolution: 1 h, spatial resolution: 28x28 km, fixed water level, signif. wave height, max. wave height, mean wave period, mean wave direction and validation based on available records from 101 wave buoys throughout Europe + North Atlantic (1996-2015) (Fig. 2)) The overlapping period of the two hindcasts (~ 35 years) was used in statistical analysis.*

*The methodology of the research (using the hindcast data sets and observed data) has to be explained more detailed and especially what that means for the interpretation of the results (for all 32 RIEN). A time series of observed water level and wave buoy of only 3 years and only for one station in the area at Hoek van Holland seems to me as being not sufficient and much too short for comparison/evaluation with the modelled (hind cast) data and the conclusions. There should much more field data (water level, surges, wind waves, river runoff) available around the 32 RIEN!*

As a general comment:

when low resolution models are used (as in this case) for reproducing time series of significant weather parameters, extremes cannot be captured with their exact (high-impact) value but in most cases only their footprints can be resolved (as extremes of a lesser value). A previous example can be seen in Petroliagis and Pinson (2012) where the footprints of extreme wind speed values over Bremen airport are captured by ERA-Interim (as footprint spikes) but they are considerably underestimated. In a similar approach, the scope of the study is to take (at least) into account such spikes (footprints) of extremes and study the statistical dependence of these spikes of storm surge and (significant) wave height.

Such footprints of extremes (resolved by hindcasts) can be found in Table 2 (Technical Supplement) where the 98.5% percentile extremes of storm surge observations are compared to their corresponding hindcast values (falling in the same 98.5% category). It becomes obvious that although hindcasts could not resolve the exact extremity of events at least their footprints were well captured. In a similar way in Table 3 (Technical Supplement) the footprints of significant wave height observation extremes are resolved by their corresponding hindcast (less intense) values.

It is important to point out that hindcasts above all were capable of identifying and resolving all seven (7) compound events that took place during the common time interval of 1,114 days.

On the same track, the set of storm surge hindcasts used in the current paper was already validated against 110 tidal gauge stations as described in Vousdoukas et al. (2016) reference paper. Vousdoukas et al. (2016) utilised both RMSE and relative (%) RMSE metrics.

Overall, the model showed to reproduce satisfactory the measurements as shown in examples given in Figure 3 (Vousdoukas et al., 2016) over four tide-gauge stations in various coastal points of European coasts (Saint-Nazaire in France, Millport in UK, Hirsthals in Denmark and Rorvik in Norway). Studying closely Figure 3 it becomes obvious that hindcasts were able to simulate quite well the available set of observations capturing also efficiently local extremes. Further, the period of validation (2008-2014) had been characterized by an increased marine storm activity including high impact events as mentioned in Bertin et al. 2014; Breilh et al. 2013; Met Office and Centre for Ecology and Hydrology 2014; Vousdoukas et al. 2012.

Referring to the suggestion of using percent maps a new reference in text will be made pointing to Figure 4 (Vousdoukas et al., 2016) scatter plot showing RMS error in m (a) and as a percentage of the SSL (Storm Surge Level) range (b) for all the available tidal gauge stations.

Concerning the validation of wave hindcasts, the set used in the study is considered as a validated set with further details to be provided in Philips et al. (2017). The data are based on a dedicated re-run of the European Centre for Medium-Range Weather Forecasts (ECMWF) ECWAM Wave Model (ECMWF, 2016) Cycle 41R1 at 28-km resolution. The model is forced by a six hourly ERA-interim (Dee et al., 2011) wind field with no wave data assimilation. The effect of water level change and surface current due to tides and surge is neglected. This global hindcast set has been produced in preparation of the ECMWF next reanalysis (ERA5).

I will add in the main text a reference to Figure 4 (Vousdoukas et al., 2016) scatter plot showing RMS error in m (a) and as a percentage of the SSL range (b) for all available tidal gauge stations. This reference will be in harmonisation with Figure 2 (current study) that is referring to the validation of wave hindcasts (RMSE values).

**6 Selection of criterion thresholds resulting in the consideration of top-80 events**

Since values of dependence ($\chi$) can be estimated for any lower or upper threshold, initial trials were performed studying the behaviour of $\chi$ over a wide range of thresholds. Findings were similar to those contained in Defra TR3 Report (2005), justifying the selection of an optimal threshold for "alpha" (a) equal to 0.1 corresponding to an annual maximum being exceeded in 9 out of 10 years (see Sect. 2.2 of the main text for details).

Such a value (0.1) of alpha was considered for both mat_chi and mat_chibar routines when utilising POT (Peaks-Over-Threshold) methodology resulting in an annual maximum of ~2.3 compound events.

Such an annual threshold of ~2.3 events corresponds to the top 80 (Top-80) compound events taking place during any (POT separated) day of the total 12,753 days and it was dictated mainly by two factors: the threshold had to be low enough to allow a sufficient

number of data points to exceed it for estimating dependence reliably, while being high enough for the data points to be regarded as extremes.

**Improvements (Ref2)**

*Impr_Ref2_01: The number of tables and graphs should be reduced and more summarized.*

I will reduce / convert tables and graphs to a more summarised form. List of changes in Figures and Tables are listed below:

- Table 1 will be moved in the new Technical Supplement to provide the reader with exact details of the selected RIENs (river ending points).

- New Table 1 (Details and Abbreviations of the main data sets) will be incorporated in the main text.

- Figure 9 (results based on statistical packages of R) will be moved in the new Statistical Supplement in Section 7 (Details and examples of the statistical packages used in the study)

- Table 3 will be moved in the new Statistical Supplement as new Table 4 (Section 3).

- Table 4 will be moved in the new Technical Supplement as new Table 6 (Section 3).

- Table 5 will be moved in the Technical Supplement as new Table 5 (Section 3).

- Table 6 will be moved in the Technical Supplement as new Table 7 (Section 3).

- Figure 12 will be skipped whereas old Table 7 will be kept as new Table 3 (in the main text) since it contains all relevant information of prevailing and dominant winds that was graphically presented in Figure 12 (upper and lower panels) over the selected 32 RIEN points.

– Old Figure 10 (now new Figure 9 in the main text) will include the spatial distribution of both correlation and statistical dependence (shown below) utilising seven (7) relevant / reference categories. Prevailing and dominant winds are to be left out of this new Figure 9 for more clarity. Their exact details contained in the old Table 7 can be found in the new Table 3.

[Figure]

*New Figure 9 (Old Figure 10)*

*Impr_Ref2_02: The paper is not easy to understand for a wide diversified audience, the length of the paper is too long and has partly too much redundancy (e.g. table 1 and fig. 1).*

I will reduce the length of the main paper by creating a separate Statistical Supplement and an additional Technical Supplement. These two new Supplements will help the reader to understand easier the main concept and findings of the current work. Redundant parts will be merged, shortened and improved.

*Impr_Ref2_03: The pure agreement between hindcast and observation of daily maximum of storm surges in Fig. 4 has to be explained.*

The pure agreement between hindcasts and observations is a clear indication of the model's (Delft3D-Flow) capability to simulate efficiently observations in hindcast mode having as input parameters (wind components and mean sea level pressure) from the ECMWF ERA-Interim reanalysis data set.

Indicative examples of such capabilities can be seen in Table 2 and Table 3 of Section 2 of the new Technical Supplement revealing that hindcasts above all were capable of identifying and resolving all seven (7) compound events (based on 98.5% percentile threshold) that took place during the common time interval of 1,114 days over HvH area of interest.

I will explain and stress this capability of Delft3D-Flow model of resolving daily maximum of storm surge observations in the main test referring also to Table 2 (Section 2) of the new Technical Supplement.
* * *
*Impr_Ref2_04: Why are small storm surges, e.g. below 0.5 m are taken in to account?*

In Figure 4, the capability of hindcasts to simulate correctly observations was done over the full range of observations, since it is important to show that model hindcasts are capable to perform well over any part of observations.

With the help of such models, it should be anticipated to have two validated sets of hindcasts resulting to the determination of the correct sign and strength of both correlation and statistical dependence.

I will point out that validation of both hindcast sets is done over the full spectrum of observations since the capability of the model to simulate correctly observations should refer to any part of the spectrum values.
* * *
*Impr_Ref2_05: What is the definition of a storm surge?*

*Storm surge is the abnormal rise in seawater level during a storm, measured as the height of the water above the normal predicted astronomical tide* https://oceanservice.noaa.gov/facts/stormsurge-stormtide.html.

Same wise the definition of significant wave height will be also included (see below).

*In physical oceanography, the significant wave height (SWH or Hs) is defined traditionally as the mean wave height (trough to crest) of the highest third of the waves (https://en.wikipedia.org/wiki/Significant_wave_height).*

I will include the definition of storm surge (and significant wave height) in the Introduction and provide the relevant (site) references.

*Impr_Ref2_06: What is the reason to use the storm between 25th December 2012 and 24th January 2013?*

It is just an example chosen for demonstrating how a compound event looks like and how it is related to the prevailing synoptic conditions (Storm Emil).

Further, it is an example of a compound event that lasts for three consecutive days (from 4 to 6 January 2012) as shown in Table 2 and Table 3 of the new Technical Supplement. During de-clustering this event will be counting only once and it will refer to its first date that this event took place (4 January 2012).

I will point out the concept of this multi-purpose demonstrating example and give emphasis in the de-clustering concept.
* * *
*Impr_Ref2_07: The pure agreement between hindcast and observation of daily maximum of the significant wave height in Fig. 6 has to be explained.*

As in the previous case (Imp_02_03), the pure agreement between hindcasts and observations is a clear indication of the model's (ECMWF / ECWAM) capability to simulate efficiently observations in hindcast mode having as input parameters (wind components) from the ECMWF ERA-Interim reanalysis data set.

Once again, indicative examples of such capabilities can be seen in Table 2 and Table 3 of Section 2 of the new Technical Supplement revealing that hindcasts above all were capable of identifying and resolving all seven (7) compound events (based on 98.5% percentile threshold) that took place during the common time interval of 1,114 days over HvH area of interest.

I will explain and stress this capability of ECMWF / ECWAM model of resolving daily maximum of significant wave height observations.
* * *
*Impr_Ref2_08: Fig. 8: The fairly pure agreement (chi) of the statistical dependence (chi) of storms surge and significant wave height between observation and hindcasts has to be explained.*

The fairly pure agreement between chi values estimated by observations (of surge and waves) and hindcasts (of surge and waves) is a clear indication that hindcasts were found capable of resolving and estimating both the correct type and strength of correlation and dependence between source variables.

I will point out the capability of the hindcasts to resolve and estimate the correct type and strength of correlation and dependence and stress the significance of such an agreement between dependence values estimated from observations and hindcasts.

*Impr_Ref2_09: Fig. 9: For the lower and higher quantiles the chi plots have to be explained and discussed.*

Values of dependence in the area of lower and higher quantiles seem (and somehow expected) to be quite unstable due the sparse of data.

I will explain and stress the behaviour of chi in lower and higher percentiles. Emphasis will be given on the stability of chi (graph) curves by identifying the area that dependence is clearly converging to a specific value (with no abrupt fluctuations).
* * *
*Impr_Ref2_10: Fig. 10: I do not find the category dependence "negative" and "zero".*

The old Figure 10 (new Figure 9) contained only dependence values (no correlations). Zero and negative values refer to a certain number of correlations contained in the old Table 3 and Table 5 (now new Table 4 and Table 5 of the new Technical Supplement) valid for both max12 and max24 configurations.

In the new Figure 9 containing both correlations and dependence (max24) values, zero correlations are marked by a grey colour whereas negative correlations by a yellow one.

I will produce the new combined Figure 9 (in place of the old Figure 10) containing both correlation and dependence values. For more clarity, the prevailing and dominant components will be skipped since they are also presented analytically in the old relevant Table 7 (now new Table 3 in the main text).
* * *
*Impr_Ref2_11: Symbol and wind N to NNW is not necessary.*

I will keep the 16 main and secondary directions of the wind (N – NNE – NE – ENE – E – ESE – SE – SSE – S – SSW – SW – WSW – W – WNW – NW – NNW).
* * *
*Impr_Ref2_12: The description of tables and figures should be improved.*

I will improve the description of both tables and figures accordingly. This will be also applied for the new updated Tables and Figures. A full description of the updated Tables and Figures is contained in author's reply to Impr_Ref2_01 comment (in improvements suggested by Ref 02 comments).

**Suggestions (Ref2)**
* * *
*Sugg_Ref2_01: I do not find a clear definition of highest intensity, page 34, row 2 and page 41, row 9, does it mean only the dominant wind? Direction and/or speed?*

Prevailing Wind is the most common wind direction over an area, i.e., the direction of wind with the highest frequency (AMS, 2017), whereas Dominant Wind is the direction of the strongest wind that might blow from a different direction than the prevailing wind, i.e., from a less common direction (Thomas, 2000). The periods most frequently used for the estimation of prevailing and dominant winds are the observational day, month, season, and year. Methods for determination vary from a simple count of periodic observations to the computation of a wind rose.

I will provide definitions of both prevailing and dominant wind and add the relevant references.

References

- AMS (American Meteorological Society) Glossary: Prevailing Wind. Glossary of Meteorology (Available online at http://glossary.ametsoc.org/wiki/Prevailing_wind_direction), 2017.

- Thomas, DG. 2000. Dictionary of physical geography. Blackwell.
* * *
*Sugg_Ref2_02: I do not find a clear definition of negative bias: Systematically underestimated parameter?*

Bias is the difference between the mean of the forecasts and the mean of the observations. It could be expressed as a percentage of the mean observation. Also known as overall bias, systematic bias, or unconditional bias (http://www.cawcr.gov.au/projects/verification/).

I will provide the definition and include the relevant (site) reference.
* * *
**Minor Improvements (Ref2)**
* * *
*Mimp_Ref2_01: page 2 row 18 "This is"*

I will correct it.
* * *
*Mimp_Ref2_02: page 5/6 row 19/1 "Matlab"*

I will correct it.

*Mimp_Ref2_03: page 7 row 22 "also uses"*

I will correct it.
* * *
*Mimp_Ref2_04: page 8 row3 "… Good (1994)"*

I will correct it.
* * *
*Mimp_Ref2_05: page 14 row 14 "to the"*

I will correct it.
* * *
*Mimp_Ref2_06: page 16 rows 10-14 "Storm Emil" as well as page 18 rows 1 and 30*

I will correct it.
* * *
*Mimp_Ref2_07: p.42, row 10: providing "us"?*

I will delete the word "us".

….

The new updated main text (manuscript) combined with the two new supplements (Statistical & Technical) has been uploaded as

nhess-2017-177-manuscript-version4

[revised manuscript text omitted]

---

## Author Response (AR2)

*Above all, I feel obliged to thank Referee 1 for his suggestions and continuous useful guidance.*

*I have read the answers of the author to my comments. Here are my further comments referring to the points listed in "nhess-2017-177-author_response-version2.pdf". I confirm that the paper is in my view interesting, the new structure has improved the readability of the article, but the new version remains affected by two shortcomings that I had already noticed in the first review:*

*a) the reliability of the hindcasts (particularly for waves and for intense/extreme events) is not convincingly documented*

*b) there insufficient effort to interpret the dependences that are found. On both these aspects I think that progresses are possible with a reasonable, but not huge effort.*

*As a general comment,*
*effort has been made to overcome both shortcomings. Details are given below.*

*Considering the specific answers of the author to my comment:*

**(1) Validation of hindcasts and its presentation**

*The goal of this paper is, indeed, not the validation of the hindcasts. However, the reliability of the analysis depends on the accuracy of the hindcasts, which is essential for this study. The minimal requirement is a documented validation of the two (storm surge and wave) hindcasts in previously published studies, which is what the author has partially added. However, the answer of the author is not fully clear on this respect, particularly as the published article validating the wave hindcast is concerned. I had formerly commented that "No sufficient attention is paid to assessing whether storm surge and wave simulations are capable of reproducing extremes values. Correlation and bias are not representative in this sense". I find in the revised version no new material answering to this request. The added information referring to Vousdoukas et al. (2016), in not relevant for intense and extreme events. Further absolute values are not representative of the actual importance of these errors. I think that values would be more informative if they are normalized with the values of the mean observation. Concerning waves, author refers to Phillips, B.T., Brown, J.M., Bidlot, J.-R. and Plater, A.J.: Role of Beach Morphology in Wave Overtopping Hazard Assessment. J. Mar. Sci. Eng. 5(1), 2017. I have checked this article, but I Have not found the validation of the wave hindcast that the author claim. I might have missed it. A specific indication by the author is required.*

*The description of the set of wave hindcasts used in the current study is contained in Page 5 (Line 8) of Phillips et al. (2017) as shown below:*

*"Here, we use data from a rerun of the European Centre Wave Model (ECWAM, [47]) Cycle 41R1 (28-km resolution) to provide a longer dataset. The model is forced by a six hourly ERA-interim wind field with no wave data assimilation. The effect of water level change and surface current due to tides and surge is negated. The output used is hourly $H_s$ (significant wave height) and $T_p$ (wave period) from 1979–2014 and was extracted from the offshore limit of the XBeach boundary. This global hindcast was produced in preparation of the ECMWF (European Centre for Medium Range Weather Forecasts) next reanalysis (ERA5)".*

*"The ECMWF wave hindcast exhibits an RMSE (root mean square error) of 0.31 m, a bias (the difference between the estimator's expected value and the true value of the parameter being estimated) of 0.06 m and a symmetric slope value of 1.11 for $H_s$ when validated against the Liverpool Bay Wave Buoy" …*

*Reference ECWAM, [47]. "European Centre for Medium Range Weather Forecasts. IFS Documentation | ECMWF; European Centre for Medium Range Weather Forecasts: Reading, UK, 2016".*
* * *
*In synthesis, considering the new version I remain unconvinced that the statement "the overall performance of both surge and wave hindcasts is considered satisfactory" is documented at a convincing level.*
* * *
*The set of storm surge hindcasts originated from Vousdoukas et al. (2016) have specifically used for projections of **extreme storm surge levels** along Europe and it seems as an appropriate dataset for the current paper. This is the main reason of utilizing such storm surge dataset in my current work. On the other hand, I have to admit that wave hindcasts based on the ERA5 significant wave reanalysis dataset used in my study have not been thoroughly tested as for the validity of their extreme values since ERA5 is still in production phase as it can be seen at*
*https://www.ecmwf.int/en/newsletter/147/news/era5-reanalysis-production*
*https://www.ecmwf.int/en/about/media-centre/science-blog/2017/era5-new-reanalysis-weather-and-climate-data*

*Table 1: Details of the wave buoys used for the validation of extremes.*

[revised manuscript text omitted]

*It becomes obvious that although hindcasts could not resolve the exact extremity of both surge and wave events, they were able to capture their footprints quite well. It is important to point out that hindcasts above all were **capable of identifying and resolving all seven (7) compound events** that took place during the common time interval of 1,114 days.*

*Based on above data and results, a new Section (Additional validation of wave hindcasts focusing on extremes) has been compiled and incorporated in the Technical Supplement as Sect. 2.*
* * *
*If the paper main goal is demonstration a methodology, this may be an acceptable limitation, but some rephrasing, admitting it and suggesting therefore, caution on the results that are found, is advisable.*
* * *
*The current work focuses on data preparation, parameter selection, methodology application and estimation of both correlation and statistical dependence between source variables. It also focuses on the prevailing (higher frequency) and dominant (higher intensity) low-level wind conditions over a set of preselected (top 80) extreme compound events. The critical time period during which such extremes take place is also analysed based on monthly frequency values of occurrence.*

*Nevertheless, taking into account the percentage of hits (% of correct spikes) presented in the previous section – ranging from 80 to 85% – caution (as requested) has been added in both Section 4 (Discussion)*

*and in Section 5 (Conclusions) just after the reference to the **compound validation over HvH** river ending point as shown below:*

> *"Since such "compound" validation is impossible to be repeated for all RIEN points, some caution with the exact levels of correlation and dependence should be bear in mind for the rest of the RIEN points".*

*Other minor points in section 3*

*Section 3.1, line 6 depending on whether the hindcast includes the computation of nonlinear tide-surge interaction, the "residual" can include tide surge interaction as well.*

*Corrected (as suggested).*

*Section 3.1 Line 1 I think correct English expression would be "observation mode" … anyway note that I am not considering in this review typos.*

*Corrected (observation mode) throughout the text.*

**(2) Description of statistical methods**

*I am positive with the new version, which I think is more readable and useful to the reader. The section "statistical dependence" has now become very short, I suggest to merge it (as a separate subsection) with "Data and Methodology" in a single section.*

*This has been done. Statistical dependence has been incorporated into Data and Methodology that has become the new Joint Section 2 (Data and Methodology). Appropriate adaptations (to the text) have taken place also.*

**(3) Spatial variations of dependence and correlation, interpretation of results.**

*Obviously, the effect of atmospheric pressure (that in steady condition is described by the inverse barometer effect) is included in the model simulations. My comment was a suggestion that lack of dependence could be related to a large contribution of atmospheric pressure on storm surges, because waves are not affected by it. Large dependence could be related to the common action of wind on sea level and waves, depending on the characteristic of the fetch and on water depth along it. On this respect I think that the comment and*

*the added material does not answer to my request and the manuscript still would benefit from a discussion with an interpretation of results.*

*The potential effect of atmospheric pressure in cases of low dependence has been added in both Section 4 (Discussion) and Section 5 (Conclusions). It seems that in cases of high (surge & wave) dependence, winds may have the leading (forcing) role whereas in cases of low dependence the contribution of atmospheric pressure tends to become more pronounced.*

*I accept that "a thorough understanding of all factors leading to a compound event is above the scope of this study ", but some discussion should be added, anyway. Any indication on under which conditions dependence would be high and when low?*

*As it has mentioned above, it seems that in cases of high dependence, winds (of an approaching storm) most probably have the leading role whereas in cases of low dependence the contribution of atmospheric pressure has the tendency of becoming more pronounced.*

*Based on this, it seems that compound events over northern sea areas are mostly driven (forced) by a common extreme (storm) wind event resulting in a high value of dependence between surge and wave, whereas a large contribution of atmospheric pressure affecting only storm surge might be one among other reasons for low dependence (of surge and wave) dependence over southern sea areas.*

*As already mentioned, the potential effect of atmospheric pressure in cases of low dependence has been added in both Section 4 (Discussion) and Section 5 (Conclusions) – see new updated version of manuscript.*

I *suggest to split the final section "discussion and conclusion" in two sections: "discussion" and separated "conclusion". Reading it is not clear to me whether the goals of the paper is to demonstrate a methodology or to discuss dependences between storm surge and wave events at a representative set of European RIEN points.*

*This (suggested) split has been taken place as Discussion (Section 4) and Conclusion (section 5). All necessary adaptations have also taken care of.*

*Comments on figures*
*In general figures need to be improved*

*As a general comment*
*almost all Figures have been updated (improved) as requested.*

*Names in figure 1 will likely not readable when reduced too journal size*
* * *
*Names in Figure 1 (River Ending Points) have been updated to be visible when reduced in journal size.*
* * *
*Quality of figure 2 is poor. I expect no geographical name will be readable and dots will be hardly visible when reduced to standard journal size*
* * *
*Figure 2 (both bias & rmse) has been updated. Quality has significantly improved.*
* * *
*Graphic quality of fig 3 is low. It contains a lot of useless details (e.g. roads and motorways labels) and unreadable (and useless in this context) names of towns and locations.*
* * *
*Figure 3 (main site positions) has been updated. Useless details have been removed.*
* * *
*In figs 4 and 6 annotations are too small. They will be unreadable in the printed journal article. This problem is present in most figures.*
* * *
*Figure 4 and 6 (using matlab) have been updated with new annotations significantly bigger so to be readable in a journal article.*

*Figures 1, 7, 8, 9 and 10 have also been updated.*

*In addition, after the split of Figure 2 a re-numbering of figures had to take place.*

*A renumbering of Sections took place also due to the merging of old Section 2 with Section 3.*